# Knowledge Distillation of Uncertainty using Deep Latent Factor Model

**Sehyun Park**
Department of Statistics
Seoul National University
`ps_hyen@snu.ac.kr`

**Jongjin Lee**
Samsung Research
`ga0408@snu.ac.kr`

**Yunseop Shin**
Department of Statistics
Seoul National University
`dbstjq48@snu.ac.kr`

**Ilsang Ohn**
Department of Statistics
Inha University
`ilsang.ohn@inha.ac.kr`

**Yongdai Kim**[*]
Department of Statistics
Seoul National University
`ydkim0903@gmail.com`

## Abstract

Deep ensembles deliver state-of-the-art, reliable uncertainty quantification, but their heavy computational and memory requirements hinder their practical deployments to real applications such as on-device AI. Knowledge distillation compresses an ensemble into small student models, but existing techniques struggle to preserve uncertainty partly because reducing the size of DNNs typically results in variation reduction. To resolve this limitation, we introduce a new method of distribution distillation (i.e. compressing a teacher ensemble into a student distribution instead of a student ensemble) called Gaussian distillation, which estimates the distribution of a teacher ensemble through a special Gaussian process called the deep latent factor model (DLF) [2] by treating each member of the teacher ensemble as a realization of a certain stochastic process. The mean and covariance functions in the DLF model are estimated stably by using the expectation-maximization (EM) algorithm. By using multiple benchmark datasets, we demonstrate that the proposed Gaussian distillation outperforms existing baselines. In addition, we illustrate that Gaussian distillation works well for fine-tuning of language models and distribution shift problems.

## 1 Introduction

While DNNs have succeeded tremendously in various AI tasks, the rapid increase in their model sizes has raised a concern about high computational resource demands, which limits their applications to real world applications such as on-device AI [1], and thus developers have increasingly compressed large-scale language models into much smaller models [1, 2, 3, 4]. A representative tool for compression is knowledge distillation (KD), which constructs a smaller DNN that mimics a given large-scale DNN [5].

Another concern is that the inherent over-parameterization of a single DNN makes them susceptible to overfitting, leading to overconfident predictions. When training predictive models, it is essential to learn models not only accurate but also reliable. For reliable prediction, proper quantification of uncertainty has become an important topic in AI research [6, 7, 8]. Deep ensemble (an ensemble of multiple DNNs) has received much attention not only for its strong predictive performance but

---

[*]Corresponding author.
[2]The source code of DLF is publicly available at `https://github.com/sehyun1094/DLF`

39th Conference on Neural Information Processing Systems (NeurIPS 2025).

also for its ability to quantify prediction uncertainty [9, 10]. An ensemble of DNNs can mitigate overconfident predictions by reflecting the uncertainty (i.e., the variation of multiple predictions made by members of an ensemble) when making a final decision.

Since deep ensemble requires even more computational resources than DNNs, KD of deep ensemble is necessary for improving its applicability. Several works have focused on distilling a given teacher ensemble to a student ensemble instead of distilling a teacher ensemble to a single student DNN to keep the uncertainty as much as possible [8, 11, 12, 13, 14, 15]. Algorithms for KD of deep ensemble can be roughly categorized into two approaches: one-to-one distillation and distribution distillation. One-to-one distillation compresses each member in a teacher ensemble to a smaller DNN, which becomes a member of a student ensemble. Various weight-sharing architectures for student DNNs, along with their corresponding learning algorithms, have been proposed [11, 13, 14, 16]. On the other hand, distribution distillation treats each ensemble member in a teacher ensemble as an independent realization of a certain distribution whose parameters are modeled by a student DNN. [17] and [18] assume that the conditional class probability vector of each member in a teacher ensemble follows a Dirichlet distribution and devise a method to estimate the parameters in the Dirichlet distribution using a student DNN.

There are still limitations in existing KD methods for deep ensemble. One-to-one distillation methods tend to lose a significant amount of uncertainty in a teacher ensemble when they compress large DNNs into smaller ones, while performance of Dirichlet distillation (the distribution distillation with a Dirichlet distribution) is inferior to one-to-one distillation partly [11, 14] because of instability in learning the parameters in the Dirichlet distribution.

The aim of this paper is to propose a new distribution distillation method that is numerically stable in learning and superior to other baselines in uncertainty quantification. In our proposed method, we treat each member in a teacher ensemble as an independent realization of a Gaussian process and estimate the mean and covariance functions of the Gaussian process based on observed predictions of members in a teacher ensemble. For this purpose, we propose the deep latent factor (DLF) model where the mean and covariance functions are modeled by a student DNN and implement an EM algorithm to estimate the maximum likelihood estimator (MLE) of the student DNN. We call our method *Gaussian distillation*.

Our contributions are summarized as follows.

- We propose a new distribution distillation method based on a specially designed Gaussian process called the DLF model that achieves superior performance in uncertainty quantification to other baselines.

- We develop an EM algorithm to estimate the student DNN in the DLF model. In particular, we propose a way of finding a good initial solution by maximizing the penalized complete log-likelihood.

- We do numerical experiments to show that Gaussian distillation outperforms other baselines for both regression and classification. We also illustrate that Gaussian distribution is a useful tool for fine-tuning language models.

- We apply the pre-trained DLF to distribution shift problems and show numerically that it outperforms baselines.

## 2 Preliminaries

### 2.1 Prediction uncertainty

In a nutshell, quantifying prediction uncertainty in supervised tasks involves efficiently estimating the predictive distribution of the output $y$ given a new input denoted as $p(y|\boldsymbol{x}^{\mathrm{new}})$. The variation in the predictive distribution can be used as a measure of uncertainty in prediction.

A typical way of estimating the predictive distribution begins with a parametric generative model for the input and output pair. Let $p(y|\boldsymbol{x}, \theta)$ be the conditional distribution of the output $y \in \mathcal{Y}$ given an input $\boldsymbol{x} \in \mathcal{X} \subset \mathbb{R}^d$, where $\theta \in \Theta$ is an unknown parameter. Then, we try to estimate $\theta$ based on training data $(\boldsymbol{x}_1, y_1), \dots, (\boldsymbol{x}_m, y_m)$ such that $p(y|\boldsymbol{x}, \hat{\theta})$ is as close as possible to $p^*(y|\boldsymbol{x})$, where $\hat{\theta}$

is an estimate of $\theta$ and $p^*(y|\boldsymbol{x})$ is the true conditional distribution. For example, the MLE minimizes the empirical KL divergence between $p(y|\boldsymbol{x}, \theta)$ and $p^*(y|\boldsymbol{x})$.

It is well known, however, that the variation in $p(y|\boldsymbol{x}, \hat{\theta})$ is smaller than that in $p^*(y|\boldsymbol{x})$ because $p(y|\boldsymbol{x}, \hat{\theta})$ does not take into account the uncertainty in estimating $\hat{\theta}$. Thus, making a decision solely with $p(y|\boldsymbol{x}, \hat{\theta})$ would lead in overconfident results. A proper uncertainty quantification in prediction should consider not only uncertainty in $p^*(y|\boldsymbol{x})$ (aleatory) [6, 7] but also uncertainty in $\hat{\theta}$ (epistemic).

A popular way of considering both aleatory and epistemic uncertainties in prediction is to use an ensemble. We construct multiple estimates $\hat{\theta}_1, \ldots, \hat{\theta}_n$ of $\theta$ and then estimate the predictive distribution as $\hat{p}(y|\boldsymbol{x}) = \sum_{i=1}^n p(y|\boldsymbol{x}, \hat{\theta}_i)/n$, which we call the averaged prediction model. For deep learning, the two most representative methods of constructing multiple estimates are deep ensemble [9, 19, 20] and Bayesian DNNs [21, 22, 23, 24, 25, 26, 27]. Deep ensemble generates multiple estimates by learning a DNN with different initial parameter, while Bayesian DNNs generate $\hat{\theta}$s from the posterior distribution. In this paper, we focus on deep ensemble, but our proposed method can be applied to Bayesian DNNs without modification.

## 2.2 Review of ensemble distillation

As mentioned in Introduction, deep ensemble has an intrinsic limitation in its practical applications due to high computational costs and times along with demands for substantial memory to store and process multiple prediction models. To resolve this problem, KD of deep ensemble has received much attention [5, 8, 11, 12, 13, 14, 15, 17]. A basic idea of KD of deep ensemble is to approximate large DNNs in a teacher ensemble by smaller student DNNs. A naive approach of KD of deep ensemble is to approximate the averaged prediction model $\hat{p}(y|\boldsymbol{x})$ of a teacher ensemble by a small single DNN [15, 28, 29]. This naive approach, however, does not perform well since it is hard to distill the uncertainty in $\hat{p}(y|\boldsymbol{x})$ into a single student DNN.

A remedy is to distill a teacher ensemble into a student ensemble. Several methods have been proposed for this purpose, which can be roughly divided into two categories that are explained in the subsequent subsections.

### 2.2.1 One-to-one distillation

The main idea of one-to-one distillation is to construct multiple student models, each of which corresponds to each teacher model. That is for given $n$ many teacher models $p_i^{(t)}(y|\boldsymbol{x}), i = 1, \ldots, n$, $n$ many student models $p_i^{(s)}(y|\boldsymbol{x})$ are constructed. To save computation time and memory further, various special neural network architectures for $n$ student models $p_i^{(s)}(y|\boldsymbol{x}), i = 1, \ldots, n$ have been proposed. Examples are *Hydra* [11], *Batch Ensembles* (BE) and *Latent Batch Ensemble* (LBE) [12, 14]. See Appendix A.1 for details.

### 2.2.2 Distribution distillation

Distribution distillation assumes that teacher models are independent realizations of a stochastic model with unknown parameters modeled by a student DNN and estimates the student DNN based on the prediction values of the teacher models [17, 18]. To be more specific, for classification problems, we assume that $\left(p_i^{(t)}(y|\boldsymbol{x}), y = 1, \ldots, c\right), i = 1, \ldots, n$ for a given $\boldsymbol{x}$ are independently generated from the Dirichlet distribution with parameters $\alpha_1(\boldsymbol{x}), \ldots, \alpha_c(\boldsymbol{x})$ and model these parameters by a student DNN. Once the student DNN is learned, ensemble members are generated from the learned Dirichlet distribution and aggregated in the prediction phase. We call this method *Dirichlet distillation*. See Appendix A.2 for details.

## 3 The Proposed Method

We propose a new method of distribution distillation. The main idea of the proposed method is that we treat members in a teacher ensemble as independent realizations of a Gaussian process and estimate the mean and covariance functions of the Gaussian process by a student DNN. Then, in the inference

phase, we generate ensemble members from the estimated Gaussian process. We call our proposed method *Gaussian distillation*. See Figure 1 for the overall process of Gaussian distillation.

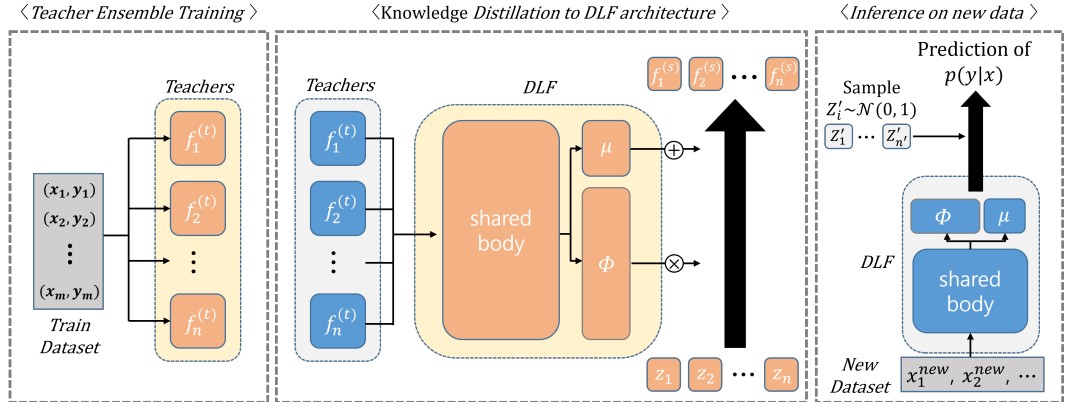

Figure 1: Overall process of Gaussian distillation

A technical difficulty of this idea is to model and estimate the covariance function. To resolve this problem, we use the DLF, which is an extension of the standard linear factor model [30], where the mean and factor loading are modeled by a student DNN.

For the probabilistic model of data, we consider $y = f(\boldsymbol{x}) + \epsilon$, where $\epsilon \sim \mathcal{N}(0, \sigma_\epsilon^2)$ for regression problems and $p(y|\boldsymbol{x}) = \exp(f_y(\boldsymbol{x}))/\sum_{v=1}^{c} \exp(f_v(\boldsymbol{x}))$ for classification problems. Thus, a teacher ensemble for regression problems consists of multiple teacher models for $f$ as well as multiple estimates of $\sigma_\epsilon^2$, while a teacher ensemble for classification problems consists of multiple teacher models for multivariate functions $\boldsymbol{f}(\cdot) = (f_1(\cdot), \ldots, f_c(\cdot))$.

## 3.1 Deep Latent Factor model

In this subsection, we introduce special Gaussian processes for $f(\cdot)$ and $\boldsymbol{f}(\cdot)$, respectively.

**Univariate case**   The DLF model for a univariate random function $f : \mathcal{X} \to \mathbb{R}$ is defined as

$$f(\cdot) = \mu_\theta(\cdot) + \Phi_\theta(\cdot)^\top \boldsymbol{Z}, \tag{1}$$

where $\mu_\theta(\cdot) : \mathcal{X} \to \mathbb{R}$ is the mean function, $\Phi_\theta(\cdot) : \mathcal{X} \to \mathbb{R}^q$ is the factor loading function and $\boldsymbol{Z} \sim \mathcal{N}_q(\boldsymbol{0}, \mathbb{I}_q)$ is the latent factor. Here, $\mathcal{N}_q$ is the $q$-dimensional Gaussian distribution and $\mathbb{I}_q$ is the $q$-dimensional identity matrix. In the DLF model, we set $(\mu_\theta(\cdot), \Phi_\theta(\cdot)^\top)$ by a student DNN parameterized by $\theta$ which has $q + 1$ output nodes.

It is easy to see that the DLF is a Gaussian process with mean function $\mu_\theta(\cdot)$ and covariance function $\Sigma_\theta(\cdot, \cdot) = \Phi_\theta(\cdot)^\top \Phi_\theta(\cdot)$. Once we have $n$ many teacher models $f_1(\cdot), \ldots, f_n(\cdot)$, we assume them to be independent realizations of the DLF model and estimate the mean and factor loading functions.

**Multivariate case**   The DLF model for a multivariate function $\boldsymbol{f}(\cdot) = (f_1(\cdot), \ldots, f_c(\cdot))^\top$ is defined as

$$\boldsymbol{f}(\cdot) = \mu_\theta(\cdot) + L\boldsymbol{Z}\Phi_\theta(\cdot), \tag{2}$$

where $\mu_\theta(\cdot) : \mathcal{X} \to \mathbb{R}^c$ is the mean function, $\Phi_\theta(\cdot) : \mathcal{X} \to \mathbb{R}^q$ is the factor loading function, $L \in \mathbb{R}^{c \times c}$ is a lower-triangular matrix and $\boldsymbol{Z} \sim \mathcal{MN}_{c,q}(0, \mathbb{I}_c, \mathbb{I}_q)$. Here, $\mathcal{MN}_{c,q}(0, \mathbb{I}_c, \mathbb{I}_q)$ is a matrix-variate Gaussian distribution. It can be shown that the DLF model is a multivariate Gaussian process $\mathcal{MGP}_c(\mu_\theta, \Sigma, \Lambda)$ with the mean function $\mu_\theta(\cdot)$, covariance function $\Sigma(\cdot, \cdot) = \Phi_\theta(\cdot)^\top \Phi_\theta(\cdot)$ and parameter matrix $\Lambda = LL^\top$. For the definition of multivariate Gaussian process, see [31].

## 3.2 Estimation of the mean and factor loading

The main idea of Gaussian distillation is to estimate the mean and factor loading functions by maximizing the corresponding log-likelihood, assuming that teacher models are independent realizations

of the DLF. For optimization, we use the EM algorithm [32]. In this section, we explain the EM algorithm for Gaussian distillation. For ease of notation, we only consider the univariate DLF model, and refer to Appendix B.2.2 for the multivariate DLF.

Suppose that $n$ many teacher models $f_1(\cdot), \ldots, f_n(\cdot)$ are given. Gaussian distillation consists of three steps. The first step is to choose $m$-many design points $\mathcal{D}^{\mathrm{design}} = \{\boldsymbol{x}_1^{(d)}, \ldots, \boldsymbol{x}_m^{(d)}\}$. We will discuss how to choose the design points in Section 3.4. The second step is to calculate the vectors of prediction values of each teacher model at the design points to have $\boldsymbol{f}_i = \left(f_i(\boldsymbol{x}_j^{(d)}), j = 1, \ldots, m\right)^\top$ for $i = 1, \ldots, n$. The final step is to estimate the parameter $\theta$ in the DLF model assuming that $f_1(\cdot), \ldots, f_n(\cdot)$ are independent realizations of a random function following the DLF model. Since $\boldsymbol{f}_i$s are independent Gaussian random vectors, the MLE can be obtained by use of the EM algorithm as follows.

To make the EM algorithm numerically stable, we consider the noisy DLF model which assumes that $\boldsymbol{f}_i = \tilde{\boldsymbol{f}}_i + \boldsymbol{v}_i$, where $\boldsymbol{v}_i \sim \mathcal{N}_m(\boldsymbol{0}, \sigma_f^2 \mathbb{I}_m)$ and $\tilde{\boldsymbol{f}}_i = (\tilde{f}_i(\boldsymbol{x}_1), \ldots, \tilde{f}_i(\boldsymbol{x}_m))^\top$ with $\tilde{f}_i(\cdot)$s following the DLF model. Specifically, each $\tilde{f}_i$ is expressed as $\tilde{f}_i(\cdot) = \mu_\theta(\cdot) + \Phi_\theta(\cdot)^\top \boldsymbol{z}_i$, where $\boldsymbol{z}_i \sim \mathcal{N}_q(\boldsymbol{0}, \mathbb{I}_q)$ denotes the latent factor corresponding to the $i$-th function realization. Then, we obtain the MLE of the parameter $\theta$ in the mean and factor loading functions as well as $\sigma_f^2$. We abuse the notation to write $\theta = (\theta, \sigma_f^2)$ unless there is any confusion.

The complete log-likelihood is given as

$$\ell^{com}(\theta|\boldsymbol{f}_{1:n}, \boldsymbol{z}_{1:n}) = -\frac{nm}{2}\log(2\pi\sigma_f^2) - \frac{nq}{2}\log(2\pi) - \frac{\sum_{i=1}^n \boldsymbol{z}_i^\top \boldsymbol{z}_i}{2}$$
$$- \frac{\sum_{i=1}^n (\boldsymbol{f}_i - \boldsymbol{\mu}_\theta - \boldsymbol{\Phi}_\theta \boldsymbol{z}_i)^\top (\boldsymbol{f}_i - \boldsymbol{\mu}_\theta - \boldsymbol{\Phi}_\theta \boldsymbol{z}_i)}{2\sigma_f^2}, \tag{3}$$

where $\boldsymbol{f}_{1:n} = \{\boldsymbol{f}_1, \ldots, \boldsymbol{f}_n\}, \boldsymbol{z}_{1:n} = \{\boldsymbol{z}_1, \ldots, \boldsymbol{z}_n\}, \boldsymbol{\mu}_\theta = (\mu_\theta(\boldsymbol{x}_1^{(d)}), \ldots, \mu_\theta(\boldsymbol{x}_m^{(d)}))^\top$ and $\boldsymbol{\Phi}_\theta = (\Phi_\theta(\boldsymbol{x}_1^{(d)}), \ldots, \Phi_\theta(\boldsymbol{x}_m^{(d)}))^\top$ is an $m \times q$ matrix.

For a given parameter $\theta^{(t-1)}$ at time $t - 1$, the E-step is to calculate the conditional expectation of the complete log-likelihood $Q(\theta|\theta^{(t-1)}) = \mathbb{E}_{\boldsymbol{z}_{1:n}|\boldsymbol{f}_{1:n}, \theta^{(t-1)}}[\ell^{com}(\theta|\boldsymbol{f}_{1:n}, \boldsymbol{z}_{1:n})]$, whose formula is given in Appendix B. In the M-step, we update $\theta^{(t)}$ by a stochastic gradient descent algorithm on mini-batches. The EM algorithm is summarized in Algorithm 1 in Appendix B.

**Choice of the initial parameter** Note that the EM algorithm may converge to a local optimum, and the choice of an initial solution significantly impacts the final estimate. For the DLF model, where the factor loading involves a complex DNN structure, this issue becomes even worse. Moreover, the identifiability issue of the factor loading makes initializing the EM algorithm from a well-chosen starting point become even more crucial.

For searching a good initial solution, we pretrain the DLF model by maximizing the following penalized complete log-likelihood with respect to $\boldsymbol{\theta}$ and $\boldsymbol{z}_i$s where

$$\ell^{pen}(\boldsymbol{\theta}, \boldsymbol{z}_{1:n}|\boldsymbol{f}_{1:n}) = \ell^{com}(\boldsymbol{\theta}|\boldsymbol{f}_{1:n}, \boldsymbol{z}_{1:n}) - \lambda\mathcal{D}_{\mathrm{MMD}}(\boldsymbol{z}_{1:n}, \boldsymbol{z}'_{1:n}) \tag{4}$$

for $\lambda > 0$, where $\boldsymbol{z}'_{1:n}$ are samples generated from the standard Gaussian distribution and $\mathcal{D}_{\mathrm{MMD}}$ is the Maximum Mean Discrepancy (MMD) with the RBF kernel. The term MMD is introduced to make the distribution of the estimated $\boldsymbol{z}_{1:n}$ similar to the standard Gaussian distribution.

**KD for $\sigma_\epsilon^2$** For regression problems, we need a KD method for $\sigma_\epsilon^2$. Let $\sigma_{\epsilon,1}^2, \ldots, \sigma_{\epsilon,n}^2$ be estimators of $\sigma_\epsilon^2$ provided by each teacher model. We assume that $\sigma_{\epsilon,i}^2, i = 1, \ldots, n$ are independently generated from the inverse gamma distribution and estimate the parameter in the distribution accordingly. In the inference phase, we generate ensemble members of $\sigma_\epsilon^2$ from the estimated inverse gamma distribution.

### 3.3 Comparison with other baselines

**Comparison with Hydra** Recall that we model $(\mu_\theta(\cdot), \Phi_\theta(\cdot)^\top)^\top$ by a DNN with $q + 1$ many heads. Note that the DLF model assumes $f_i(\cdot) = \mu_\theta(\cdot) + \Phi_\theta(\cdot)^\top \boldsymbol{z}_i$ for $i = 1, \ldots, n$, and thus we

can interpret $(\mu_\theta(\cdot), \Phi_\theta(\cdot)^\top)^\top$ as the body and $\boldsymbol{z}_i$s as the model-specific weights at the head. In view of sharing the body, the DLF model is quite similar to the conventional multi-head structure used in Hydra [11]. The main difference is that the DLF model treats $\boldsymbol{z}_i$s as random quantities and thus integrates out before estimating the MLE while Hydra treats $\boldsymbol{z}_i$s as fixed effects and estimates $\theta$ and $\boldsymbol{z}_i$s simultaneously by minimizing a given loss. It is well-known that treating random effects as fixed effects is highly susceptible to bias [33, 34, 35, 36]. Our experimental results in Section 5 amply demonstrate that treating $\boldsymbol{z}_i$s as random is better than treating $\boldsymbol{z}_i$s as fixed effects.

**Comparison with Dirichlet distillation**   At least, there are two advantages of Gaussian distillation compared to Dirichlet distillation. Gaussian distillation can be applied to both regression and classification models while Dirichlet distillation is only applicable to classification problems. The second advantage is that estimation of the mean and factor loading in the DLF model is easier than estimation of the parameter in the Dirichlet distribution owing to the nice EM algorithm. This stability makes Gaussian distillation perform better than Dirichlet distillation. The inferior performance of Dirichlet distillation compared to one-to-one distillation, as reported in [11, 13], is confirmed by our experiments in Section 5.

### 3.4   Choice of design points

Let $\hat{\mu}(\cdot)$ and $\hat{\Sigma}(\cdot, \cdot)$ be the estimate of $\mu_*(\cdot)$ and $\Sigma_*(\cdot, \cdot)$ by the DLF model with design points $\mathcal{D}^{\mathrm{design}}$. For a given $\boldsymbol{x} \in \mathcal{X}$, let $\hat{p}_{\boldsymbol{x}}$ and $p_{*,\boldsymbol{x}}$ be the distributions of $f(\boldsymbol{x})$ under the assumption that $f(\cdot)$ is a Gaussian process with the parameters $(\hat{\mu}, \hat{\Sigma})$ and $(\mu_*, \Sigma_*)$, respectively. In Theorem G.3 in Appendix G.2, we prove that $\sup_{\boldsymbol{x} \in \mathcal{D}^{\mathrm{design}}} d_1(\hat{p}_{\boldsymbol{x}}, p_{*,\boldsymbol{x}})$ converges to 0 as $n \to \infty$ if we choose the architecture of a student DNN for $(\mu_\theta(\cdot), \Phi_\theta(\cdot)^\top)$ appropriately, where $d_1$ is the $\ell_1$ metric.

For $\boldsymbol{x} \notin \mathcal{D}^{\mathrm{design}}$, if $\hat{\mu}$ and $\mu_*$, as well as $\hat{\Sigma}$ and $\Sigma_*$ are (coordinate-wise) Lipschitz, it can be shown that $d_1(\hat{p}_{\boldsymbol{x}}, p_{*,\boldsymbol{x}}) \leq d_1(\hat{p}_{\boldsymbol{x}_{(1)}}, p_{*,\boldsymbol{x}_{(1)}}) + C\|\boldsymbol{x} - \boldsymbol{x}_{(1)}\|$ for a positive constant $C$, where $\boldsymbol{x}_{(1)}$ is the nearest point in $\mathcal{D}^{\mathrm{design}}$ to $\boldsymbol{x}$. See Theorem G.3 in Appendix G.2. Note that the term $\|\boldsymbol{x} - \boldsymbol{x}_{(1)}\|$ is affected by the choice of design points. Suppose that $\boldsymbol{x}$ is a realization of a random vector $\boldsymbol{X} \sim \mathbb{P}$. Then, the expected nearest-neighbor distance $\mathbb{E}_{\boldsymbol{X} \sim \mathbb{P}}\|\boldsymbol{X} - \boldsymbol{X}_{(1)}\|$ becomes smaller when the design points are located in a higher-density region of $\mathbb{P}$. This observation suggests that design points similar to test data would be better. Validation data (dataset whose distribution is the same as the training data used for learning a teacher ensemble) would be a promising candidate for the design points. See Appendix D.1.1 for numerical experiments.

## 4   Application to distribution shift problems

The pre-trained DLF can be applied to distribution shift problems. We say that given new data is shifted in distribution if the distribution of new data is different from that of training data. Distribution shift problems, whose aim is to efficiently learn a prediction model on new data when the size of new data is small, have been studied extensively [37, 38, 39, 40, 41, 42]. A popular method is to learn a DNN on training data first and retrain the head of the DNN on new data while the body is fixed [43, 44].

Note that the DLF model is given as

$$f_j(\cdot) = \hat{\mu}_j(\cdot) + \sum_{k=1}^{q} \sum_{l=1}^{c} \hat{\Phi}_k(\cdot)\hat{L}_{jl} z_{jlk}, \tag{5}$$

for $j = 1, \ldots, c$, where $z_{jkl}$s are independent standard Gaussian random variables. For distribution shift problems, we can treat $(\hat{\mu}(\cdot), \hat{\Phi}(\cdot)^\top, \hat{L})$ as a learned body and $z_j$s are the weights in the prediction head. Then we learn only the weights of the head on new data while fixing the body. In Section 5.3, we show empirically that this method outperforms its competitors. The superior performance of the DLF model for distribution shift problems indicates that Gaussian distillation is good at not only uncertainty quantification but estimating the feature vector $(\hat{\mu}(\cdot), \hat{\Phi}(\cdot)^\top)$.

# 5 Experiments

In this section, we investigate Gaussian distillation by analyzing multiple benchmark datasets. We compare Gaussian distillation with existing baselines including the naive distillation (one-to-one distillation without sharing weights between student DNNs, small-Ens), Hydra [11] and BE [12] for regression and classification problems as well as fine-tuning of language models in view of uncertainty quantification. For classification, we also evaluate Proxy–Dirichlet Distillation (Proxy-End$^2$) [18] and Ensemble Distillation via Flow Matching (EDFM) [45]. In addition, we show that a pre-trained DLF outperforms its competitors for distribution shift problems.

## 5.1 Uncertainty quantification for regression and classification problems

### 5.1.1 Regression case

**Datasets** We analyze six benchmark datasets from the UCI repository [46] including Boston housing, Concrete, Energy, Wine, Power Plant, and Kin8nm. Each dataset is randomly split into 90% training and 10% testing, and teacher models are trained following the experimental protocol of [47]. We repeat this procedure 10 times to obtain 10 measures of the evaluation metrics of each methods and report the averages (with the standard errors). See Appendix C.1 for details of implementation.

**Results** Table 1 presents the results of the four evaluation metrics (see Appendix C.1.2 for the definitions) for performance and uncertainty quantification. DLF outperforms Hydra and BE in most cases. Even when it is not the best, DLF is at least the second best. The coverage probabilities of Hydra and BE are sometimes much lower than those of DLF (Boston housing and Concrete for Hydra, and Boston housing and kim8nm for BE), which suggests that deterministic distillation methods fail to fully preserve the uncertainty in a teacher ensemble. This observation is not surprising since variation of smaller models is in general smaller than that of larger models and weight sharing would reduce the variation further. Thus, a way of adding additional uncertainty to a student ensemble is needed, and distribution distillation is such a solution.

Table 1: Results on UCI benchmark datasets. ($*$ : closer to the coverage probability of a teacher ensemble is better)

| Metric | Method | Datasets | | | | | |
| | | Boston housing | Concrete | Energy | Wine | Power | Kin8nm |
|---|---|---|---|---|---|---|---|
| RMSE ↓ | Teachers | 2.5786 | 5.6191 | 0.5692 | 0.5497 | 4.2197 | 0.0794 |
| | small-Ens | 2.7280 (0.0184) | 5.6952 (0.0494) | 0.6367 (0.0182) | 0.6002 (0.0083) | 4.2430 (0.0864) | 0.0865 (0.0007) |
| | Hydra | 2.8346 (0.0835) | 6.0558 (0.1366) | 0.6549 (0.0360) | 0.5689 (0.0114) | 4.2284 (0.0074) | 0.0932 (0.0023) |
| | BE | 2.8375 (0.0729) | 5.9777 (0.1475) | 0.6661 (0.0354) | 0.556 (0.0187) | 4.2367 (0.0041) | 0.0962 (0.0059) |
| | DLF | **2.6687 (0.1700)** | **5.6047 (0.1771)** | **0.5659 (0.0239)** | **0.5506 (0.0104)** | **4.2211 (0.0010)** | **0.0825 (0.0010)** |
| NLL ↓ | Teachers | 2.3850 | 3.1134 | 0.8533 | 0.7980 | 2.8586 | -1.1109 |
| | small-Ens | **2.4150 (0.0085)** | 3.1672 (0.0167) | 0.9829 (0.0263) | 0.8861 (0.0160) | 2.8622 (0.0191) | -1.032 (0.0108) |
| | Hydra | 2.4843 (0.0478) | 3.2586 (0.0293) | 0.9914 (0.0660) | 0.8322 (0.0166) | 2.8604 (0.0017) | -0.9434 (0.0307) |
| | BE | 2.4892 (0.0408) | 3.2241 (0.0332) | 0.9879 (0.0511) | **0.8065 (0.0138)** | 2.8628 (0.0010) | -0.9115 (0.0808) |
| | DLF | 2.4346 (0.1147) | **3.1584 (0.0463)** | **0.8525 (0.0471)** | 0.8230 (0.0199) | **2.8591 (0.0010)** | **-1.0754 (0.0126)** |
| CRPS ↓ | Teachers | 1.4425 | 2.9926 | 0.3137 | 0.2962 | 2.3360 | 0.0443 |
| | small-Ens | 1.5233 (0.0087) | **2.9953 (0.0304)** | 0.3461 (0.0077) | 0.3307 (0.0067) | 2.3392 (0.0321) | 0.0475 (0.0005) |
| | Hydra | 1.6041 (0.0494) | 3.3084 (0.0639) | 0.3622 (0.0216) | 0.3075 (0.0040) | 2.3405 (0.0048) | 0.0518 (0.0012) |
| | BE | 1.6158 (0.0490) | 3.2320 (0.0894) | 0.3617 (0.0168) | 0.3020 (0.0059) | 2.3483 (0.0024) | 0.0532 (0.0035) |
| | DLF | **1.4317 (0.1029)** | 3.0622 (0.0954) | **0.3163 (0.0119)** | **0.2980 (0.0043)** | **2.3364 (0.0010)** | **0.0458 (0.0005)** |
| 95% Coverage Probability $*$ | Teachers | 0.9608 | 0.9515 | 1.0000 | 0.9750 | 0.9697 | 0.9610 |
| | small-Ens | **0.9408 (0.0016)** | **0.9431 (0.0041)** | 0.9921 (0.0011) | 0.9569 (0.0042) | 0.9669 (0.0012) | **0.9649 (0.0016)** |
| | Hydra | 0.8995 (0.0109) | 0.9097 (0.0115) | 0.9948 (0.0091) | 0.9594 (0.0053) | 0.9782 (0.0003) | 0.9407 (0.0046) |
| | BE | 0.9093 (0.0090) | 0.9282 (0.0099) | 0.9922 (0.0110) | 0.9612 (0.0092) | 0.9778 (0.0014) | 0.9080 (0.0213) |
| | DLF | 0.9240 (0.0154) | 0.9291 (0.0125) | **1.0000 (0.0000)** | **0.9681 (0.0055)** | **0.9711 (0.0053)** | 0.9761 (0.0036) |

### 5.1.2 Classification case

**Datasets** CIFAR-10 and CIFAR-100 consist of 50,000 training and 10,000 test images. In this experiment, the training data are further split into 80% training and 20% validation, and teacher models are trained on the training data and the number of epochs is determined by the validation

data. Implementation details of the distillation methods are given in Appendix C.2. Experiments are repeated with 5 different random initializations for each method.

**Results**  As shown in Table 2, DLF outperforms the other baselines consistently in terms of not only uncertainty quantification but also accuracy. In particular, improvements of DLF with respect to ECE are noticeable. The definitions of the evaluation metrics are given in Appendix C.2.2.

Table 2: Results on CIFAR-10 and CIFAR-100.

| dataset | method | Acc(%) $\uparrow$ | NLL $\downarrow$ | ECE(%) $\downarrow$ |
|---|---|---|---|---|
| CIFAR-10 | Teachers | 94.24 | 0.1539 | 0.9 |
| | small-Ens | 92.87 (0.35) | 0.2377 (0.0019) | 3.93 (0.14) |
| | Hydra | 93.16 (0.06) | 0.2660 (0.0063) | 4.15 (0.01) |
| | LBE | 93.25 (0.26) | 0.2480 (0.0053) | 4.11 (0.10) |
| | Proxy-EnD$^2$ | 90.92 (0.28) | 0.2861 (0.0027) | 2.08 (0.19) |
| | EDFM | 90.62 (0.23) | 0.2858 (0.0025) | 2.78 (0.17) |
| | DLF | **93.40 (0.14)** | **0.2246 (0.0023)** | **2.79 (0.20)** |
| CIFAR-100 | Teachers | 81.36 | 0.7167 | 1.41 |
| | small-Ens | 79.29 (0.26) | 1.0413 (0.0145) | 12.90 (0.24) |
| | Hydra | 77.42 (0.15) | 1.2912 (0.0272) | 12.70 (0.37) |
| | LBE | 79.58 (0.40) | 1.0110 (0.0087) | 13.42 (0.42) |
| | Proxy-EnD$^2$ | 67.62 (0.22) | 1.2355 (0.0151) | **7.35 (0.25)** |
| | EDFM | 64.17 (0.32) | 1.6741 (0.0242) | 11.35 0.47) |
| | DLF | **79.68 (0.23)** | **0.8974 (0.0042)** | 9.45 (0.31) |

## 5.2  Application to fine-tuning of language models

In this section, we apply the proposed distillation framework to downstream binary classification tasks using pretrained language models. Given pre-trained teacher and student language models, fine-tuned teacher and student models are obtained using Low-Rank Adaptation (LoRA) [48]. For the teacher and student pre-trained language models, "*RoBERTa*" [49] and "*DistilRoBERTa*" [1] [3] are used. As a teacher ensemble, we obtain four fine-tuned models by combining LoRA and *RoBERTa* with randomly selected initializations for each task. Then, an ensemble of fine-tuned student language models is constructed by applying LoRA to *DistilRoBERTa* for each distillation method.

**Datasets**  We analyze three GLUE [50] and SuperGLUE [51] sub-tasks: RTE, MRPC, and WiC. All three datasets are binary classification tasks. Implementation details of the distillation methods are given in Appendix C.3.

**Results**  As shown in Table 3, Gaussian distillation outperforms Hydra and LBE with large margins. We conjecture that the performance gap of Gaussian distillation to Hydra and LBE would become larger when the complexity gap between teacher and student models becomes larger. This is a reasonable conjecture since smaller models could preserve less variations in teacher models. Gaussian distillation would add additional variations to the student models through variations of the latent vector $Z$.

## 5.3  Application to distribution shift problems

We compare DLF with two baselines which fine-tune only the head on new data while the body is learned by either (1) a standard DNN or (2) applying Hydra on training data of CIFAR-10. For distribution-shifted new data, we swap the labels of CIFAR-10 as is done by [44]. See implementation details in Appendix C.4. The results are given in Figure 2 which amply show that DLF is superior. It is interesting to see that DLF outperforms even when the sample size of the new data is large, which implies that the learned body by DLF is qualitatively different from those by DNN and Hydra. We do not know the reason but an implication is that Gaussian distillation is good at learning not only quantifying uncertainty but also learning the feature vector (i.e. the body).

---

[3] `https://www.huggingface.co/distilroberta-base`

Table 3: Results on GLUE and SuperGLUE benchmark datasets

| dataset | method | Acc (%) ↑ | NLL ↓ | ECE (%) ↓ |
|---|---|---|---|---|
| RTE | Teachers | 75.09 | 0.8401 | 17.70 |
| | small-Ens | 67.15 (0.0057) | 0.6739 (0.0124) | 13.09 (0.0148) |
| | Hydra | 62.82 (0.1650) | 0.9034 (0.1733) | 23.31 (0.4907) |
| | LBE | 65.97 (0.1406) | 0.9235 (0.0664) | 25.63 (0.1909) |
| | DLF | **67.06 (0.1040)** | **0.6658 (0.0762)** | **9.74 (0.5050)** |
| MRPC | Teachers | 87.25 | 0.3435 | 4.77 |
| | small-Ens | 83.092 (0.0172) | 0.4596 (0.0396) | **9.84 (0.0164)** |
| | Hydra | 82.19 (0.0357) | 0.6429 (0.1274) | 11.79 (0.0136) |
| | LBE | 82.23 (0.0153) | 0.6527 (0.0544) | 13.6 (0.0114) |
| | DLF | **83.094 (0.0035)** | **0.4526 (0.0113)** | 10.72 (0.0392) |
| WiC | Teachers | 68.03 | 0.6395 | 9.14 |
| | small-Ens | 65.02 (0.0062) | **0.7674 (0.0206)** | 16.69 (0.0116) |
| | Hydra | 65.05 (0.0210) | 1.1628 (0.0355) | 26.26 (0.0191) |
| | LBE | 65.36 (0.1209) | 0.8809 (0.0950) | 20.28 (0.0300) |
| | DLF | **66.18 (0.0146)** | 0.7706 (0.0543) | **15.99 (0.0198)** |

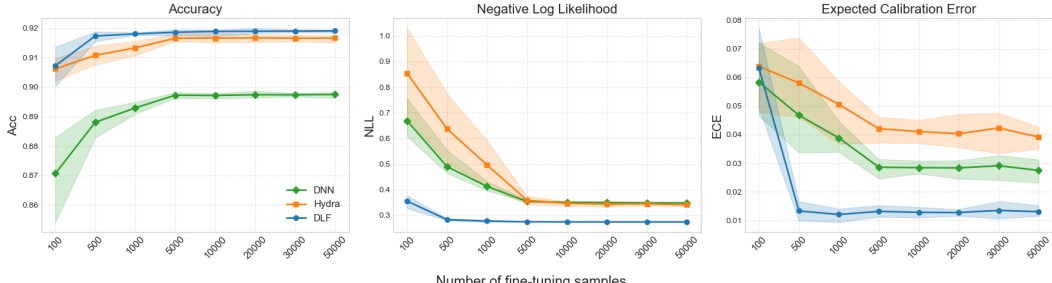

Figure 2: Comparison of performances of the three learning algorithms (DNN, Hydra and DLF) pretrained on CIFAR-10 and fine-tuned on CIFAR10-Flip as the sample size of CIFAR 10-Flip data varies. The solid curves are the means and the shaded bands are the min-max spreads obtained from 5 training models on 5 randomly selected new data of CIFAR10-Flip.

## 5.4 Ablation studies

In Appendix D, we present the results of ablation studies including the sensitivity of Gaussian distillation to the choice of design points, the dimension of latent factor, the architecture size of student DNNs and the number of ensemble members used in a teacher and student ensemble. In addition, we compare the results when the initial solution is randomly selected in Gaussian distillation.

## 6 Conclusion

We proposed a novel method for distilling deep ensembles, specifically addressing the challenges associated with computational costs, inference time, and storage capacities inherent in traditional deep ensemble approaches. The key innovation lies in modeling the covariance structure of deep ensembles through the DLF model, enabling efficient preservation of uncertainty in a teacher ensemble with significantly reduced inference costs.

There are several future research topics. First, in this paper, we only focused on deep ensembles. It would be valuable to consider Bayesian DNNs, as they provide a framework for uncertainty quantification [25, 26, 27] and can potentially serve as a prior for on-device posterior updates. Second, for distillation of a fine-tuned language model, we used *DistilRoBERTa* [1], a pretrained distilled language model. It would be promising to distill the pretrained language model and the model for LoRA simultaneously. Third, the DLF could be used for online Bayesian learning by approximating the posterior with respect to old data by the DLF and using it for the prior of new data. We will pursue this idea in a near future.

## Acknowledgements

This work was partly supported by the National Research Foundation of Korea(NRF) grant funded by the Korea government(MSIT) (No. 2022R1A5A7083908), the National Research Foundation of Korea(NRF) grant funded by the Korea government(MSIT) (RS-2025-00556079), and by Institute of Information & communications Technology Planning & Evaluation (IITP) grant funded by the Korea government(MSIT) [NO.RS-2021-II211343, Artificial Intelligence Graduate School Program (Seoul National University)].

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

# APPENDIX

## A  Review of Ensemble Distillation

### A.1  One-to-one Distillation

For given teacher members $p_i^{(t)}, i = 1, \ldots, n$, student $p_i^{(s)}, i = 1, \ldots, n$ are trained to minimize

$$\sum_{i=1}^n \mathbb{E}_{\boldsymbol{x}} \left[ \mathrm{KL} \left( p_i^{(t)}(y|\boldsymbol{x}) || p_i^{(s)}(y|\boldsymbol{x}) \right) \right]$$

where $\mathbb{E}_{\boldsymbol{x}}$ is the expectation operator of a certain probability distribution on the input space $\mathcal{X}$. The following two methods use specially designed DNNs for student models which share the weights between student models.

#### A.1.1  Hydra

Hydra [11] employs a multi-head architecture in which a single shared body network extracts common features and a set of $n$ distinct linear heads generates predictions for each ensemble member. Formally, the student DNN is parameterized by

$$\theta_{\mathrm{hydra}} = \{ \theta_{\mathrm{body}}, \theta_{\mathrm{head},1}, \ldots, \theta_{\mathrm{head},n} \},$$

where $\theta_{\mathrm{body}}$ is used across all heads. At inference, the body is evaluated once, and each head $h_{\theta_{\mathrm{head},i}}(\cdot)$ produces a member-specific output $p_i^{(s)}(y \mid \boldsymbol{x}) = h_{\theta_{\mathrm{head},i}} \left( b_{\theta_{\mathrm{body}}}(\boldsymbol{x}) \right)$. This design captures ensemble diversity while reducing both computation and memory compared to maintaining $n$ independent networks.

#### A.1.2  Batch Ensemble and Latent Batch Ensemble

[12] introduces Batch Ensemble (BE) to reduce the memory and computational burden of deep ensembles. In this architecture, all student networks share a core weight matrix $\boldsymbol{W}^l$ at each layer, while individual ensemble members are differentiated by rank-one perturbations. Specifically, for the $i$-th student at layer $l$,

$$\boldsymbol{W}_i^l = \boldsymbol{W}^l \circ \left( \boldsymbol{r}_i^l \boldsymbol{s}_i^{l\top} \right),$$

where $\boldsymbol{r}_i^l \in \mathbb{R}^{d_{\mathrm{out}}}$ and $\boldsymbol{s}_i^l \in \mathbb{R}^{d_{\mathrm{in}}}$ modulate the rows and columns of $\boldsymbol{W}^l$, respectively. This construction enables each sub-network to maintain member-specific behaviors while reusing the majority of parameters, yielding significant savings in both storage and inference costs compared to training $n$ independent models.

Building on BE, [14] proposes Latent Batch Ensemble (LBE), which further compresses the ensemble at inference time. Instead of maintaining $n$ distinct perturbations, LBE computes the average rank-one mask across all students:

$$\boldsymbol{W}_s^l = \boldsymbol{W}^l \circ \left( \frac{1}{n} \sum_{i=1}^n \boldsymbol{r}_i^l \boldsymbol{s}_i^{l\top} \right).$$

The resulting single student network requires only one forward pass per input while capturing the ensemble's mean perturbation in weight space. Empirical results demonstrate that LBE matches or exceeds the calibration performance of standard BE, with inference cost reduced by a factor of $n$. However, the proposed Latent Batch Ensemble is a specialized method designed for ensemble distillation for classification problems.

### A.2  Distribution Distillation

Distribution distillation frames the output of an ensemble at the input $\boldsymbol{x}$ as samples from an input-dependent Dirichlet distribution [17, 18]. Let $\boldsymbol{f}_i, i = 1, \ldots, n$ be given teacher models for a classification task. For a given $\boldsymbol{x}$, let $\boldsymbol{\pi}_i(\boldsymbol{x}) = \mathrm{softmax} \left( \boldsymbol{f}_i(\boldsymbol{x}) \right)$, and we assume that $\boldsymbol{\pi}_1(\boldsymbol{x}), \ldots, \boldsymbol{\pi}_n(\boldsymbol{x})$

independently follow the Dirichlet distribution with parameter $\boldsymbol{\alpha}(\boldsymbol{x}) = (\alpha_1(\boldsymbol{x}), \ldots, \alpha_c(\boldsymbol{x}))$. Then, we model $\boldsymbol{\alpha}(\boldsymbol{x})$ by a student DNN $\boldsymbol{\Psi}(\boldsymbol{x}|\theta)$ and estimate $\theta$ by maximizing

$$\mathcal{L}(\theta, \mathcal{D}^{\text{design}}) = \sum_{j=1}^{m} \sum_{i=1}^{n} \left[ \ln p(\boldsymbol{\pi}_i(\boldsymbol{x}_j^{(d)}) | \boldsymbol{\Psi}(\boldsymbol{x}_j^{(d)}|\theta)) \right],$$

where $p(\boldsymbol{\pi}|\boldsymbol{\alpha})$ is the density of the Dirichlet distribution with parameter $\boldsymbol{\alpha}$. See [17, 18] for details.

### A.2.1 Proxy-Dirichlet Distillation

[18] proposed Proxy–Dirichlet Distillation (Proxy-EnD$^2$) to mitigate the convergence difficulties of standard Dirichlet distillation [17] when the number of classes is very high. The method constructs a proxy Dirichlet distribution from the teacher models and trains the student DNN $\boldsymbol{\Psi}(\boldsymbol{x}|\theta)$ by minimizing the reverse KL divergence between $p(\boldsymbol{\pi}|\boldsymbol{\Psi}(\boldsymbol{x}|\theta))$ and the proxy distribution.

The proxy distribution is obtained from teacher models as

$$\hat{\beta}_l(\boldsymbol{x}) = \frac{1}{n} \sum_{i=1}^{n} \pi_{il}(\boldsymbol{x}),$$

$$\hat{\alpha}_l(\boldsymbol{x}) = \frac{\hat{\beta}_l(\boldsymbol{x})\,(c-1)}{2 \sum_{l=1}^{c} \hat{\beta}_l(\boldsymbol{x})\Big(\ln \hat{\beta}_l(\boldsymbol{x}) - \frac{1}{n}\sum_{i=1}^{n} \ln \pi_{il}(\boldsymbol{x})\Big)}, \quad l = 1, 2, \ldots \cdots, c.$$

Then, estimate $\theta$ by minimizing

$$\sum_{j=1}^{m} \text{KL}\Big( p\Big(\boldsymbol{\pi} \mid \exp\big(\boldsymbol{\Psi}(\boldsymbol{x}_j^{(d)} \mid \theta)\big)\Big) \,\Big\|\, p\Big(\boldsymbol{\pi} \mid \hat{\boldsymbol{\alpha}}(\boldsymbol{x}_j^{(d)}) + 1\Big)\Big),$$

where $\hat{\boldsymbol{\alpha}}(\cdot) = [\hat{\alpha}_1(\cdot), \ldots, \hat{\alpha}_c(\cdot)]$.

### A.2.2 Ensemble Distribution Distillation via Flow Matching

Ensemble Distribution Distillation via Flow Matching (EDFM) [45] models a conditional distribution of logits $\boldsymbol{h}_1$ given $\boldsymbol{x}$ over $\mathbb{R}^c$. The method constructs the target distribution $p_1(\boldsymbol{h}|\boldsymbol{x})$ as an empirical distribution using $\boldsymbol{f}_i(\boldsymbol{x})$.

They model using Flow matching such that

$$d\boldsymbol{h}_t(\boldsymbol{x}) = \boldsymbol{\Psi}_\theta(t, \boldsymbol{h}_t(\boldsymbol{x}), \boldsymbol{x})dt,\ t \in [0,1],\ \boldsymbol{h}_0(\boldsymbol{x}) \sim \mathcal{N}_c(0, \sigma^2 \mathbb{I}_c) \text{ and } \boldsymbol{h}|\boldsymbol{x} \overset{d}{=} \boldsymbol{h}_1(\boldsymbol{x})$$

where $\boldsymbol{\Psi}_\theta$ is student DNN.
Then, the network is trained by minimizing the conditional flow matching loss

$$\sum_{j=1}^{m} \mathbb{E}_{\boldsymbol{h}_0, \boldsymbol{h}_1(\boldsymbol{x}_j^{(d)}), t} \left[ \lambda(t) \| \boldsymbol{\Psi}_\theta(t, \boldsymbol{h}_t(\boldsymbol{x}_j^{(d)}), \boldsymbol{x}_j^{(d)}) - (\boldsymbol{h}_1(\boldsymbol{x}_j^{(d)}) - \boldsymbol{h}_0) \|^2 \right]$$

where $\boldsymbol{h}_0 \sim \mathcal{N}_c(0, \sigma^2 \mathbb{I}_c), \boldsymbol{h}_1(\boldsymbol{x}_j^{(d)}) \sim p_1(\cdot|\boldsymbol{x}_j^{(d)}), t \sim U[0,1], \boldsymbol{h}_t(\boldsymbol{x}_j^{(d)}) = t\boldsymbol{h}_1(\boldsymbol{x}_j^{(d)}) + (1-t)\boldsymbol{h}_0$
and $\lambda$ denotes a time dependent weighting function.
Sampling proceeds by numerically integrating the learned flow $\boldsymbol{\Psi}_\theta(t, \boldsymbol{h}_t(\boldsymbol{x}), \boldsymbol{x})$ with a standard ordinary differential equation (ODE) solver. For more detailed information, see [45].

## B   Details of Gaussian distillation

### B.1   EM algorithm for the univariate DLF model

The main idea of Gaussian distillation is to estimate the mean and factor loading functions based on given teacher models assuming that teacher models are independent realizations of the DLF and estimate the parameter in the student DNNs by maximizing the corresponding log-likelihood. For optimization, we use the EM algorithm [32].

Suppose that $n$ many teacher models $f_1(\cdot), \ldots, f_n(\cdot)$ are given. Gaussian distillation consists of three steps. The first step is to choose $m$-many design points $\mathcal{D}^{\mathrm{design}} = \{\boldsymbol{x}_1^{(d)}, \ldots, \boldsymbol{x}_m^{(d)}\}$. The second step is to calculate the vectors of prediction values of each teacher model at the design points to have $\boldsymbol{f}_i = \left(f_i(\boldsymbol{x}_j^{(d)}), j = 1, \ldots, m\right)^\top$ for $i = 1, \ldots, n$. The final step is to estimate the parameter $\theta$ in the DLF by the MLE assuming that $f_1(\cdot), \ldots, f_n(\cdot)$ are independent realizations of a random function following the DLF model. Since $\boldsymbol{f}_i$s are independent Gaussian random vectors, the MLE can be obtained by use of the EM algorithm as follows.

To make the EM algorithm numerically stable, we consider the noisy DLF model which assumes that $\boldsymbol{f}_i = \tilde{\boldsymbol{f}}_i + \boldsymbol{v}_i$, where $\boldsymbol{v}_i \sim \mathcal{N}_m(\boldsymbol{0}, \sigma_f^2 \mathbb{I}_m)$ and $\tilde{\boldsymbol{f}}_i = (\tilde{f}_i(\boldsymbol{x}_1), \ldots, \tilde{f}_i(\boldsymbol{x}_m))^\top$ with $\tilde{f}_i(\cdot)$s following the DLF model. Specifically, each $\tilde{f}_i$ is expressed as $\tilde{f}_i(\cdot) = \mu_\theta(\cdot) + \Phi_\theta(\cdot)^\top \boldsymbol{z}_i$, where $\boldsymbol{z}_i \sim \mathcal{N}_q(\boldsymbol{0}, \mathbb{I}_q)$ denotes the latent factor corresponding to the $i$-th function realization. Then, we obtain the MLE of the parameter $\theta$ in the mean and factor loading functions as well as $\sigma_f^2$. We abuse the notation to write $\theta = (\theta, \sigma_f^2)$ unless there is any confusion.

The complete log-likelihood is given as

$$\ell^{com}(\theta | \boldsymbol{f}_{1:n}, \boldsymbol{z}_{1:n}) = -\frac{nm}{2} \log(2\pi\sigma_f^2) - \frac{nq}{2} \log(2\pi) - \frac{\sum_{i=1}^n \boldsymbol{z}_i^\top \boldsymbol{z}_i}{2}$$
$$- \frac{\sum_{i=1}^n (\boldsymbol{f}_i - \boldsymbol{\mu}_\theta - \boldsymbol{\Phi}_\theta \boldsymbol{z}_i)^\top (\boldsymbol{f}_i - \boldsymbol{\mu}_\theta - \boldsymbol{\Phi}_\theta \boldsymbol{z}_i)}{2\sigma_f^2},$$

where $\boldsymbol{f}_{1:n} = \{\boldsymbol{f}_1, \ldots, \boldsymbol{f}_n\}, \boldsymbol{z}_{1:n} = \{\boldsymbol{z}_1, \ldots, \boldsymbol{z}_n\}, \boldsymbol{\mu}_\theta = (\mu_\theta(\boldsymbol{x}_1^{(d)}), \ldots, \mu_\theta(\boldsymbol{x}_m^{(d)}))^\top$ and $\boldsymbol{\Phi}_\theta = (\Phi_\theta(\boldsymbol{x}_1^{(d)}), \ldots, \Phi_\theta(\boldsymbol{x}_m^{(d)}))^\top$ is an $m \times q$ matrix.

Thus, for a given parameter $\theta^{(t-1)}$ at time $t-1$, the E-step is to calculate the conditional expectation of the complete log-likelihood which is given as

$$Q(\theta | \theta^{(t-1)}) = \mathbb{E}_{\boldsymbol{z}_{1:n} | \boldsymbol{f}_{1:n}, \theta^{(t-1)}}[\ell^{com}(\theta | \boldsymbol{f}_{1:n}, \boldsymbol{z}_{1:n})]$$
$$= -\frac{nm}{2} \log(2\pi\sigma_f^2) - \frac{nq}{2} \log(2\pi) - \frac{\sum_{i=1}^n \operatorname{tr} \mathbb{E}[\boldsymbol{z}_i \boldsymbol{z}_i^\top | \theta^{(t-1)}, \boldsymbol{f}_{1:n}]}{2}$$
$$- \frac{1}{2\sigma_f^2} \sum_{i=1}^n \Big\{ (\boldsymbol{f}_i - \boldsymbol{\mu}_\theta)^\top (\boldsymbol{f}_i - \boldsymbol{\mu}_\theta)$$
$$+ 2(\boldsymbol{f}_i - \boldsymbol{\mu}_\theta)^\top \boldsymbol{\Phi}_\theta \mathbb{E}[\boldsymbol{z}_i | \theta^{(t-1)}, \boldsymbol{f}_{1:n}] + \operatorname{tr}\Big(\boldsymbol{\Phi}_\theta^\top \boldsymbol{\Phi}_\theta \mathbb{E}[\boldsymbol{z}_i \boldsymbol{z}_i^\top | \theta^{(t-1)}, \boldsymbol{f}_{1:n}]\Big) \Big\},$$

where

$$\mathbb{E}[\boldsymbol{z}_i | \theta^{(t-1)}, \boldsymbol{f}_{1:n}] = \mathbb{V}[\boldsymbol{z} | \theta^{(t-1)}] \boldsymbol{\Phi}_{\theta^{(t-1)}}^\top (\boldsymbol{f}_i - \boldsymbol{\mu}_{\theta^{(t-1)}}) / \sigma_f^{2^{(t-1)}}$$
$$\mathbb{E}[\boldsymbol{z}_i \boldsymbol{z}_i^\top | \theta^{(t-1)}, \boldsymbol{f}_{1:n}] = \mathbb{V}[\boldsymbol{z} | \theta^{(t-1)}] + \mathbb{E}[\boldsymbol{z}_i | \theta^{(t-1)}, \boldsymbol{f}_{1:n}] \mathbb{E}[\boldsymbol{z}_i | \theta^{(t-1)}, \boldsymbol{f}_{1:n}]^\top$$

with $\mathbb{V}[\boldsymbol{z} | \theta^{(t-1)}] = \left(\mathbb{I}_q + \boldsymbol{\Phi}_{\theta^{(t-1)}}^\top \boldsymbol{\Phi}_{\theta^{(t-1)}} / \sigma_f^{2^{(t-1)}}\right)^{-1}$.

In the M-step, we usually update $\theta^{(t)}$ by maximizing $Q(\theta | \theta^{(t-1)})$. Instead of maximizing $Q(\theta | \theta^{(t-1)})$, we update $\theta$ by using a stochastic gradient descent algorithm (i.e., a gradient descent algorithm on a given mini-batches). The EM algorithm is summarized in Algorithm 1.

**Algorithm 1:** EM algorithm for the univariate DLF model

---

**Input:** Design points $\mathcal{D}^{\text{design}} = \{\boldsymbol{x}_1^{(d)}, \ldots, \boldsymbol{x}_m^{(d)}\}$,
Teacher ensemble members $f_1(\cdot), \ldots, f_n(\cdot)$, learning rate $\eta > 0$
**Output:** Estimated parameter $\theta$
Initialize parameter: $\theta^{(0)}$
**for** $t = 1, 2 \ldots T$ **do**

    Shuffle the dataset $\mathcal{D}^{\text{design}}$ and divide into mini-batches $\{\mathcal{B}_1, \ldots, \mathcal{B}_M\}$
    **for** $l = 1, 2, \ldots, M$ **do**

        Calculate prediction values of teacher model $\boldsymbol{f}_{1:n,\mathcal{B}_l}$, where
            $\boldsymbol{f}_{1:n,\mathcal{B}_l} = (\boldsymbol{f}_{1:n}(\boldsymbol{x}_j^{\text{design}}), \boldsymbol{x}_j^{\text{design}} \in \mathcal{B}_l)$.
        Calculate $\mathbb{E}[\boldsymbol{z}|\theta^{(t-1)}]$ and $\mathbb{E}[\boldsymbol{z}\boldsymbol{z}^\top|\theta^{(t-1)}]$ using $\boldsymbol{f}_{1:n,\mathcal{B}_l}$
        $Q(\theta|\theta^{(t-1)}) := \mathrm{E}_{\boldsymbol{z}_{1:n}|\boldsymbol{f}_{1:n,\mathcal{B}_l},\theta^{(t-1)}}[\ell^{com}(\theta; \boldsymbol{f}_{1:n,\mathcal{B}_l}, \boldsymbol{z}_{1:n})]$
        Calculate gradient $g^{(t)} := \nabla_\theta \left(-Q(\theta|\theta^{(t-1)})\right)$
        Update $\theta^{(t)} \leftarrow \theta^{(t-1)} - \eta \cdot g^{(t)}$
    **end**

**end**

---

## B.2 Multivariate DLF model

In this section, we provide details of the multivariate DLF model introduced in Section 3.1. With given the design points $\{\boldsymbol{x}_1^{(d)}, \ldots, \boldsymbol{x}_m^{(d)}\}$, let $\boldsymbol{f} = (\boldsymbol{f}(\boldsymbol{x}_j^{(d)}), j = 1, \ldots, m)^\top$ and $\boldsymbol{\mu}_\theta = (\mu_\theta(\boldsymbol{x}_j^{(d)}), j = 1, \ldots, m)^\top$ be $m \times c$ matrices, and let $\boldsymbol{\Phi}_\theta = (\Phi_\theta(\boldsymbol{x}_j^{(d)}), j = 1, \ldots, m)^\top$ be an $m \times q$ matrix. Then, we assume that $\boldsymbol{f}$ follows the matrix-variate Gaussian distribution

$$\boldsymbol{f} \sim \mathcal{MN}_{m,c}(\boldsymbol{\mu}_\theta, \boldsymbol{\Phi}_\theta\boldsymbol{\Phi}_\theta^\top, LL^\top).$$

Using the properties of the matrix-variate Gaussian distribution, we can vectorize $\boldsymbol{f}$, thereby allowing the multivariate DLF model to be handled the same as the univariate case.

### B.2.1 Vectorization

According to Definition 4 in [31], the vectorization of $\boldsymbol{f}$ can be expressed as follows:

$$\text{vec}(\boldsymbol{f}) \sim \mathcal{N}_{mc}(\text{vec}(\mu_\theta), LL^\top \otimes \boldsymbol{\Phi}_\theta\boldsymbol{\Phi}_\theta^\top)$$

where $\otimes$ denotes the Kronecker product. This association arises from a factorization of the covariance matrix of the multivariate Gaussian distribution into the Kronecker product of matrices $\boldsymbol{\Phi}_\theta^\top\boldsymbol{\Phi}_\theta$ and $LL^\top$.

### B.2.2 EM algorithm for the multivariate DLF model

We explain the EM algorithm for training the multivariate DLF model in the same way as in Appendix B.1. In the multivariate case, matrix vectorization and its associated properties serve as the central tools, as detailed in Section 10.2.2 of [52].

Suppose that $n$ teacher ensemble members $\boldsymbol{f}_1(\cdot), \ldots, \boldsymbol{f}_n(\cdot)$ are given, where each $\boldsymbol{f}_i(\cdot) \colon \mathcal{X} \to \mathbb{R}^c$ is a multivariate function. As in the univariate case, KD via the multivariate DLF involves three steps. First, we choose $m$ design points $\mathcal{D}^{\text{design}} = \{\boldsymbol{x}_1^{(d)}, \ldots, \boldsymbol{x}_m^{(d)}\}$. The second step is to calculate the prediction values of $n$ many ensemble members at the design points to have $\boldsymbol{f}_i = (\boldsymbol{f}_i(\boldsymbol{x}_j^{(d)}), j = 1, \ldots, m)^\top$ for $i = 1, \ldots, n$. Here, unlike in the univariate case, each $\boldsymbol{f}_i$ should be regarded as an $m \times c$ matrix. Similarly to the univariate case, the final step employs the EM algorithm to obtain the MLE.

Similarly to the univariate case, we assume that $\boldsymbol{f}_i = \tilde{\boldsymbol{f}}_i + \boldsymbol{v}_i$, where $\boldsymbol{v}_i \sim \mathcal{MN}_{m,c}(\boldsymbol{0}, \sigma_f^2\mathbb{I}_m, \mathbb{I}_c)$ and $\tilde{\boldsymbol{f}}_i = (\tilde{\boldsymbol{f}}_i(\boldsymbol{x}_1), \ldots, \tilde{\boldsymbol{f}}_i(\boldsymbol{x}_m))^\top$ with $\tilde{\boldsymbol{f}}_i(\cdot)$s following the multivariate DLF model. From the property of the matrix-variate Gaussian distribution, we can rewrite the following $\text{vec}(\boldsymbol{f}_i) = \text{vec}(\tilde{\boldsymbol{f}}_i) +$

$\text{vec}(\boldsymbol{v}_i)$, where $\text{vec}(\boldsymbol{v}_i) \sim \mathcal{N}_{mc}(\mathbf{0}, \sigma_f^2 \mathbb{I}_{mc})$. And in the case of multivariate case, we abuse the notation to write $\theta^{(t)} := (\theta^{(t)}, L^{(t)}, {\sigma_f}^{(t)})$ like the univariate case.

From this, the complete log-likelihood is given as

$$
\begin{aligned}
\ell^{com}(\theta | \boldsymbol{f}_{1:n}, \boldsymbol{z}_{1:n}) = & -\frac{cnm}{2} \log(2\pi\sigma_f^2) - \frac{cnq}{2} \log(2\pi) - \frac{\sum_{i=1}^{n} \text{vec}(\boldsymbol{z}_i)^\top \text{vec}(\boldsymbol{z}_i)}{2} \\
& - \frac{\sum_{i=1}^{n} (\text{vec}(\boldsymbol{f}_i) - \text{vec}(\tilde{\boldsymbol{f}}_i))^\top (\text{vec}(\boldsymbol{f}_i) - \text{vec}(\tilde{\boldsymbol{f}}_i))}{2\sigma_f^2} \\
= & -\frac{cnm}{2} \log(2\pi\sigma_f^2) - \frac{cnq}{2} \log(2\pi) - \frac{\sum_{i=1}^{n} \text{vec}(\boldsymbol{z}_i)^\top \text{vec}(\boldsymbol{z}_i)}{2} \\
& - \frac{\sum_{i=1}^{n} \boldsymbol{B}_i^\top \boldsymbol{B}_i}{2\sigma_f^2},
\end{aligned}
$$

where $\boldsymbol{B}_i = \text{vec}(\boldsymbol{f}_i) - \text{vec}(\boldsymbol{\mu}_\theta) - (\boldsymbol{\Phi}_\theta \otimes L) \text{vec}(\boldsymbol{z}_i)$, $\boldsymbol{f}_{1:n} = \{\boldsymbol{f}_1, \ldots, \boldsymbol{f}_n\}, \boldsymbol{z}_{1:n} = \{\boldsymbol{z}_1, \ldots, \boldsymbol{z}_n\}, \boldsymbol{\mu}_\theta = (\mu_\theta(\boldsymbol{x}_1^{(d)}), \ldots, \mu_\theta(\boldsymbol{x}_m^{(d)}))^\top$ is an $m \times c$ matrix and $\boldsymbol{\Phi}_\theta = (\Phi_\theta(\boldsymbol{x}_1^{(d)}), \ldots, \Phi_\theta(\boldsymbol{x}_m^{(d)}))^\top$ is an $m \times q$ matrix. Thus, for given parameter $\theta^{(t-1)}$ at time $t-1$, the E-step is to calculate the conditional expectation of the complete log-likelihood which is given as

$$
\begin{aligned}
Q(\theta | \theta^{(t-1)}) = & \mathbb{E}_{\boldsymbol{z}_{1:n} | \boldsymbol{f}_{1:n}, \theta^{(t-1)}} [\ell^{com}(\theta | \boldsymbol{f}_{1:n}, \boldsymbol{z}_{1:n})] \\
= & -\frac{cnm}{2} \log(2\pi\sigma_f^2) - \frac{cnq}{2} \log(2\pi) - \frac{\sum_{i=1}^{n} \text{tr}\,\mathbb{E}[\text{vec}(\boldsymbol{z}_i) \text{vec}(\boldsymbol{z}_i)^\top | \theta^{(t-1)}, \boldsymbol{f}_{1:n}]}{2} \\
& - \frac{1}{2\sigma_f^2} \sum_{i=1}^{n} \bigg\{ (\text{vec}(\boldsymbol{f}_i) - \text{vec}(\boldsymbol{\mu}_\theta))^\top (\text{vec}(\boldsymbol{f}_i) - \text{vec}(\boldsymbol{\mu}_\theta)) \\
& + 2(\text{vec}(\boldsymbol{f}_i) - \text{vec}(\boldsymbol{\mu}_\theta))^\top (\boldsymbol{\Phi}_\theta \otimes L) \mathbb{E}[\text{vec}(\boldsymbol{z}_i) | \theta^{(t-1)}, \boldsymbol{f}_{1:n}] \\
& + \text{tr}\left( (L^\top \otimes \boldsymbol{\Phi}_\theta^\top)(\boldsymbol{\Phi}_\theta \otimes L) \mathbb{E}[\text{vec}(\boldsymbol{z}_i) \text{vec}(\boldsymbol{z}_i)^\top | \theta^{(t-1)}, \boldsymbol{f}_{1:n}] \right) \bigg\},
\end{aligned}
$$

where

$$
\mathbb{E}\left[ \text{vec}(\boldsymbol{z}_i) | \theta^{(t-1)}, \boldsymbol{f}_{1:n} \right] = \frac{\mathbb{V}\left[ \text{vec}(\boldsymbol{z}) | \theta^{(t-1)}, \boldsymbol{f}_{1:n} \right] \left( {L^{(t-1)}}^\top \otimes \boldsymbol{\Phi}_{\theta^{(t-1)}}^\top \right) (\boldsymbol{f}_i - \boldsymbol{\mu}_{\theta^{(t-1)}})}{\sigma_f^{2\,(t-1)}}
$$

$$
\begin{aligned}
\mathbb{E}\left[ \text{vec}(\boldsymbol{z}_i) \text{vec}(\boldsymbol{z}_i)^\top | \theta^{(t-1)}, \boldsymbol{f}_{1:n} \right] = & \mathbb{V}\left[ \text{vec}(\boldsymbol{z}) | \theta^{(t-1)}, \boldsymbol{f}_{1:n} \right] \\
& + \mathbb{E}\left[ \text{vec}(\boldsymbol{z}) | \theta^{(t-1)}, \boldsymbol{f}_{1:n} \right] \mathbb{E}\left[ \text{vec}(\boldsymbol{z}) | \theta^{(t-1)}, \boldsymbol{f}_{1:n} \right]^\top
\end{aligned}
$$

where $\mathbb{V}\left[ \text{vec}(\boldsymbol{z}) | \theta^{(t)} \right] = \left( I_{qk} + \left( {L^{(t)}}^\top \otimes \boldsymbol{\Phi}_{\theta^{(t)}}^\top \right) \left( L^{(t)} \otimes \boldsymbol{\Phi}_{\theta^{(t)}} \right) / \sigma_f^{2\,(t)} \right)^{-1}$.

In the M-step, instead of maximizing $Q(\theta | \theta^{(t-1)})$, we update $\theta$ by use of a stochastic gradient descent algorithm.

## C   Experimental details

In this section, we describe the overall setup of our experiments in detail, focusing on the datasets, model architectures, training procedures, and evaluation metrics used in regression and classification problems. Four baselines are considered: (i) a small ensemble of lightweight networks ("small-Ens"), (ii) Hydra, and (iii) BE for regression and LBE for classification.

### C.1   Regression case

We consider the standard regression problem

$$
y = f^*(x) + \epsilon, \epsilon \sim \mathcal{N}\left(0, \sigma_\epsilon^2\right).
$$

Suppose that there are $n$ many teacher models $f_i^{(t)}(\cdot)$ and $\sigma_{\epsilon,i}^{(t)}$. Then, the predictive distribution of $y$ given $\boldsymbol{x}$ is constructed as $p^{(t)}(y|\boldsymbol{x}) = \frac{1}{n}\sum_{i=1}^{n} \mathcal{N}(y|f_i^{(t)}(\boldsymbol{x}), \sigma_{\epsilon,i}^{(t)2})$, where $\mathcal{N}(y|\mu, \sigma^2)$ is the density of the Gaussian distribution with mean $\mu$ and variance $\sigma^2$ The predictive distribution $p^{(s)}(y|\boldsymbol{x})$ based on student ensemble members is defined similarly.

We obtain 50 teacher models of DNNs with two hidden layers and 100 nodes at each layer which are learned by minimizing the sum of squared residuals of the training data with 50 randomly selected initial solutions. For the design points used in the distillation, we use the training data themselves. The architecture of student models comprises of an one-hidden-layer MLP with 50 units. The number of parameters in the student ensemble of each method is summarized in Table 4. Note that the number of parameters of DLF is smaller than those of the other baselines because the dimension of the latent factor is 10 instead of 50.

Table 4: The number of parameters in the student ensemble.

|  | Method | Datasets | | | | | |
|---|---|---|---|---|---|---|---|
|  |  | Boston housing | Concrete | Energy | Wine | Power | Kin8nm |
| # of parameters | Teachers | 90,000 | 65,000 | 65,000 | 80,000 | 45,000 | 65,000 |
|  | small-Ens | 45,000 | 32,500 | 32,500 | 40,000 | 22,500 | 32,500 |
|  | Hydra | 6,650 | 6,150 | 6,150 | 6,450 | 5,750 | 6,150 |
|  | BE | 7,351 | 6,601 | 6,601 | 7,051 | 6,001 | 6,001 |
|  | DLF(factor dim = 10) | 1,862 | 1,612 | 1,612 | 1,762 | 1,412 | 1,612 |

The Adam [53] is used for the optimization.

### C.1.1 Dataset

We consider the following 6 UCI datasets (Boston housing, Concrete, Energy, Wine, Power Plant, Kin8nm) [46]. We divide each dataset into a 9:1 ratio randomly for the training and test data. The experiment is repeated with 10 random split to have 10 evaluation metrics.

Table 5: Description of UCI benchmark datasets used in the experiment

| Dataset | size | # of features |
|---|---|---|
| Boston housing | 506 | 13 |
| Concrete | 1030 | 8 |
| Energy | 1030 | 8 |
| Wine | 9568 | 11 |
| Power | 768 | 4 |
| Kin8nm | 8192 | 8 |

### C.1.2 Evaluation metric

Let $\{(\boldsymbol{x}_1, y_1), \cdots, (\boldsymbol{x}_{m_{\text{test}}}, y_{m_{\text{test}}})\}$ be given test data.

**Root Mean Square Error** Root Mean Square Error (RMSE) is defined as

$$\text{RMSE} = \sqrt{\frac{1}{m_{\text{test}}} \sum_{j}^{m_{\text{test}}} (y_j - \hat{\mathbb{E}}(y|\boldsymbol{x}_j))^2}$$

where $\hat{\mathbb{E}}(y|\boldsymbol{x}) = \int y\hat{p}(y|\boldsymbol{x})dy$ and $\hat{p}(y|\boldsymbol{x})$ is an estimated predictive distribution.

**Negative Log Likelihood** Negative Log Likelihood (NLL) is defined as

$$\text{NLL} = -\sum_{j=1}^{m_{\text{test}}} \log \hat{p}(y_j \mid \boldsymbol{x_j}).$$

**Continuous Ranked Probability Score**    Continuous Ranked Probability Score (CRPS) is defined as

$$\text{CRPS} = \frac{1}{m_{\text{test}}} \sum_{j=1}^{m_{\text{test}}} \int_{\mathbb{R}} \left[ \widehat{F}_j(v) - \mathbf{1}(v \geq y_j)) \right]^2 dv$$

where $\widehat{F}_j(v) = \int_{-\infty}^{v} \hat{p}(y \mid \boldsymbol{x}_j) dy$.

## C.2   Classification case

We consider a $c$-class classification problem where

$$p(y|\boldsymbol{x}, \boldsymbol{f}) = \frac{\exp\left(f_y(\boldsymbol{x})\right)}{\sum_{l=1}^{c} \exp\left(f_l(\boldsymbol{x})\right)}, \quad y \in \{1, \ldots, c\}$$

for a given (vector-valued) function $\boldsymbol{f}(\cdot) = (f_1(\cdot), \ldots, f_c(\cdot))$. For a given teacher ensemble $\boldsymbol{f}_i^{(t)}, i = 1, \ldots, n$, the teacher predictive distribution is estimated by $p^{(t)}(y|\boldsymbol{x}) = \sum_{i=1}^{n} p(y|\boldsymbol{x}, \boldsymbol{f}_i^{(t)})/n$. The student predictive distribution for a given student ensemble is defined similarly.

### C.2.1   Dataset

In classification settings, we analyze two CIFAR datasets [54]. Each dataset contains 50,000 training and 10,000 test images of natural scenes, sized $32 \times 32$ pixels.

Table 6: Description of CIFAR-10 and CIFAR-100

| Dataset | Train size | Test size | # of labels |
|---------|-----------|-----------|-------------|
| CIFAR-10 | 50000 | 10000 | 10 |
| CIFAR-100 | 50000 | 10000 | 100 |

We follow the set-up of experiments in [14]. As a teacher, we use an ensemble of four neural networks, where each model is a Wide-ResNet (WRN) [55]. Specifically, WRN-28-1 is used for CIFAR-10 and WRN-28-4 is used for CIFAR-100. And, the student model uses the WRN-16-1 network for CIFAR-10 and the WRN-28-1 network for CIFAR-100.

Training lasts 200 epochs on a single GPU using SGD with Nesterov momentum of 0.9, weight decay of $5 \times 10^{-4}$, and batch size of 128. A one-cycle cosine annealing schedule with a five-epoch linear warm-up (from 0.001 to 0.1) is employed. The number of parameters in each ensemble is summarized in Table 7.

Table 7: Number of model parameters in each ensemble.

| | Method | Datasets | |
|---|--------|----------|---|
| | | CIFAR-10 | CIFAR-100 |
| | Teachers | 1.48M | 23.488M |
| | small-Ens | 0.70M | 1.50M |
| # of parameters | Hydra | 0.18M | 0.39M |
| | LBE | 0.18M | 0.38M |
| | DLF(factor dim = 8) | 0.18M | 0.38M |

### C.2.2   Evaluation metric

**Accuracy**    Accuracy (ACC) is defined as

$$\text{ACC} = \frac{1}{m_{\text{test}}} \sum_{j=1}^{m_{\text{test}}} \mathbf{1}(y_j = \hat{y}_j),$$

where $\hat{y}_j = \arg\max_y \hat{p}(y|\boldsymbol{x}_j)$.

**Negative Log-Likelihood** Negative log-likelihood (NLL) is defined as

$$\text{NLL} = -\frac{1}{m_{\text{test}}} \sum_{j=1}^{m_{\text{test}}} \sum_{k=1}^{c} 1(y_j = k) \log \hat{p}(y = k \mid \boldsymbol{x}_j).$$

**Expected Calibration Error** Expected Calibration Error (ECE) [56] is defined as

$$\text{ECE} = \sum_{l=1}^{M} \frac{|B_l|}{m_{\text{test}}} \left| \frac{1}{|B_l|} \sum_{(\boldsymbol{x}_j, y_j) \in B_l} \mathbf{1}(y_j = \hat{y}_j) - \frac{1}{|B_l|} \sum_{(\boldsymbol{x}_j, y_j) \in B_l} p(y_j \mid \boldsymbol{x}_j) \right|,$$

where $\{B_1, \ldots, B_M\}$ is a partition of the test data $\mathcal{D}_{\text{test}}$ such that $B_l = \{(\boldsymbol{x}, y) \in \mathcal{D}_{\text{test}} \mid p(\hat{y} \mid \boldsymbol{x}) \in ((l-1)/M, l/M]\}$. In this work, $M$ is set to be 15.

### C.3 Fine-tuning of language models

The experiments are conducted on three datasets from the GLUE [50] and SuperGLUE [51] benchmark : RTE, MRPC, and WiC. Table 8 summarizes the details of each dataset.

Table 8: Description of GLUE and SuperGLUE benchmark datasets used in the experiments

| Dataset | Task Type | Description |
|---------|-----------|-------------|
| RTE | Recognizing Textual Entailment | Determines whether a given hypothesis can be inferred from a given premise sentence. |
| MRPC | Paraphrase Detection | Identifies whether two sentences are semantically equivalent. The dataset consists of sentence pairs from news sources. |
| WiC | Word Sense Disambiguation | Determines whether a specific word used in two different contexts has the same meaning. Contextual understanding of word senses is essential. |

**Model Construction** For teacher models, *RoBERTa* [49] is fine-tuned with the Low-Rank Adaptation (LoRA) method [48] on each task. For each dataset, four fine-tuned models are trained using different random initializations to construct teacher ensemble.

Student models are based on *DistilRoBERTa* [1], which is also fine-tuned with LoRA. Features are extracted through the shared backbone, and task-specific prediction heads are constructed depending on the design of each distillation method.

**Training and Evaluation Settings** The experimental settings, including learning rate, batch size, number of epochs, and the rank of LoRA, follow the configuration used in [57]. Model performances are evaluated using the three metrics described in Appendix C.2: Acc, NLL and ECE.

### C.4 Application to distribution shift

To apply the framework introduced in Section 5.3 to classification tasks, we consider the following model:

$$y \mid x, \mathbf{W} \sim \text{Multinomial}\big(\text{softmax}\big(f(x; \mathbf{W})\big)\big),$$
$$f(x; \mathbf{W}) = \mu_\theta(x) + L\mathbf{W}\Phi_\theta(x),$$

where $\mathbf{W}$ is the weight matrix. For new data $\mathcal{D}^{new}$, we estimate $\mathbf{W}$ by minimizing the cross-entropy on new data while $\theta$ and $L$ are fixed at $\hat{\theta}$ and $\hat{L}$ estimated on training data.

For numerical study, we consider a distribution shift scenario where the conditional distribution $P(X \mid Y)$ changes [58]. To generate synthetic data, we use CIFAR-10 and flip the labels by $y \mapsto 9 - y$ to construct CIFAR 10-Flip dataset [44]. We first learn the body by a DNN, Hydra, and DLF on the training data of 50,000 CIFAR-10 images under the WRN-16-1 architecture for the teacher and student ensembles. Then, we train the weights of the linear head on new data while the body is fixed.

## C.5 Hardwares

All our experiments are done through Python 3.9.16 with Intel(R) Xeon(R) Silver 4310 CPU @ 2.10GHz, NVIDIA TITAN Xp GPU and 128GB RAM.

# D Ablation Study

We do the following ablation studies on Boston housing data.

- We investigate how the choice of design points affects distillation performance.
- The effect of the choice of latent factor dimension affects distillation performance.
- The effect of the MMD-based initialization is investigated to assess its role in stabilizing the EM algorithm during training.
- The capacity of the student models is varied by adjusting the network width, and the effect of this change is analyzed for each baseline.
- We investigate the effect of ensemble size by varying the number of ensemble members used in the teacher and student ensembles.

Quantitative results are presented using RMSE, NLL, and CRPS, aggregated over ten independent runs.

## D.1 Choice of design points

### D.1.1 Comparison of different design points selection methods

We investigate how different strategies of selection design points influence the performance of the Gaussian distillation. The entire dataset is partitioned into three disjoint subsets: $\mathcal{D} = \mathcal{D}_{\text{teacher}}^{\text{train}} \oplus \mathcal{D}_{\text{new}}^{\text{train}} \oplus \mathcal{D}^{\text{test}}$, with a fixed ratio of $4.5 : 4.5 : 1$. The teacher ensemble is trained on $\mathcal{D}_{\text{teacher}}^{\text{train}}$, and the student model is distilled using various forms of design points $\mathcal{D}^{\text{design}}$. To analyze the impact of the choice of design points, the four distinct strategies for selecting $\mathcal{D}^{\text{design}}$ are considered:

- **Design 1:** Directly using the teacher training data, $\mathcal{D}^{\text{design}} = \mathcal{D}_{\text{teacher}}^{\text{train}}$.
- **Design 2:** Using mixup samples from $\mathcal{D}_{\text{teacher}}^{\text{train}}$. $\mathcal{D}^{\text{design}} \subset \text{mixup}\{\mathcal{D}_{\text{teacher}}^{\text{train}}\}$.
- **Design 3:** Using a training data not used in training teacher ensemble, $\mathcal{D}^{\text{design}} = \mathcal{D}_{\text{new}}^{\text{train}}$.
- **Design 4:** Using mixup samples from $\mathcal{D}_{\text{new}}^{\text{train}}$. $\mathcal{D}^{\text{design}} \subset \text{mixup}\{\mathcal{D}_{\text{new}}^{\text{train}}\}$

Here, $\text{mixup}\{\mathcal{D}_{\text{teacher}}^{\text{train}}\}$ denotes the set of samples generated by linearly combining two randomly selected samples from $\mathcal{D}_{\text{teacher}}^{\text{train}}$. For each index $j$, we randomly draw two data pairs $(\boldsymbol{x}_j, y_j)$ and $(\boldsymbol{x}_j^c, y_j^c)$ from $\mathcal{D}_{\text{teacher}}^{\text{train}}$ and form the mixed sample

$$(\boldsymbol{x}_j^m, y_j^m) := \lambda_j(\boldsymbol{x}_j, y_j) + (1 - \lambda_j)(\boldsymbol{x}_j^c, y_j^c), \quad \lambda_j \in [0, 1].$$

The set $\text{mixup}\{\mathcal{D}_{\text{teacher}}^{\text{train}}\}$ consists of all mixed pairs $(\boldsymbol{x}_j^m, y_j^m)$, with the number of generated design points in Designs 2 and 4 matched to the size of $\mathcal{D}_{\text{teacher}}^{\text{train}}$. These strategies are designed to cover both scenarios where design points are reused from the teacher training data and where additional or perturbed data are incorporated. All methods are evaluated on the reserved test data $\mathcal{D}^{\text{test}}$.

Table 9: Performances of Gaussian distillation for different strategies of the design point selection

| Design Strategy | RMSE | NLL | CRPS |
|:---:|:---:|:---:|:---:|
| Design 1 | 3.2316 (0.0587) | 2.6317 (0.0267) | 1.7896 (0.0271) |
| Design 2 | 3.5856 (0.1263) | 2.8003 (0.0638) | 1.9648 (0.0681) |
| Design 3 | **2.8786 (0.0322)** | **2.4816 (0.0129)** | **1.6427 (0.0165)** |
| Design 4 | 3.1787 (0.2498) | 2.6122 (0.1144) | 1.792 (0.1310) |

The results summarized in Table 9 and visualized in Figure 3 illustrate the effect of different choices of design points on the performance of the Gaussian distillation.. Among the four strategies, the use

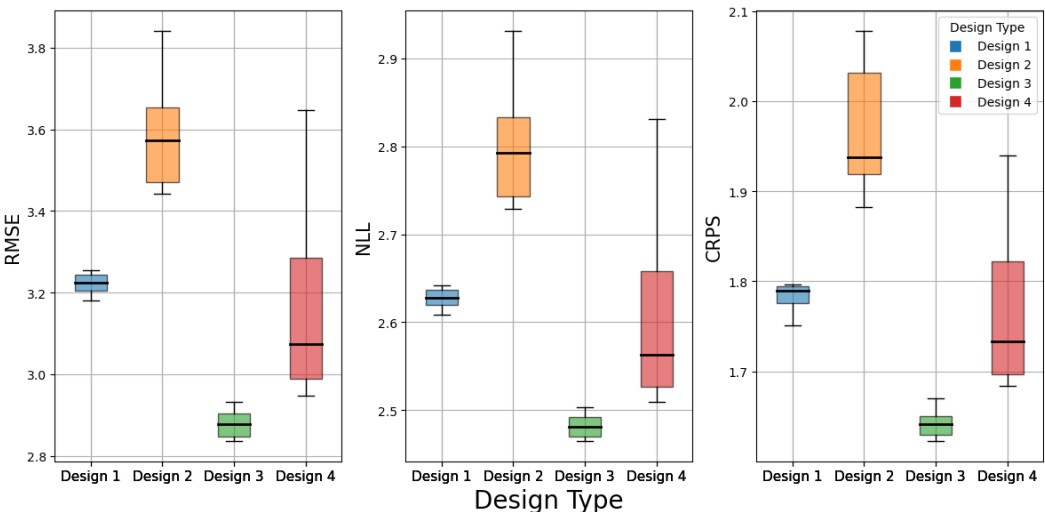

Figure 3: Box plot of evaluation metrics (RMSE, NLL, CRPS) for different strategies of the design point selection

of new training data (Design 3) for design points consistently yields the best performance across all metrics. In contrast, strategies involving mixup consistently yield inferior performances regardless of whether teacher training data (Design 2) or new data (Design 4) are used.

To sum up, the results suggest that using new training data is the best for Gaussian distillation. However, the size of the teacher training data becomes smaller, which might lead to performance degradation. In practice, we could find the optimal partition of teacher training data and new training data based on additional validation data.

### D.1.2 Comparison for different selection of the number of design points

We also investigate the effect of the number of design points on the student model. As in the previous experimental setup, the teacher ensemble is trained on $\mathcal{D}_{\text{teacher}}^{\text{train}}$, and the student model is distilled using various sizes of design sets $\mathcal{D}^{\text{design}}$. We only consider the design type $\mathcal{D}_{\text{teacher}}^{\text{train}}$ (Design 1) and $\mathcal{D}_{\text{new}}^{\text{train}}$ (Design 3). The number of design points is controlled by the *design ratio*

$$r \in \{0.2, 0.4, 0.6, 0.8, 1.0\}$$

meaning that Gaussian distillation uses $r \times |\mathcal{D}^{\text{design}}|$ samples for distillation.

Table 10: Performance for Gaussian distillation for the number of design points

| Design Type | | Design 1 | | | Design 3 | |
|---|---|---|---|---|---|---|
| design_ratio | RMSE | NLL | CRPS | RMSE | NLL | CRPS |
| 0.2 | 3.9829 (0.4823) | 3.0244 (0.2738) | 2.1773 (0.1704) | 3.8643 (0.3277) | 2.951 (0.1728) | 2.145 (0.1349) |
| 0.4 | 3.576 (0.3231) | 2.8017 (0.1612) | 1.9818 (0.1349) | 3.0703 (0.0735) | 2.5612 (0.0316) | 1.7464 (0.0396) |
| 0.6 | 3.3996 (0.1750) | 2.711 (0.0834) | 1.9122 (0.0573) | 3.0055 (0.0634) | 2.5337 (0.0267) | 1.7161 (0.0243) |
| 0.8 | 3.2548 (0.1407) | 2.6433 (0.0642) | 1.8444 (0.0663) | 3.0229 (0.0945) | 2.5413 (0.0395) | 1.7197 (0.0444) |
| 1 | 3.2316 (0.0587) | 2.6317 (0.0267) | 1.7896 (0.0271) | 2.8786 (0.0322) | 2.4816 (0.0129) | 1.6427 (0.0165) |

The results summarized in Table 10 and Figure 5 show that increasing the design ratio consistently improves the performance of the Gaussian distillation. That is, the larger the number of design points, the better the performance of Gaussian distillation is.

### D.2 Dimension of the latent factor

We investigate the influence of the dimension of the latent factor on the performance of Gaussian distillation. The result visualized in Figure 5 indicates that the performance of Gaussian distillation is not sensitive to the dimension of the latent factor unless the dimension is too small.

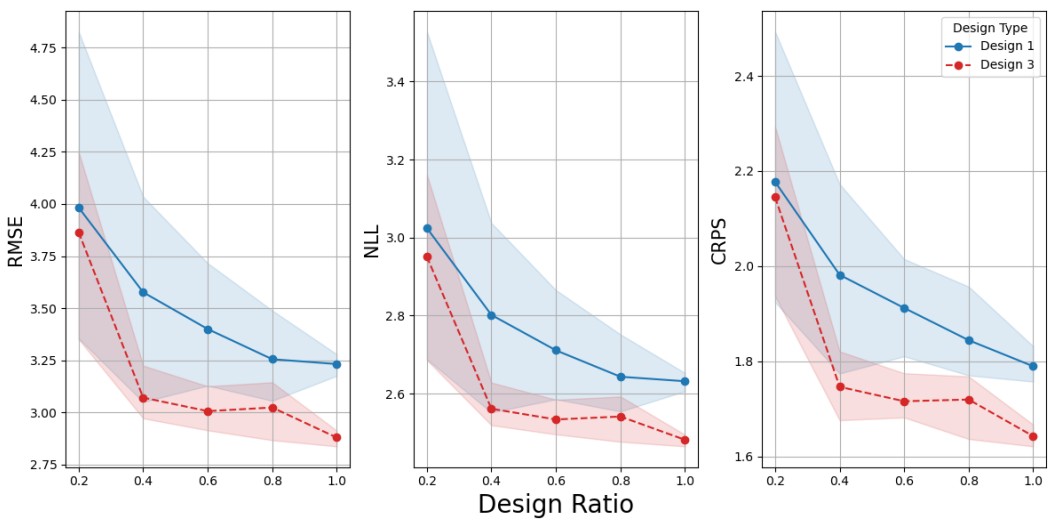

Figure 4: Evaluation metrics (RMSE, NLL, CRPS) versus the design ratio.

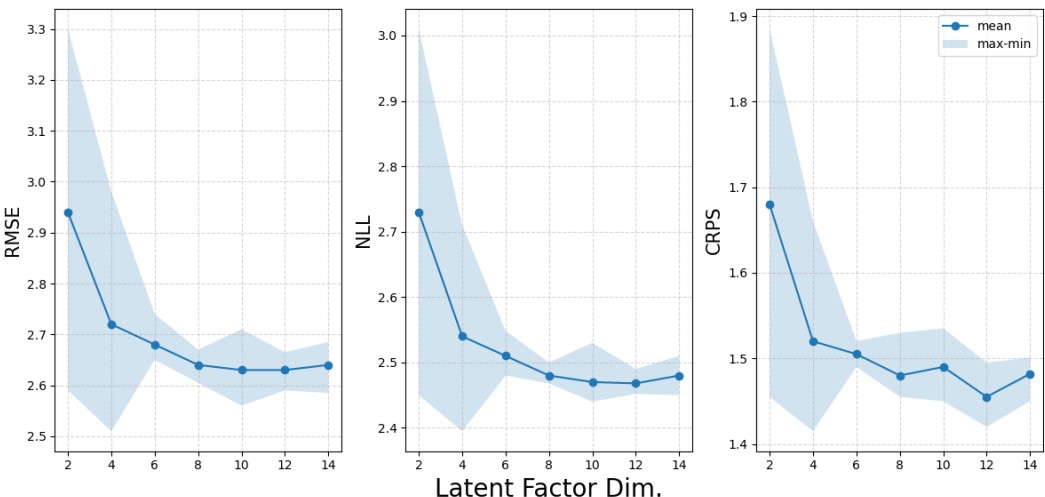

Figure 5: Evaluation metrics (RMSE, NLL, CRPS) versus latent factor dimension.

### D.3 MMD vs random initial

We compare the proposed MMD initialization in Section 3.2 with the random initialization. As shown in Figure 6, the MMD initialization strategy outperforms the random initialization unless the dimension of the latent factor is too small. In addition, the variations of the evaluation metrics for the MMD initialization are much smaller than those of the random initialization. That is, the MMD initialization is indispensable for the superior performance of Gaussian distillation.

### D.4 Capacity of student models

The effect of the capacity of student models is examined by varying the number of nodes in the one-layer DNN architecture. We increase the number of nodes gradually from 50 to 60, 70, 80, 90, and 100, and obtain the evaluation metrics of the distillation methods. The results in Figure 7 show that the performances of DLF and small-Ens keep improving as the number of nodes increases, while

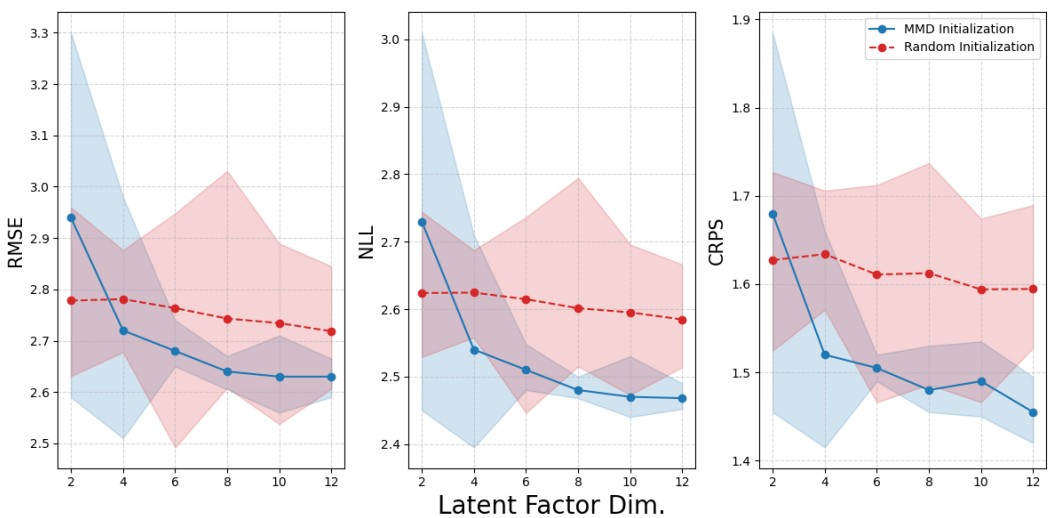

Figure 6: Evaluation metrics (RMSE, NLL, CRPS) versus latent factor size with or without the MMD initialization

the performances of Hydra and BE are saturated. Apparently, DLF behaves similarly to small-Ens, which is interesting since small-Ens demands heavier computation and much more storage.

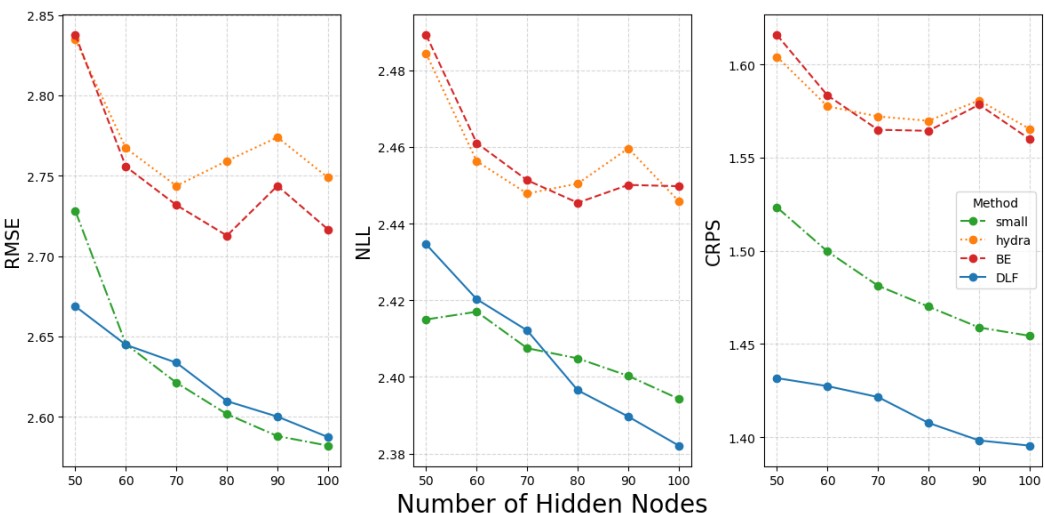

Figure 7: Evaluation metrics (RMSE, NLL, CRPS) versus the number of hidden nodes.

### D.5   Number of ensemble members

To evaluate the effect of ensemble size, we evaluate the performance of the KD methods by varying the number of ensemble members from 10 to 50. As we can see from Figure 8, for all methods, the performances keep improving as the ensemble size increases. Note that the number of weights in the DLF model is not proportional to the ensemble sizes (instead, it is proportional to the dimension of the latent factor), while it is proportional for the other baselines. This is an additional advantage of Gaussian distillation.

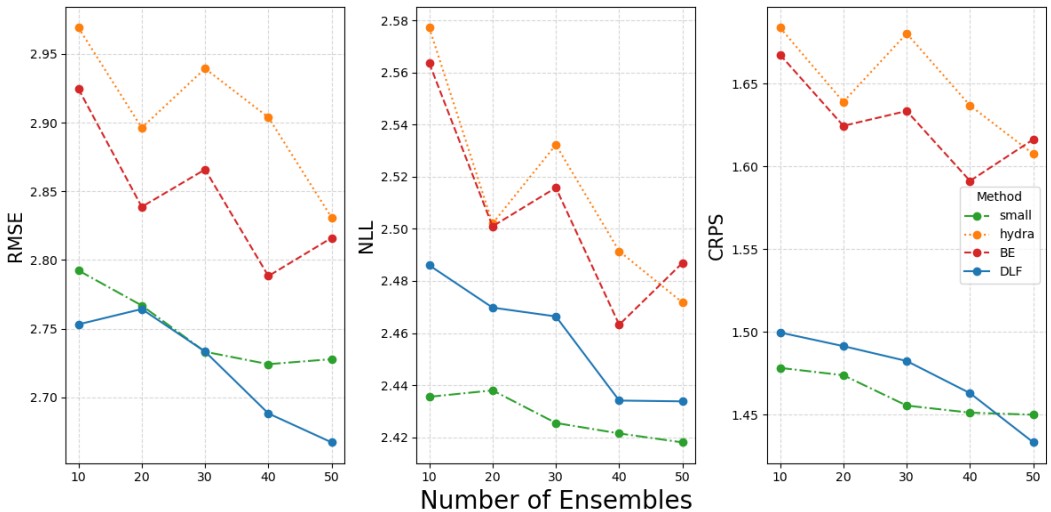

Figure 8: Evaluation metrics (RMSE, NLL, CRPS) versus the number of ensemble members.

### D.6 Application to large dataset

To further examine the scalability and robustness of our proposed model, we conducted an additional experiment on a larger and more complex dataset, Tiny ImageNet [59]. This dataset extends the original ImageNet hierarchy but contains a reduced subset of classes and image resolutions, making it a challenging yet computationally manageable benchmark for evaluating model generalization on large-scale visual domains.

Tiny ImageNet consists of 200 classes with 100,000 training images and 10,000 test images, where each image is resized to 64×64 pixels. To ensure fair evaluation and avoid overfitting, we randomly split the training set into 80% for training and 20% for validation.

We compared our method against Small Ensemble, Hydra, LBE, and Proxy-EnD$^2$. The teacher network was implemented using WRN-28-4, while the student network employed WRN-16-4. Both teacher and student models were trained under an ensemble size of four, following the same distillation pipeline described in Appendix C.2.

Table 11: Results on Tiny ImageNet.

| dataset | method | Acc(%) ↑ | NLL ↓ | ECE(%) ↓ |
|---|---|---|---|---|
| Tiny ImageNet | Teacher | 68.74 | 1.2806 | 3.22 |
| | Small-Ens | 57.11 (0.0027) | 1.854 (0.0032) | 13.51 (0.0032) |
| | Hydra | 56.80 (0.0037) | 1.7503 (0.0092) | 2.75 (0.092) |
| | LBE | 46.08 (0.0119) | 2.24 (0.0407) | 2.22 (0.0053) |
| | Proxy-EnD$^2$ | 51.64 (0.0022) | 2.0195 (0.0069) | 2.77 (0.0038) |
| | DLF | **58.34 (0.2121)** | **1.6737 (0.0043)** | **1.92 (0.1098)** |

Table 11 shows that our approach outperforms all other methods across all measures, even when the number of classes and samples is large, suggesting that the proposed framework generalizes well beyond small-scale datasets. In contrast, both LBE and Proxy-EnD$^2$ exhibit a significant drop in performance when applied to complex data, confirming that our latent factor modeling remains effective even in such challenging settings.

## E   Uncertainty quantification

We conduct experiments to evaluate whether each model appropriately quantifies uncertainty by detecting out-of-distribution (OOD) data in a classification problem. We adopt predictive mutual

information, which estimates epistemic uncertainty [6], as the OOD detection score :

$$\widehat{\mathbb{I}}\big[y, \theta \mid \boldsymbol{x}\big] := -\sum_{y=1}^{c}\left(\frac{1}{n}\sum_{i=1}^{n}p(y \mid \boldsymbol{x}, \hat{\theta}_i)\right)\log\left(\frac{1}{n}\sum_{i=1}^{c}p(y \mid \boldsymbol{x}, \hat{\theta}_i)\right)$$
$$+ \frac{1}{n}\sum_{y=1}^{c}\sum_{i=1}^{n}p(y \mid \boldsymbol{x}, \hat{\theta}_i)\log p(y \mid \boldsymbol{x}, \hat{\theta}_i).$$

Specifically, we denote the in-distribution (ID) dataset by $\{\boldsymbol{x}_1^{\text{in}}, ..., \boldsymbol{x}_{m_1}^{\text{in}}\}$, the OOD dataset by $\{\boldsymbol{x}_1^{\text{out}}, ..., \boldsymbol{x}_{m_2}^{\text{out}}\}$ and $y$ is the output of the model with corresponding predictive mutual information $\widehat{\mathbb{I}}\big[y, \theta \mid \boldsymbol{x}_i^{\text{in}}\big]$ for $i = 1, ..., m_1$ and $\widehat{\mathbb{I}}\big[y, \theta \mid \boldsymbol{x}_i^{\text{out}}\big]$ for $i = 1, ..., m_2$.

For evaluation, we assign label 0 to ID data and label 1 to OOD data, meaning that lower predictive mutual information indicates ID whereas higher values indicate OOD. We then compute the AUROC between these labels and the predictive mutual information scores.

We use CIFAR-10 as the ID and SVHN, CIFAR-100, and Tiny ImageNet as OOD. All models are trained on the CIFAR-10 training set with 50,000 images, and OOD detection is evaluated on the CIFAR-10 test set with 10,000 images versus the test sets of the OOD datasets, where SVHN has 26,032 images, CIFAR-100 has 10,000 images, and Tiny ImageNet has 10,000 images. We repeat the entire procedure five times with different random seeds and report the mean of AUROC and its standard error. Table 12 show that DLF performs best on SVHN and remains competitive across CIFAR-100 and Tiny ImageNet.

Table 12: AUROC Results on in-distribution and out-of-distribution detection.

| Method | Out-Of-Distribution Datasets | | |
|---|---|---|---|
| | SVHN | CIFAR-100 | Tiny ImageNet |
| Teachers | 0.9403 | 0.8604 | 0.9400 |
| small-Ens | 0.9093 (0.0037) | 0.8000 (0.0036) | 0.7956 (0.0023) |
| Hydra | 0.7178 (0.0107) | 0.6644 (0.0083) | 0.6709 (0.0059) |
| LBE | 0.8329 (0.0249) | 0.8107 (0.0130) | 0.8152 (0.0141) |
| Proxy-EnD$^2$ | 0.8427 (0.0344) | **0.8427 (0.0183)** | **0.8456 (0.0175)** |
| DLF | **0.9359 (0.0212)** | 0.8357 (0.0062) | 0.8291 (0.0067) |

# F    Computational Cost

In this section, we report a comparison of the computational costs between our proposed method and baseline method during training.

## F.1    Training time

First, we evaluate the training time on the CIFAR-10 and CIFAR-100 datasets. For a fair comparison, all experiments are conducted under the same hardware environment. We follow the experimental setup described in Appendix C.2. Each experiment is repeated four times, and the average training times are reported in hours.

Table 13: Comparison of training time on CIFAR-10 and CIFAR-100 datasets.

| Method | CIFAR-10 Training Time | CIFAR-100 Training Time |
|---|---|---|
| Teachers | 10.2 | 17.4 |
| small-Ens | 4.6 | 12.5 |
| Hydra | 3.1 | 10.28 |
| LBE | 6.5 | 21.5 |
| Proxy-EnD$^2$ | 4.6 | 9.2 |
| DLF (Ours) | 4.2 | 11.28 |

As shown in Table 13, our proposed DLF model also exhibits a relatively short training time compared to small ensemble, Hydra, and Proxy-EnD$^2$. In the DLF model, $\mu_\theta$ and $\Phi_\theta$ are local networks whose

output sizes depend only on the target and latent dimensions; hence, they do not scale with the student model size. Moreover, a key advantage of DLF is that it can be effectively trained using EM algorithm. As detailed in Appendix B, the E-step has a closed-form solution, making the overall training process computationally efficient. In contrast, LBE requires more computational resources and longer training time as the model becomes larger, which will be discussed further in the following subsection.

### F.2 Floating-point operations

For a further analyze computational complexity, we compute Floating-point operations (FLOPs) per training for Hydra, Batch Ensemble and DLF. FLOPs provide a hardware-independent measure of computational cost. For example, ResNet [60] and EfficientNet [61] use FLOPs to evaluate and compare model efficiency across architectures.

Let $c$ denote the number of classes, $B$ the batch size, and $n$ the number of ensemble members. The latent dimension in DLF is denoted by $q$. $F_{\text{body}}$ and $F_{\text{head}}$ represent the FLOPs for one forward pass through the share body (e.g., Wide-ResNet [55]) and one Hydra head, respectively. In the DLF model, $F_{\mu,\Phi}$ corresponds to the FLOPs for the small fully connected layers that produce $\mu_\theta$ and $\Phi_\theta$. Finally, $\alpha$ indicates the multiplier accounting for both forward and backward passes. Then, the FLOPs per training step for Hydra, Batch Ensemble, and DLF are formulated as follows:

- **Hydra:**
$$FLOPs_{\text{Hydra}} = \alpha B(F_{\text{body}} + nF_{\text{head}})$$

- **Batch Ensemble :**
$$FLOPs_{\text{BE}} \approx \alpha B(1 + \epsilon)F_{\text{body}}$$
where $\epsilon$ (extra cost from rank-1 matrix) is usually $\leq 0.05$.

- **DLF:**
$$FLOPs_{\text{DLF}} = \alpha B(F_{\text{body}} + F_{\mu,\Phi}) + Bq^2 + q^3 + n(Bcq + q^2)$$

Note that when the shared backbone is a Wide-ResNet, the corresponding $F_{\text{body}}$ is typically on the order of $10^9$ FLOPs. In Batch Ensemble, the additional cost comes from the overhead factor $(1 + \epsilon)$ applied to the full backbone. Although $\epsilon$ is usually small (typically less than 0.05), its effect is not negligible due to the large scale of $F_{\text{body}}$.

In contrast, Hydra shares the backbone and only adds a small cost from the lightweight head modules. Since $F_{\text{head}} \ll F_{\text{body}}$, the total complexity remains dominated by the shared body, even with large ensemble sizes $n$.

Similarly, although DLF requires additional computations such as $F_{\mu,\Phi}$, and matrix operations such as $Bq^2$, $q^3$, and $n(Bcq + q^2)$, their contributions are negligible since $q$ is typically small (often less than 20 in practice) and $F_{\mu,\Phi}, Bq^2, q^3, n(Bcq + q^2) \ll F_{\text{body}}$. Consequently, the dominant cost in DLF also comes from the shared backbone.

## G   Theoretical results

We intend to similarly investigate the convergence of the sieve maximum likelihood estimation (MLE), as discussed by [62], using the results of estimating smooth functions within the sparse deep neural network (DNN) function class proposed by [63] and sieve MLE's convergence rates by [64]. We will investigate the convergence rate in terms of decaying rates of the eigenvalues.

**Notation**   For a natural number $m$, we define $[m] = \{1, 2, \ldots, m\}$. For a $m \times q$-dimensional matrix $A$, we denote the spectral norm of the matrix $A$ by $\|A\|_2$ and the Frobenius norm by $\|A\|_F$, that is, $\|A\|_2 = \sup_{z \in \mathbb{R}^q : \|z\|_2 = 1} \|Az\|_2$ and $\|A\|_F = \sqrt{\text{Tr}(A^\top A)}$. For a square matrix $A$, let $\lambda_{\min}(A)$ denote the smallest eigenvalue of $A$. For two positive sequences $(a_n)_{n \in \mathbb{N}}$ and $(b_n)_{n \in \mathbb{N}}$, we write $a_n \lesssim b_n$ or $b_n \gtrsim a_n$, if there exists a positive constant $C > 0$ such that $a_n \leq Cb_n$ for any $n \in \mathbb{N}$. We write $a_n \asymp b_n$ if both $a_n \gtrsim b_n$ and $a_n \lesssim b_n$ hold. For a vector-valued function $\boldsymbol{f} = (f_1, \ldots, f_m)^\top$ defined on a domain $\mathcal{X}$, we denote the "elementwise-maximum" supremum norm by $\|\boldsymbol{f}\|_\infty = \max_{1 \leq j \leq m} \|f_j\|_\infty = \max_{1 \leq j \leq m} \sup_{x \in \mathcal{X}} |f_j(x)|$. The Hölder space of smoothness

$\beta > 0$ with domain $[0,1]^d$ and radius $K > 0$ is defined as, letting $s$ be the smallest integer larger than or equal to $\beta - 1$,

$$\mathcal{H}_d^\beta(K) := \left\{ f \in \mathcal{C}_d^s : \|f\|_{\mathcal{H}_d^\beta} \le K \right\},$$

where $\mathcal{C}_d^s$ denotes the set of $s$-times differentiable functions on $[0,1]^d$ and $\|\cdot\|_{\mathcal{H}_d^\beta}$ denotes the Hölder norm defined by

$$\|f\|_{\mathcal{H}^\beta(\mathcal{X})} = \max_{(\alpha_1,\dots,\alpha_d)\in\mathbb{N}_0^d:\sum_{j=1}^d \alpha_j < \beta} \|\partial^{\alpha_1}\cdots\partial^{\alpha_d} f\|_\infty$$

$$+ \max_{(\alpha_1,\dots,\alpha_d)\in\mathbb{N}_0^d:\sum_{j=1}^d \alpha_j = s} \sup_{\boldsymbol{x}_1,\boldsymbol{x}_2\in[0,1]^d, \boldsymbol{x}_1\ne\boldsymbol{x}_2} \frac{|\partial^{\alpha_1}\cdots\partial^{\alpha_d} f(\boldsymbol{x}_1) - \partial^{\alpha_1}\cdots\partial^{\alpha_d} f(\boldsymbol{x}_2)|}{\|\boldsymbol{x}_1 - \boldsymbol{x}_2\|_\infty^{\beta-s}},$$

where $\mathbb{N}_0 = \{0,1,2,\dots\}$. Let $\mathcal{F}$ be a given class of functions defined on $\mathcal{X}$. A collection $\{f_i : i \in [N]\}$ is called a $\delta$-covering set of $\mathcal{F}$ with respect to a certain norm $\|\cdot\|$ defined on $\mathcal{X}$, if, for all $f \in \mathcal{F}$, there exists $f_i$ in the collection such that $\|f - f_i\| \le \delta$. The cardinality of the minimal $\delta$-covering set is called the $\delta$-covering number of $\mathcal{F}$ with respect to the norm $\|\cdot\|$, and is denoted by $\mathcal{N}(\delta, \mathcal{F}, \|\cdot\|)$. A collection $\{(l_i, u_i) : i \in [N]\}$ of pairs of functions with $l_i \le u_i$ is called a $\delta$-bracketing set of $\mathcal{F}$ with respect to a norm $\|\cdot\|$ if, for all $f \in \mathcal{F}$, there exists $(l_i, u_i)$ in the collection such that $l_i \le f \le u_i$ and $\|l_i - u_i\| < \delta$. The cardinality of the minimal $\delta$-bracketing set is called the $\delta$-bracketing number of $\mathcal{F}$ with respect to the norm $\|\cdot\|$, and is denoted by $\mathcal{N}_{[]}(\delta, \mathcal{F}, \|\cdot\|)$. For two probability density functions $p_1$ and $p_2$, let us denote the Hellinger distance between them by $h(p_1, p_2) = [\int (p_1^{1/2}(\boldsymbol{x}) - p_2^{1/2}(\boldsymbol{x}))^2 d\boldsymbol{x}]^{1/2}$.

## G.1 Problem formulation

Let $\tilde{f}_1, \dots, \tilde{f}_n$ be $n$ independent realizations of the Gaussian process with mean function $\mu_*(\cdot)$ and covariance kernel $\Sigma_*(\cdot,\cdot)$. Suppose that we have $m$ many $d$-dimensional design points $\mathcal{D} = \{\boldsymbol{x}_1^{(d)}, \dots, \boldsymbol{x}_m^{(d)}\}$ (where we omit the superscript "design" unlike the main body of the manuscript for simplicity). We assume that without loss of generality, $\boldsymbol{x}_i^{(d)} \in [0,1]^d$ for every $j \in [m]$ by appropriate normalization. Then we observe $\boldsymbol{f}_{i|\mathcal{D}} = (\tilde{f}_i(\boldsymbol{x}_1^{(d)}) + v_{i1}, \dots, \tilde{f}_i(\boldsymbol{x}_m^{(d)}) + v_{im})^\top$, where $v_{im}$s are independent Gaussian random variables with mean 0 and variance $\sigma_*^2$. Note that $\boldsymbol{f}_{i|\mathcal{D}}$ follows the multivariate normal distribution $p_* := \mathcal{N}(\boldsymbol{\mu}_{*|\mathcal{D}}, \boldsymbol{\Sigma}_{*|\mathcal{D}})$, where

$$\boldsymbol{\mu}_{*|\mathcal{D}} = (\mu_*(\boldsymbol{x}_u^{(d)}))_{u\in[m]} \in \mathbb{R}^m$$
$$\boldsymbol{\Sigma}_{*|\mathcal{D}} = (\Sigma_*(\boldsymbol{x}_u^{(d)}, \boldsymbol{x}_v^{(d)}))_{u\in[m],v\in[m]} + \sigma_*^2 \mathbb{I}_m \in \mathbb{R}^{m\times m}.$$

Let $\lambda_{*,j}$ and $\psi_{*,j}(\cdot)$ be the $j$-th eigenvalues and eigenfunctions of the kernel $\Sigma_*$, ordered by their magnitude and $\phi_{*,j}(\cdot) = \sqrt{\lambda_j}\psi_{*,j}(\cdot)$ be the scaled eigenfunctions. Then, the covariance matrix can be decomposed into the $q$-leading part and the low-rank part

$$\boldsymbol{\Sigma}_{*|\mathcal{D}} = \boldsymbol{\Phi}_{*|\mathcal{D}}\boldsymbol{\Phi}_{*|\mathcal{D}}^\top + \boldsymbol{\Sigma}_{*>q|\mathcal{D}} + \sigma_*^2 \mathbb{I}_m$$

where $\boldsymbol{\Phi}_{*|\mathcal{D}} = (\boldsymbol{\phi}_*(\boldsymbol{x}_1^{(d)}), \dots, \boldsymbol{\phi}_*(\boldsymbol{x}_m^{(d)}))^\top \in \mathbb{R}^{m\times q}$ with $\boldsymbol{\phi}_*(\boldsymbol{x}) = (\phi_{*j}(\boldsymbol{x}))_{j\in[q]}$ and $\boldsymbol{\Sigma}_{*>q|\mathcal{D}} = \left(\sum_{j>q} \phi_{*j}(\boldsymbol{x}_u^{(d)})\phi_{*j}(\boldsymbol{x}_v^{(d)})\right)_{u\in[m],v\in[m]}$. For the sake of notational simplicity, we consider the case where $\sigma_*^2$ is known. The proof can be extended easily for the case of unknown $\sigma_*^2$.

Our aim is to estimate $p^*$ based on observations $\boldsymbol{f}_{1|m}, \dots, \boldsymbol{f}_{n|m}$ by modeling $\mu_*$ and $\phi_*$ by a specially design DNN. For given $\boldsymbol{\mu} \in \mathbb{R}^m$ and $\boldsymbol{\Sigma} \in \mathbb{R}^{m\times m}$, let $p_{\boldsymbol{\mu},\boldsymbol{\Sigma}}$ be the density of the Gaussian distribution with mean $\boldsymbol{\mu}$ and covariance matrix $\boldsymbol{\Sigma}$. For give mean function $\mu_\theta(\cdot)$ and a vector of $q$ many scaled eigenfunctions $\boldsymbol{\phi}_\theta(\cdot)$ parameterized by $\theta$, we consider $p_{\boldsymbol{\mu}_{\theta|\mathcal{D}},\boldsymbol{\Sigma}_{\theta|\mathcal{D}}}$, where

$$\boldsymbol{\mu}_{\theta|\mathcal{D}} = (\mu_\theta(\boldsymbol{x}_u^{(d)}))_{u\in[m]} \in \mathbb{R}^m$$
$$\boldsymbol{\Sigma}_{\theta|\mathcal{D}} = \boldsymbol{\Phi}_{\theta|\mathcal{D}}\boldsymbol{\Phi}_{\theta|\mathcal{D}}^\top + \sigma_*^2 \mathbb{I}_m \in \mathbb{R}^{m\times m}$$

with $\boldsymbol{\Phi}_{\theta|\mathcal{D}} = (\boldsymbol{\phi}_\theta(\boldsymbol{x}_1^{(d)}), \dots, \boldsymbol{\phi}_\theta(\boldsymbol{x}_m^{(d)}))^\top$. We model $(\mu_\theta(\cdot), \boldsymbol{\phi}_\theta(\cdot))$ by a DNN with $q+1$ many outputs and estimate $\theta$ by a (sieve) maximum likelihood estimator (MLE) that is defined as $\hat{\theta} =$

$\arg\max_{\theta \in \Theta_n} \ell_n(\theta)$, where

$$\ell_n(\theta) = -\frac{n}{2}\log|\boldsymbol{\Sigma}_{\theta|\mathcal{D}}| - \frac{1}{2}\sum_{i=1}^{n}(\boldsymbol{f}_{i|\mathcal{D}} - \boldsymbol{\mu}_{\theta|\mathcal{D}})^{\top}\boldsymbol{\Sigma}_{\theta|\mathcal{D}}^{-1}(\boldsymbol{f}_{i|\mathcal{D}} - \boldsymbol{\mu}_{\theta|\mathcal{D}}).$$

Here, the sieve $\Theta_n$ depends on the architecture of DNNs. We will prove that the estimated Gaussian distribution converges to the true Gaussian distribution $p_*$ in probability as $n \to \infty$ under regularity conditions while $\mathcal{D}$ is fixed, provided that the sieve $\Theta_n$ is selected appropriately.

## G.2 Results

For the sieve $\Theta_n$, we consider a set of parameters, whose elements are in $[-1, 1]$ (following [63]), of sparse DNNs with $L_n$ many layers, $r_n$ many nodes at each hidden layers, $S_n$ many nonzero elements and $q_n + 1$ output nodes. When we would like to clarify such architectural choices, we will sometimes use the notation $\Theta_n = \Theta(L_n, r_n, S_n, q_n)$. For a DNN parameter $\theta \in \Theta_n$, we let $\boldsymbol{g}_\theta$ be the corresponding realized DNN function, but for a technical reason, the outputs of this function are truncated at $[-B, B]$, so that $\boldsymbol{g}_\theta$ is a function from $\mathbb{R}^d$ to $[-B, B]^{q+1}$. We denote by $\mathcal{G}(\Theta_n) = \{\boldsymbol{g}_\theta : \theta \in \Theta_n\}$. Such sparse DNNs have been considered in many previous studies [e.g., 62, 63, 65] to investigate the asymptotic properties of DNNs.

Given the design points $\mathcal{D} = \{\boldsymbol{x}_1^{(d)}, \ldots, \boldsymbol{x}_m^{(d)}\}$, for each DNN parameter $\theta \in \Theta_n$ with the realized DNN $\boldsymbol{g}_\theta = (g_{\theta,1}, \ldots, g_{\theta,q_n+1})^{\top}$, we define the $m$-dimensional vector $\boldsymbol{\mu}_{\theta|\mathcal{D}} = (g_{\theta,1}(\boldsymbol{x}_u^{(d)}))_{u \in [m]}$ and $m \times m$ symmetric matrix $\boldsymbol{\Sigma}_{\theta|\mathcal{D}} = \boldsymbol{\Phi}_{\theta|\mathcal{D}}\boldsymbol{\Phi}_{\theta|\mathcal{D}}^{\top} + \sigma_*^2\mathbb{I}_m$, where $\boldsymbol{\Phi}_{\theta|\mathcal{D}} = (g_{\theta,j+1}(\boldsymbol{x}_u^{(d)}))_{u \in [m], j \in [q_n]}$. For notational simplicity, we write $p_{\theta|\mathcal{D}} = p_{\boldsymbol{\mu}_{\theta|\mathcal{D}}, \boldsymbol{\Sigma}_{\theta|\mathcal{D}}}$, the density of the Gaussian distribution with mean $\mu_{\theta|\mathcal{D}}$ and covariance matrix $\Sigma_{\theta|\mathcal{D}}$. We let the class of such Gaussian distributions $\mathcal{P}(\Theta_n; \mathcal{D}) = \{p_{\theta|\mathcal{D}} : \theta \in \Theta_n\}$.

**Lemma G.1.** *Let $\mathcal{D}$ be an arbitrary set of $m$ design points. There exists an absolute constant $C_1 > 0$ such that for any $\delta \in (0, C_1/q_n)$, the following holds*

$$\log\mathcal{N}_{[]}(\delta, \mathcal{P}(\Theta_n; \mathcal{D}), h) \leq \log\mathcal{N}\left(\frac{\sigma_*^2\delta}{26\max\{2mq_nB, \sqrt{m}\sigma_*\}}, \mathcal{G}(\Theta_n), \|\cdot\|_\infty\right).$$

**Theorem G.2.** *Suppose that $\mu_*$ and $\phi_{*,j}, j = 1, \ldots$ belong to $\mathcal{H}_d^\beta(K)$. Consider the sieve MLE $\hat{p} = p_{\hat{\theta}}$ over $\Theta_n = \Theta(L_n, r_n, S_n, q_n)$ with $L_n \asymp \log n$ and $r_n, S_n, q_n \lesssim n$. Define*

$$(\epsilon_n^*)^2 = q_n\left(\frac{S_n}{\log n}\right)^{-2\beta/d} + \sum_{j>q_n}\lambda_j^2 + S_n\frac{(\log n)^2}{n}.$$

*Assume that $q_n \to \infty$, $\epsilon_n^* q_n \to 0$ and $n(\epsilon_n^*)^2 \to \infty$ as $n \to \infty$. Then, we have*

$$P_*(h(\hat{p}, p_*) > C_2\epsilon_n^*) \to 0$$

*as $n \to \infty$ for some absolute constant $C_2 > 0$.*

We can make $\epsilon_n^*$ converge to 0 by letting $q_n$ and $S_n$ diverge with a appropriate speed provided the eigenvalues $\lambda_j, j \geq q_n$ converge to 0 sufficiently fast (e.g. $\lambda_j \asymp \exp(-j)$).

The upper bound of Theorem G.2 is about the estimated Gaussian distribution at the design points $\mathcal{D}^{\text{design}}$. For prediction, we need an upper bound of the estimated Gaussian distribution at a new point $\boldsymbol{x}$. The following theorem, whose proof is given in Appendix G.4, provides an upper bound.

**Theorem G.3** (Upper bound at a new input). *If $\hat{\mu}$ and $\hat{\Phi}_j, j = 1, \ldots, q_n$ are Lipschitz, the probability of*

$$d_1(\hat{p}_{\boldsymbol{x}}, p_{*,\boldsymbol{x}}) \leq d_1(\hat{p}_{\boldsymbol{x}_{(1)}}, p_{*,\boldsymbol{x}_{(1)}}) + C_3\|\boldsymbol{x} - \boldsymbol{x}_{(1)}\|$$

*for a certain positive constant $C_3$ and*

$$\sup_{\boldsymbol{x} \in \mathcal{D}^{\text{design}}} d_1(\hat{p}_{\boldsymbol{x}}, p_{*,\boldsymbol{x}}) \leq C_2\epsilon_n^*$$

*converges to 1 as $n \to \infty$, where $d_1(g, h) = \int_z |g(z) - h(z)|dz$ for given two probability densities on $\mathbb{R}$.*

### G.3 Auxiliary Lemmas

Before proving Lemma G.1, we introduce the following two lemmas.

**Lemma G.4.** *If* $\boldsymbol{\Sigma}_2 - \boldsymbol{\Sigma}_1$ *is positive definite, then*

$$\frac{p_{\boldsymbol{\mu}_1,\boldsymbol{\Sigma}_1}(x)}{p_{\boldsymbol{\mu}_2,\boldsymbol{\Sigma}_2}(x)} \leq \sqrt{\frac{|\boldsymbol{\Sigma}_2|}{|\boldsymbol{\Sigma}_1|}} \exp\left(\frac{1}{2}(\boldsymbol{\mu}_2 - \boldsymbol{\mu}_1)^\top (\boldsymbol{\Sigma}_2 - \boldsymbol{\Sigma}_1)^{-1}(\boldsymbol{\mu}_2 - \boldsymbol{\mu}_1)\right)$$

*Proof.* Define $\boldsymbol{\mu}_* = (\boldsymbol{\Sigma}_1^{-1} - \boldsymbol{\Sigma}_2^{-1})^{-1}(\boldsymbol{\Sigma}_1^{-1}\boldsymbol{\mu}_1 - \boldsymbol{\Sigma}_2^{-1}\boldsymbol{\mu}_2)$. Note that by assumption $\boldsymbol{\Sigma}_2 - \boldsymbol{\Sigma}_1$ is invertible, and thus we have

$$(x - \boldsymbol{\mu}_1)^\top \boldsymbol{\Sigma}_1^{-1}(x - \boldsymbol{\mu}_1) - (x - \boldsymbol{\mu}_2)^\top \boldsymbol{\Sigma}_2^{-1}(x - \boldsymbol{\mu}_2)$$
$$= (x - \boldsymbol{\mu}_*)^\top (\boldsymbol{\Sigma}_1^{-1} - \boldsymbol{\Sigma}_2^{-1})(x - \boldsymbol{\mu}_*) - \boldsymbol{\mu}_*^\top (\boldsymbol{\Sigma}_1^{-1} - \boldsymbol{\Sigma}_2^{-1})\boldsymbol{\mu}_* + \boldsymbol{\mu}_1^\top \boldsymbol{\Sigma}_1^{-1}\boldsymbol{\mu}_1 - \boldsymbol{\mu}_2^\top \boldsymbol{\Sigma}_2^{-1}\boldsymbol{\mu}_2.$$

The sum of the second and third terms is further simplified as

$$-\boldsymbol{\mu}_*^\top (\boldsymbol{\Sigma}_1^{-1} - \boldsymbol{\Sigma}_2^{-1})\boldsymbol{\mu}_* + \boldsymbol{\mu}_1^\top \boldsymbol{\Sigma}_1^{-1}\boldsymbol{\mu}_1 - \boldsymbol{\mu}_2^\top \boldsymbol{\Sigma}_2^{-1}\boldsymbol{\mu}_2$$
$$-\boldsymbol{\mu}_*^\top \left(\boldsymbol{\Sigma}_1^{-1} - \boldsymbol{\Sigma}_2^{-1}\right)\boldsymbol{\mu}_* + \boldsymbol{\mu}_1^\top \boldsymbol{\Sigma}_1^{-1}\boldsymbol{\mu}_1 - \boldsymbol{\mu}_2^\top \boldsymbol{\Sigma}_2^{-1}\boldsymbol{\mu}_2$$
$$= -\left(\boldsymbol{\Sigma}_1^{-1}\boldsymbol{\mu}_1 - \boldsymbol{\Sigma}_2^{-1}\boldsymbol{\mu}_2\right)^\top \left(\boldsymbol{\Sigma}_1^{-1} - \boldsymbol{\Sigma}_2^{-1}\right)^{-1}\left(\boldsymbol{\Sigma}_1^{-1}\boldsymbol{\mu}_1 - \boldsymbol{\Sigma}_2^{-1}\boldsymbol{\mu}_2\right) + \boldsymbol{\mu}_1^\top \boldsymbol{\Sigma}_1^{-1}\boldsymbol{\mu}_1 - \boldsymbol{\mu}_2^\top \boldsymbol{\Sigma}_2^{-1}\boldsymbol{\mu}_2$$
$$= -\boldsymbol{\mu}_1^\top \boldsymbol{\Sigma}_1^{-1}\left(\mathbf{I} - \boldsymbol{\Sigma}_1 \boldsymbol{\Sigma}_2^{-1}\right)^{-1}\boldsymbol{\mu}_1 - \boldsymbol{\mu}_2^\top \boldsymbol{\Sigma}_2^{-1}\left(\boldsymbol{\Sigma}_2 \boldsymbol{\Sigma}_1^{-1} - \mathbf{I}\right)^{-1}\boldsymbol{\mu}_2$$
$$\quad + 2\boldsymbol{\mu}_2^\top \boldsymbol{\Sigma}_2^{-1}\left(\boldsymbol{\Sigma}_1^{-1} - \boldsymbol{\Sigma}_2^{-1}\right)\boldsymbol{\Sigma}_1^{-1}\boldsymbol{\mu}_1 + \boldsymbol{\mu}_1^\top \boldsymbol{\Sigma}_1^{-1}\boldsymbol{\mu}_1 - \boldsymbol{\mu}_2^\top \boldsymbol{\Sigma}_2^{-1}\boldsymbol{\mu}_2$$
$$= -\boldsymbol{\mu}_1^\top \boldsymbol{\Sigma}_1^{-1}\boldsymbol{\Sigma}_1\boldsymbol{\Sigma}_2^{-1}\left(\mathbf{I} - \boldsymbol{\Sigma}_1 \boldsymbol{\Sigma}_2^{-1}\right)^{-1}\boldsymbol{\mu}_1 - \boldsymbol{\mu}_2^\top \boldsymbol{\Sigma}_2^{-1}\boldsymbol{\Sigma}_2\boldsymbol{\Sigma}_1^{-1}\left(\boldsymbol{\Sigma}_2 \boldsymbol{\Sigma}_1^{-1} - \mathbf{I}\right)^{-1}\boldsymbol{\mu}_2$$
$$\quad + 2\boldsymbol{\mu}_2^\top \boldsymbol{\Sigma}_2^{-1}\left(\boldsymbol{\Sigma}_1^{-1} - \boldsymbol{\Sigma}_2^{-1}\right)^{-1}\boldsymbol{\Sigma}_1^{-1}\boldsymbol{\mu}_1$$
$$= -\boldsymbol{\mu}_1^\top \left(\boldsymbol{\Sigma}_2 - \boldsymbol{\Sigma}_1\right)^{-1}\boldsymbol{\mu}_1 - \boldsymbol{\mu}_2^\top \left(\boldsymbol{\Sigma}_2 - \boldsymbol{\Sigma}_1\right)^{-1}\boldsymbol{\mu}_2 + 2\boldsymbol{\mu}_2^\top \left(\boldsymbol{\Sigma}_2^{-1} - \boldsymbol{\Sigma}_1^{-1}\right)^{-1}\boldsymbol{\mu}_1$$
$$= -(\boldsymbol{\mu}_1 - \boldsymbol{\mu}_2)^\top \left(\boldsymbol{\Sigma}_2 - \boldsymbol{\Sigma}_1\right)^{-1}(\boldsymbol{\mu}_1 - \boldsymbol{\mu}_2).$$

Therefore, we have

$$\frac{p_{\boldsymbol{\mu}_1,\boldsymbol{\Sigma}_1}(x)}{p_{\boldsymbol{\mu}_2,\boldsymbol{\Sigma}_2}(x)} = \sqrt{\frac{|\boldsymbol{\Sigma}_2|}{|\boldsymbol{\Sigma}_1|}} \exp\left(-\frac{1}{2}\{(x - \boldsymbol{\mu}_1)^\top \boldsymbol{\Sigma}_1^{-1}(x - \boldsymbol{\mu}_1) - (x - \boldsymbol{\mu}_2)^\top \boldsymbol{\Sigma}_2^{-1}(x - \boldsymbol{\mu}_2)\}\right)$$

$$= \sqrt{\frac{|\boldsymbol{\Sigma}_2|}{|\boldsymbol{\Sigma}_1|}} \exp\left(-\frac{1}{2}(x - \boldsymbol{\mu}_*)^\top (\boldsymbol{\Sigma}_1^{-1} - \boldsymbol{\Sigma}_2^{-1})(x - \boldsymbol{\mu}_*)\right.$$
$$\left. + \frac{1}{2}(\boldsymbol{\mu}_1 - \boldsymbol{\mu}_2)^\top (\boldsymbol{\Sigma}_2 - \boldsymbol{\Sigma}_1)^{-1}(\boldsymbol{\mu}_1 - \boldsymbol{\mu}_2)\right)$$

$$\leq \sqrt{\frac{|\boldsymbol{\Sigma}_2|}{|\boldsymbol{\Sigma}_1|}} \exp\left(\frac{1}{2}(\boldsymbol{\mu}_1 - \boldsymbol{\mu}_2)^\top (\boldsymbol{\Sigma}_2 - \boldsymbol{\Sigma}_1)^{-1}(\boldsymbol{\mu}_1 - \boldsymbol{\mu}_2)\right),$$

which completes the proof. $\qquad\square$

**Lemma G.5.** *Let* $\sigma^2$ *be the lower bound of the minimum eigenvalues of* $\boldsymbol{\Sigma}_1$ *and* $\boldsymbol{\Sigma}_2$. *If* $\|\boldsymbol{\Sigma}_2 - \boldsymbol{\Sigma}_1\|_2 \leq c\sigma^2$ *for a given* $c > 0$, *the following inequalities hold:*

$$x^\top ((1 + \zeta)\boldsymbol{\Sigma}_2 - \boldsymbol{\Sigma}_1)x \geq \sigma^2(\zeta - (1 + \zeta)c)\|x\|^2$$

*and*

$$x^\top (\boldsymbol{\Sigma}_1 - (1 + \zeta)^{-1}\boldsymbol{\Sigma}_2)x \geq \frac{(\zeta - c)\sigma^2}{1 + \zeta}\|x\|^2,$$

*where* $\zeta = 3\sigma^2 c$.

*Proof.* The first inequality holds because

$$x^\top ((1 + \zeta)\boldsymbol{\Sigma}_2 - \boldsymbol{\Sigma}_1)x \geq (1 + \zeta)x^\top (\boldsymbol{\Sigma}_2 - \boldsymbol{\Sigma}_1)x + \zeta x^\top \boldsymbol{\Sigma}_1 x$$
$$\geq -(1 + \zeta)\|\boldsymbol{\Sigma}_2 - \boldsymbol{\Sigma}_1\|_2\|x\|_2^2 + \zeta\sigma^2\|x\|^2$$
$$\geq (\zeta\sigma^2 - (1 + \zeta)c\sigma^2)\|x\|^2.$$

The second inequality follows similarly. $\qquad\square$

### G.4 Proofs

**Proof of Lemma G.1**

*Proof.* Fix $\epsilon > 0$. Let $\{\boldsymbol{g}_1, \ldots, \boldsymbol{g}_N\}$ with $N = \mathcal{N}(\epsilon, \mathcal{G}(\Theta_n), \|\cdot\|_\infty)$ be a $\epsilon$-covering of $\mathcal{G}(\Theta_n)$. For each $i \in [N]$, let $\theta_i$ be the parameter of $\boldsymbol{g}_i$ and let $\boldsymbol{\mu}_i = \boldsymbol{\mu}_{\theta_i|\mathcal{D}}$ and $\boldsymbol{\Sigma}_i = \boldsymbol{\Sigma}_{\theta_i|\mathcal{D}}$. Then for any $\theta \in \Theta_n$, letting $\boldsymbol{\mu} = \boldsymbol{\mu}_{\theta|\mathcal{D}}$ and $\boldsymbol{\Sigma} = \boldsymbol{\Sigma}_{\theta|\mathcal{D}}$ for simplicity, we have

$$\|\boldsymbol{\mu} - \boldsymbol{\mu}_i\|_2 \leq \sqrt{m}\epsilon,$$

and

$$\begin{aligned}
\|\boldsymbol{\Sigma} - \boldsymbol{\Sigma}_i\|_2 &\leq \|(\boldsymbol{\Phi}\boldsymbol{\Phi}^\top - \boldsymbol{\Phi}_i\boldsymbol{\Phi}_i^\top)\|_2 \\
&\leq (\|\boldsymbol{\Phi}\|_2 + \|\boldsymbol{\Phi}_i\|_2)\|\boldsymbol{\Phi} - \boldsymbol{\Phi}_i\|_F \\
&\leq 2mq_n B\epsilon,
\end{aligned}$$

where the third line follows from that $\|\boldsymbol{\Phi}\|_2, \|\boldsymbol{\Phi}_i\|_2 \leq \sqrt{mq_n}B$. Now, let $\delta = \epsilon\max\{2mq_n B, \sqrt{m}\sigma_*\}/\sigma_*^2$ so that $\|\boldsymbol{\mu} - \boldsymbol{\mu}_i\|_2 \leq \sigma_*\delta$ and $\|\boldsymbol{\Sigma} - \boldsymbol{\Sigma}_i\|_2 \leq \sigma_*^2\delta$. Let $\zeta = 3\delta$. Then we will show that $[l_i, u_i]$ is a Hellinger bracket of the density $p_{\boldsymbol{\mu},\boldsymbol{\Sigma}}(x)$ when we define

$$\begin{aligned}
u_i &= (1 + 2\zeta)^m p_{\boldsymbol{\mu}_i, (1+\zeta)\boldsymbol{\Sigma}_i} \\
l_i &= (1 + 2\zeta)^{-m} p_{\boldsymbol{\mu}_i, (1+\zeta)^{-1}\boldsymbol{\Sigma}_i},
\end{aligned}$$

Then by Lemma G.5, $(1+\zeta)\boldsymbol{\Sigma}_i - \boldsymbol{\Sigma}$ and $\boldsymbol{\Sigma} - (1+\zeta)^{-1}\boldsymbol{\Sigma}_i$ are both positive definite. So by Lemma G.4, we have for any $\boldsymbol{x} \in \mathbb{R}^m$

$$\frac{p_{\boldsymbol{\mu},\boldsymbol{\Sigma}}(\boldsymbol{x})}{u_i(\boldsymbol{x})} \leq (1 + 2\zeta)^{-m} \sqrt{\frac{|(1+\zeta)\boldsymbol{\Sigma}_i|}{|\boldsymbol{\Sigma}|}} \exp\left(\frac{1}{2}(\boldsymbol{\mu} - \boldsymbol{\mu}_i)^\top((1+\zeta)\boldsymbol{\Sigma}_i - \boldsymbol{\Sigma})^{-1}(\boldsymbol{\mu}_i - \boldsymbol{\mu})\right).$$

By Lemma G.5 again, we have $\|((1+\zeta)\boldsymbol{\Sigma}_i - \boldsymbol{\Sigma})^{-1}\|_2 \leq (\sigma_*^2(\zeta - (1+\zeta)\delta))^{-1} = (\sigma_*^2(2-\zeta)\delta)^{-1} \leq (\sigma_*^2\delta)^{-1}$ for any sufficiently small $\epsilon$. Moreover, by Weyl's inequality,

$$\frac{|(1+\zeta)\boldsymbol{\Sigma}_i|}{|\boldsymbol{\Sigma}|} \leq (1+\zeta)^m \left(1 + \frac{\sigma_*^2\delta}{\sigma_*^2}\right)^m \leq (1 + 2\zeta)^m.$$

Thus we have

$$\frac{p_{\boldsymbol{\mu},\boldsymbol{\Sigma}}(\boldsymbol{x})}{u_i(\boldsymbol{x})} = (1 + 2\zeta)^{-m/2} \exp\left(\frac{\|\boldsymbol{\mu} - \boldsymbol{\mu}_i\|_2^2}{2\sigma_*^2\delta}\right).$$

Using the inequality $\log(1 + z) \geq z/2$ for $z \in [0, 2]$, we have

$$\begin{aligned}
\log \frac{p_{\boldsymbol{\mu},\boldsymbol{\Sigma}}(\boldsymbol{x})}{u_i(\boldsymbol{x})} &= -\frac{m}{2}\log(1 + 2\zeta) + \frac{1}{2\sigma_*^2\delta}\|\boldsymbol{\mu} - \boldsymbol{\mu}_i\|_2^2 \\
&\leq -\frac{m}{2}\zeta + \frac{m}{2\sigma_*^2\delta}(\sigma_*\delta)^2 \\
&\leq \left(-\frac{3m}{2} + \frac{m}{2}\right)\delta \leq 0,
\end{aligned}$$

which implies $p_{\boldsymbol{\mu},\boldsymbol{\Sigma}}(\boldsymbol{x}) \leq u_i(\boldsymbol{x})$. Similarly, we also have

$$\frac{l_i(\boldsymbol{x})}{p_{\boldsymbol{\mu},\boldsymbol{\Sigma}}(\boldsymbol{x})} = (1 + 2\zeta)^{-m/2} \exp\left(\frac{\|\boldsymbol{\mu} - \boldsymbol{\mu}_i\|_2^2}{2\sigma_*^2\delta}\right).$$

and so $l_i(\boldsymbol{x}) \leq p_{\boldsymbol{\mu},\boldsymbol{\Sigma}}(\boldsymbol{x})$ for any $\boldsymbol{x} \in \mathbb{R}^m$.

We now bound the size of the bracket. Note that

$$h^2(l_i, u_i) = (1 + 2\zeta)^m + (1 + 2\zeta)^{-m} - (2 - h^2(p_{\boldsymbol{\mu}_i, (1+\zeta)\boldsymbol{\Sigma}_i}, p_{\boldsymbol{\mu}_i, (1+\zeta)^{-1}\boldsymbol{\Sigma}_i})).$$

Due to the inequality $z^2/2 \geq z - \log(1+z)$ for any $z \geq 0$,

$$
\begin{aligned}
h^2(p_{\boldsymbol{\mu_i},(1+\zeta)\boldsymbol{\Sigma}_i}, p_{\boldsymbol{\mu_i},(1+\zeta)^{-1}\boldsymbol{\Sigma}_i}) &\leq \mathrm{KL}(p_{\boldsymbol{\mu_i},(1+\zeta)\boldsymbol{\Sigma}_i}, p_{\boldsymbol{\mu_i},(1+\zeta)^{-1}\boldsymbol{\Sigma}_i}) \\
&= \frac{1}{4}m(-\log(1+\zeta)^2 + (1+\zeta)^2 - 1) \\
&\leq \frac{m}{8}((1+\zeta)^2 - 1)^2 \\
&= \frac{m}{4}(\zeta + \zeta^2/2)^2 \\
&\leq \frac{9}{4}m\zeta^2
\end{aligned}
$$

Moreover, by taking $\epsilon$ sufficiently small so that $\zeta < 3/m$, we have

$$
\begin{aligned}
(1+2\zeta)^m + (1+2\zeta)^{-m} - 2 &\leq 2(1+2\zeta)^m - 2 \\
&\leq 4m\zeta(1+2\zeta)^{m-1} \\
&\leq 4m\zeta(1+2/(3m))^{m-1} \leq 4e^{2/3}\zeta
\end{aligned}
$$

Thus, we have $h^2(l_i, u_i) \leq (9/4m\zeta + 4e^{2/3})\zeta \leq (3/4 + 4e^{2/3})\zeta \leq 26\delta$. Hence, redefining constant as $26\delta \to \delta$, we complete the proof. $\qquad\square$

**Proof of Theorem G.2**

*Proof.* The proof follows a similar reasoning in the proof of Theorem 3 in [62], which is based on Theorem 4 in [64] with $\alpha = 0+$. We divide the proof into the following four steps.

**Bounding the estimation error: Check Eq. (3.1) of [64]** For the class of DNN parameters $\Theta_n = \Theta(L_n, r_n, S_n, q_n)$, by Lemma 5 in [63], we can get the following covering number bound

$$
\begin{aligned}
\log \mathcal{N}(\delta, \mathcal{G}(\Theta_n), \|\cdot\|_\infty) &\leq (S_n + 1)\log\left(2\delta^{-1}(L_n+1)d^2(q_n+1)^2 r_n^{2L_n}\right) \\
&\lesssim L_n S_n \log(n\delta^{-1})
\end{aligned}
$$

for any $\delta > 0$. Applying Lemma G.1, for $0 < \delta < C_1/q_n$, we have

$$
\begin{aligned}
\log \mathcal{N}_{[]}(\delta, \mathcal{P}(\Theta_n; \mathcal{D}), h) &\leq \log \mathcal{N}\left(\frac{\sigma_*^2 \delta}{26\max\{2mq_n B, \sqrt{m}\sigma_*\}}, \mathcal{G}(\Theta_n), \|\cdot\|_\infty\right) \\
&\lesssim S_n L_n \log(n\delta^{-1})
\end{aligned}
$$

Moreover, for a positive constant $\epsilon$ such that $\sqrt{2}\epsilon \leq C_1/q_n$, we have

$$
\int_{\epsilon^2/2^8}^{\sqrt{2}\epsilon} \sqrt{\log \mathcal{N}_{[]}(\delta, \mathcal{P}(\Theta_n; \mathcal{D}), h)}d\delta \lesssim \epsilon\sqrt{S_n L_n \log(n\epsilon^{-1})}.
$$

Then the above display is bounded by $n^{1/2}\epsilon^2$ up to an absolute constant when we take $\epsilon = \epsilon_n = \sqrt{S_n(\log n)^2/n}$ as $L_n \asymp \log n$. Thus, Eq. (3.1) of [64] is satisfied.

**Bounding the Kullback-Liebler approximation error** We first note that for any $\theta \in \Theta_n$, we have

$$
\begin{aligned}
\mathrm{KL}\left(p_* \| p_{\boldsymbol{\mu}_{\theta|\mathcal{D}}, \boldsymbol{\Sigma}_{\theta|\mathcal{D}}}\right) &= \frac{1}{2}\left(-\log\left|\boldsymbol{\Sigma}_{\theta|\mathcal{D}}\boldsymbol{\Sigma}_{*|\mathcal{D}}^{-1}\right| + \mathrm{Tr}\left(\boldsymbol{\Sigma}_{\theta|\mathcal{D}}\boldsymbol{\Sigma}_{*|\mathcal{D}}^{-1} - \mathbb{I}_m\right)\right) \\
&\quad + \frac{1}{2}\left(\boldsymbol{\mu}_{\theta|\mathcal{D}} - \boldsymbol{\mu}_{*|\mathcal{D}}\right)^\top \boldsymbol{\Sigma}_{*|\mathcal{D}}^{-1}\left(\boldsymbol{\mu}_{\theta|\mathcal{D}} - \boldsymbol{\mu}_{*|\mathcal{D}}\right) \\
&= -\frac{1}{2}\log\left|\mathbb{I}_m + \boldsymbol{\Sigma}_{*|\mathcal{D}}^{-1/2}(\boldsymbol{\Sigma}_{\theta|\mathcal{D}} - \boldsymbol{\Sigma}_{*|\mathcal{D}})\boldsymbol{\Sigma}_{*|\mathcal{D}}^{-1/2}\right| \\
&\quad + \frac{1}{2}\mathrm{Tr}\left(\boldsymbol{\Sigma}_{*|\mathcal{D}}^{-1/2}(\boldsymbol{\Sigma}_{\theta|\mathcal{D}} - \boldsymbol{\Sigma}_{*|\mathcal{D}})\boldsymbol{\Sigma}_{*|\mathcal{D}}^{-1/2}\right) + \frac{1}{2\sigma_*^2}\|\boldsymbol{\mu}_{\theta|\mathcal{D}} - \boldsymbol{\mu}_{*|\mathcal{D}}\|^2 \\
&\leq \frac{1}{4}\|\mathbb{I}_m - \boldsymbol{\Sigma}_{\theta|\mathcal{D}}\boldsymbol{\Sigma}_{*|\mathcal{D}}^{-1}\|_F^2 + \frac{1}{2\sigma_*^2}\|\boldsymbol{\mu}_{\theta|\mathcal{D}} - \boldsymbol{\mu}_{*|\mathcal{D}}\|^2, \\
&\leq \frac{1}{4\sigma_*^2}\|\boldsymbol{\Sigma}_{\theta|\mathcal{D}} - \boldsymbol{\Sigma}_{*|\mathcal{D}}\|_F^2 + \frac{1}{2\sigma_*^2}\|\boldsymbol{\mu}_{\theta|\mathcal{D}} - \boldsymbol{\mu}_{*|\mathcal{D}}\|^2, \quad (6)
\end{aligned}
$$

To bound the two terms in Eq. (6), we use the well-known results about the approximation ability of sparse DNNs to Hölder smooth functions [e.g., Theorem 5 of 63]. Namely, there exists $\theta^\dagger \in \Theta_n$ such that

$$\max\left\{\|g_{\theta^\dagger,1} - \mu_*\|, \max_{j\in[q_n]}\|g_{\theta^\dagger,j+1} - \phi_{*,j}\|_\infty\right\} \lesssim (S_n/L_n)^{-\beta/d}. \tag{7}$$

For the first term in Eq. (6), we define

$$\kappa_n = \|\mathbf{\Sigma}_{*>q_n|\mathcal{D}}\|_F = \left(\sum_{j>q_n}\lambda_j^2\right)^{1/2}$$

and establish the upper bound

$$\|\mathbf{\Sigma}_{\theta^\dagger|\mathcal{D}} - \mathbf{\Sigma}_{*|\mathcal{D}}\|_F \leq \|\mathbf{\Phi}_{\theta^\dagger|\mathcal{D}}\mathbf{\Phi}_{\theta^\dagger|\mathcal{D}}^\top - \mathbf{\Phi}_{*|\mathcal{D}}\mathbf{\Phi}_{*|\mathcal{D}}^\top\|_F + \kappa_n$$
$$\leq \|(\mathbf{\Phi}_{\theta^\dagger|\mathcal{D}} - \mathbf{\Phi}_{*|\mathcal{D}})(\mathbf{\Phi}_{\theta^\dagger|\mathcal{D}} - \mathbf{\Phi}_{*|\mathcal{D}})^\top + 2\mathbf{\Phi}_{*|\mathcal{D}}(\mathbf{\Phi}_{\theta^\dagger|\mathcal{D}} - \mathbf{\Phi}_{*|\mathcal{D}})^\top\|_F + \kappa_n$$
$$\leq \|\mathbf{\Phi}_{\theta^\dagger|\mathcal{D}} - \mathbf{\Phi}_{*|\mathcal{D}}\|_F^2 + 2\|\mathbf{\Phi}_{*|\mathcal{D}}\|_2\|\mathbf{\Phi}_{\theta^\dagger|\mathcal{D}} - \mathbf{\Phi}_{*|\mathcal{D}}\|_F + \kappa_n.$$

But by Eq. (7), $\|\mathbf{\Phi}_{\theta^\dagger|\mathcal{D}} - \mathbf{\Phi}_{*|\mathcal{D}}\|_F \lesssim \sqrt{q_n}(S_n/L_n)^{-\beta/d}$ which converges to 0 as $n \to \infty$. This implies that, as $\|\mathbf{\Phi}_{*|\mathcal{D}}\|_2$ is bounded, $\|\mathbf{\Phi}_{\theta^\dagger|\mathcal{D}} - \mathbf{\Phi}_{*|\mathcal{D}}\|_F^2$ is smaller than $2\|\mathbf{\Phi}_{*|\mathcal{D}}\|_2\|\mathbf{\Phi}_{\theta^\dagger|\mathcal{D}} - \mathbf{\Phi}_{*|\mathcal{D}}\|_F$ eventually. Therefore, we have

$$\|\mathbf{\Sigma}_{\theta^\dagger|\mathcal{D}} - \mathbf{\Sigma}_{*|\mathcal{D}}\|_F^2 \lesssim q_n(S_n/L_n)^{-2\beta/d} + \kappa_n$$

Moreover, by Eq. (7), it is immediate that $\|\boldsymbol{\mu}_{\theta^\dagger|\mathcal{D}} - \boldsymbol{\mu}_{*|\mathcal{D}}\|^2 \lesssim (S_n/L_n)^{-2\beta/d}$. Adopting the notation of [64], we have $\delta_n = q_n(S_n/L_n)^{-2\beta/d} + \kappa_n \asymp q_n(S_n/\log n)^{-2\beta/d} + \kappa_n$.

**Bounding the Kullback-Liebler variation** The last ingredient of the proof is to bound the so-called Kullback-Liebler variation defined as

$$\text{KLV}\left(p_* \| p_{\boldsymbol{\mu}_{\theta|\mathcal{D}}, \mathbf{\Sigma}_{\theta|\mathcal{D}}}\right) = \int \left\{\log \frac{p_*(\boldsymbol{x})}{p_{\boldsymbol{\mu}_{\theta|\mathcal{D}}, \mathbf{\Sigma}_{\theta|\mathcal{D}}}(\boldsymbol{x})}\right\}^2 p_*(\boldsymbol{x})d\boldsymbol{x}.$$

We will find a suitable network parameter to get a manageable upper bound of the above, which is denoted by $\tau_n$ in [64]. We use Lemma G.5 for this purpose. As in the argument used in the previous step, we can find a network parameter $\theta^\dagger$ such that $\|\mathbf{\Sigma}_{\theta^\dagger|\mathcal{D}} - \mathbf{\Sigma}_{*|\mathcal{D}}\|_F^2 \leq C'\delta_n$ for some absolute constant $C' > 0$. As $q_n \to \infty$, we have $\xi = \lambda_{\min}(\mathbf{\Phi}_{\theta^\dagger|\mathcal{D}}\mathbf{\Phi}_{\theta^\dagger|\mathcal{D}}^\top) > 0$. We then construct the network $\theta^\ddagger$ satisfying $\boldsymbol{\mu}_{\theta^\ddagger|\mathcal{D}} = \boldsymbol{\mu}_{\theta^\dagger|\mathcal{D}}$ and $\mathbf{\Phi}_{\theta^\ddagger|\mathcal{D}} = (1 + (1 + C')\delta_n/\xi)^{1/2}\mathbf{\Phi}_{\theta^\dagger|\mathcal{D}}$. Then, by Weyl's inequality,

$$\lambda_{\min}(\mathbf{\Sigma}_{\theta^\ddagger|\mathcal{D}} - \mathbf{\Sigma}_{*|\mathcal{D}}) = \lambda_{\min}(\mathbf{\Sigma}_{\theta^\dagger|\mathcal{D}} - \mathbf{\Sigma}_{*|\mathcal{D}} + (1 + C')\delta_n/\xi\mathbf{\Phi}_{\theta^\dagger|\mathcal{D}}\mathbf{\Phi}_{\theta^\dagger|\mathcal{D}}^\top)$$
$$\geq \lambda_{\min}((1 + C')\delta_n/\xi\mathbf{\Phi}_{\theta^\dagger|\mathcal{D}}\mathbf{\Phi}_{\theta^\dagger|\mathcal{D}}^\top) - \|\mathbf{\Sigma}_{\theta^\dagger|\mathcal{D}} - \mathbf{\Sigma}_{*|\mathcal{D}}\|_2$$
$$\geq (1 + C')\delta_n - \|\mathbf{\Sigma}_{\theta^\dagger|\mathcal{D}} - \mathbf{\Sigma}_{*|\mathcal{D}}\|_F \geq \delta_n,$$

which implies that $\mathbf{\Sigma}_{\theta^\ddagger|\mathcal{D}} - \mathbf{\Sigma}_{*|\mathcal{D}}$ is positive definite. Using Lemma G.5, we have

$$\frac{p_*(x)}{p_{\boldsymbol{\mu}_{\theta^\ddagger|\mathcal{D}}, \mathbf{\Sigma}_{\theta^\ddagger|\mathcal{D}}}(x)} \leq \sqrt{\frac{|\mathbf{\Sigma}_{\theta^\ddagger|\mathcal{D}}|}{|\mathbf{\Sigma}_{*|\mathcal{D}}|}} \exp\left(\frac{1}{2}(\boldsymbol{\mu}_{\theta^\ddagger|\mathcal{D}} - \boldsymbol{\mu}_{*|\mathcal{D}})^\top(\mathbf{\Sigma}_{\theta^\ddagger|\mathcal{D}} - \mathbf{\Sigma}_{*|\mathcal{D}})^{-1}(\boldsymbol{\mu}_{\theta^\ddagger|\mathcal{D}} - \boldsymbol{\mu}_{*|\mathcal{D}})\right)$$
$$\leq \sqrt{\frac{|\mathbf{\Sigma}_{\theta^\ddagger|\mathcal{D}}|}{|\mathbf{\Sigma}_{*|\mathcal{D}}|}} \exp\left(\frac{1}{2\delta_n}\|\boldsymbol{\mu}_{\theta^\ddagger|\mathcal{D}} - \boldsymbol{\mu}_{*|\mathcal{D}}\|^2\right)$$
$$\leq (1 + \|\mathbf{\Sigma}_{\theta^\ddagger|\mathcal{D}} - \mathbf{\Sigma}_{*|\mathcal{D}}\|_2/\sigma_*^2)^{m/2}$$
$$\leq (1 + \|(1 + C')\delta_n/(\xi\sigma_*^2)\mathbf{\Phi}_{\theta^\dagger|\mathcal{D}}\mathbf{\Phi}_{\theta^\dagger|\mathcal{D}}^\top\|_F + \|\mathbf{\Sigma}_{\theta^\dagger|\mathcal{D}} - \mathbf{\Sigma}_{*|\mathcal{D}}\|_2/\sigma_*^2)^{m/2}$$
$$\leq (1 + (1 + C')(Bmq_n)^{1/2}\delta_n/(\xi\sigma_*^2) + (C')^{1/2}\delta_n/\sigma_*^2)^{m/2}$$

where we use Weyl's inequality for the third inequality. Therefore, we have

$$\text{KLV}\left(p_* \| p_{\boldsymbol{\mu}_{\theta^\ddagger|\mathcal{D}}, \mathbf{\Sigma}_{\theta^\ddagger|\mathcal{D}}}\right) \lesssim \log(1 + (q_n)^{1/2}\delta_n).$$

Adopting the notation of [64], we set $\tau_n = \log(1 + (q_n)^{1/2}\delta_n)$.

**Combining the pieces together** Let $\epsilon_n^* = \epsilon_n \vee \sqrt{\delta_n}$. Then by Theorem 4 of [64], there exists an absolute constant $C'' > 0$ such that

$$P_* \left( h\left(\hat{p}, p_*\right) > C_2 \epsilon_n^* \right) \lesssim e^{-C''n(\epsilon_n^*)^2} + \frac{\tau_n}{n(\epsilon_n^*)^2},$$

which tends to zero as $n \to \infty$ by the assumptions $\epsilon_n^* q_n \to 0$ and $n(\epsilon_n^*)^2 \to \infty$. $\qquad\square$

**Proof of Theorem G.3**

*Proof.* Since the total variation norm is upper bounded by the Hellinger distance, the total variation norm between $\hat{p}$ and $p_*$ is also upper bounded by $C_2\epsilon_*$ with probability converging to 1. In turn, by the definition of the total variation norm, $\sup_{\boldsymbol{x}\in\mathcal{D}^{\mathrm{design}}} d_1(\hat{p}_{\boldsymbol{x}}, p_{*,\boldsymbol{x}})$ is upper bounded by $C_2\epsilon_*$ with probability converging to 1. Due to the Lipschitz condition of $\hat{\mu}, \hat{\Sigma}$ as well as $\mu_*, \Sigma_*$, there exists a constant $L > 0$ such that $d_1(\hat{p}_{\boldsymbol{x}}, \hat{p}_{\boldsymbol{x}'}) \leq L\|\boldsymbol{x} - \boldsymbol{x}'\|$ and $d_1(p_{*,\boldsymbol{x}}, p_{*,\boldsymbol{x}'}) \leq L\|\boldsymbol{x} - \boldsymbol{x}'\|$ for any $\boldsymbol{x}$ and $\boldsymbol{x}'$ in $\mathbb{R}^d$. Finally, we have

$$
\begin{aligned}
d_1(\hat{p}_{\boldsymbol{x}}, p_{*,\boldsymbol{x}}) &\leq& d_1(\hat{p}_{\boldsymbol{x}}, \hat{p}_{\boldsymbol{x}_{(1)}}) + d_1(\hat{p}_{\boldsymbol{x}_{(1)}}, p_{*,\boldsymbol{x}_{(1)}}) + d_1(p_{*,\boldsymbol{x}_{(1)}}, p_{*,\boldsymbol{x}}) \\
&\leq& d_1(\hat{p}_{\boldsymbol{x}_{(1)}}, p_{*,\boldsymbol{x}_{(1)}}) + 2L\|\boldsymbol{x} - \boldsymbol{x}_{(1)}\| \\
&\leq& C_2\epsilon_n^* + 2L\|\boldsymbol{x} - \boldsymbol{x}_{(1)}\|
\end{aligned}
$$

with probability converging to 1. The proof is complete by letting $C_3 = C_2$ and $C_4 = 2L$.

$\qquad\square$

