# OpenReview forum: "Knowledge Distillation of Uncertainty using Deep Latent Factor Model"
_NeurIPS.cc/2025/Conference — NeurIPS 2025 poster_

### Official Review · Reviewer_jk47 · 2025-06-30

**Clarity:** 2
**Significance:** 2
**Originality:** 2
**Rating:** 4
**Confidence:** 4

**Summary:**

In this work, the authors introduce a new distribution distillation framework to estimate the distribution of a teacher ensemble through a special Gaussian process. In this distillation process, each member of the teacher ensemble is considered as a realization of a certain stochastic process. Multiple tasks, including regression tasks, classification, as well as fine-tuning of LLMs, among others, are conducted for experimental validation.

**Questions:**

**Q1**

Could the authors provide some theoretical justifications or reasons for assuming that the many teacher models are independent realizations of the DLF model, and to estimate the mean and factor loading functions? In the settings of deep ensemble, the teachers (ensemble members) are trained using the same data but with different random initialization. How could we prove these modes are independent?

**Q2**

Would the DLF model perform better than the method in [1], which achieves comparable or even better uncertainty quantification performance of deep ensembles but requires a single model?

[1] Mukhoti, Jishnu, et al. "Deep deterministic uncertainty: A new simple baseline." CVPR. 2023.

**Ethical Concerns:**

["NO or VERY MINOR ethics concerns only"]

**Final Justification:**

The responses from the authors and the new results conducted have basically addressed the concerns and enhanced my confidence in my assessments. Therefore, I increased my scores and confidence to 4. The main reasons for not further increasing my score are: There is no theoretical analysis/certificate for the main assumption in the proposed method, namely, each teacher model is a Gaussian process, and thus the prediction values from the same teacher model are dependent. The empirical evaluations do show the improved performance; I believe further experiments on larger-scale datasets (such as ImageNet)  would enhance the strengths of the proposed method.

**Limitations:**

Yes, the authors discussed them as future work.

**Paper Formatting Concerns:**

No major formatting issues are observed.

**Quality:**

2

**Strengths And Weaknesses:**

**Strengths**

1. The paper has a relatively clear structure and is easy to follow.

2. The experimental validations are clearly described, and the implementation details are given.

3. The main results shown in the main body demonstrated the outperformance of the proposed method on the applied evaluation metrics.

4. Some ablation studies are given to enhance the evaluation further.

**Weaknesses**

1. *Reliable or improved uncertainty quantification (UQ)* is claimed as one of the highlights of this work. So it would be better to briefly introduce how the prediction uncertainties are measured in the framework (especially, the authors mentioned about aleatory and epistemic uncertainty in the main body). Most importantly, the evaluation of UQ can hardly reflect the state-of-the-art benchmarks. For instance, out-of-distribution detection and active learning can be used for proxy tasks, as shown in recent literature, such as [1] [2] [3].

[1] Mukhoti, Jishnu, et al. "Deep deterministic uncertainty: A new simple baseline." CVPR. 2023.;
[2] Wang, Kaizheng, et al. "Credal deep ensembles for uncertainty quantification."NeurIPS 2024;
[3] Mucsányi, Bálint, et al. "Benchmarking uncertainty disentanglement: Specialized uncertainties for specialized tasks." NeurIPS 2024.

2. I fully agree that Dirichlet distillation is limited to classification tasks. As shown in their original paper, they achieve comparable performance to the ensemble on ECE metrics and other UQ evaluations. Clearly, they also have a simpler architecture, the same as a single ensemble member of deep ensembles. So, would it be possible to also report the results of Dirichlet distillation in the classification tasks?

3. It is true that the proposed methods significantly reduce the complexity of deep ensembles. I think it would be beneficial to include the training complexity and the inference complexity comparisons of the proposed method and the other baselines so that the community will have a better sense of applying the method in practice.

4. I appreciate the ablation study of the latent factor dimension using the Boston housing data in regression. Would the conclusion hold also for more complex classification tasks? It might also be beneficial to report the applied latent factor dimension in the main result tables, if applicable.

---

> ### Author Rebuttal · Authors · 2025-07-31
>
> Thank you for your constructive review and for expressing strong interest in our work. We are encouraged by your willingness to engage further in discussion and truly appreciate your thoughtful comments on the core contributions of our paper.
>
> # Weakness
>
> ## 1.
> As explained in the Introduction, the predictive distribution is obtained by $\hat{p}(y|\mathbf{x})=\sum_{i=1}^n p(y|\mathbf{x},\theta_i)/n,$ where $\theta_1,\ldots,\theta_n$ are ensemble members. For DLF, as shown in Figure 1 (“Inference on new data”), we can generate $\theta_1,\ldots,\theta_n$ by  sampling $n$ independent latent vectors  $\mathbf{z}_1,\ldots,\mathbf{z}_n$ from $\mathcal N_q(\mathbf 0,\mathbf I_q)$, forwarding each sample through the network to have ensemble members.  All of the uncertainty measures are calculated based on $\hat{p}(y|\mathbf{x}).$
>
> As the referee suggested, we evaluated DLF and Deep Deterministic Uncertainty (DDU) under the same OOD-detection protocol. CIFAR-10 serves as the in-distribution dataset, while SVHN, CIFAR-100, and Tiny-ImageNet are treated as OOD. AUROC is the evaluation metric, and all tests use a Wide-ResNet-16-1 backbone (≈ 0.17 M parameters).
>
> In (Mukhoti et al,. 2023), the authors report results with a much larger backbone (Wide-ResNet-28-10, $\geq$ 214 $\times$ Wide-ResNet-16-1). After downsizing DDU to match our architecture, DLF achieves higher AUROC on all three OOD datasets. **These findings demonstrate that DLF achieves well uncertainty estimation performance relative to its compact model size, which aligns well with the objective of knowledge distillation**.
>
> |method|# of parameters|SVHN|CIFAR-100|Tiny-ImageNet|
> |-|-|-|-|-|
> |small-Ens|0.70M|0.9040|0.7991|0.7952|
> |Hydra|0.18M|0.7331|0.6767|0.6794|
> |LBE|0.18M|0.8570|0.8033|0.8068|
> |DDU|0.18M|0.8772|0.6901|0.6660|
> |DLF|0.18M|0.8984|0.7897|0.7853|
>
> We evaluated OOD detection performance using AUROC as our primary metric. Although LBE achieves slightly higher AUROC scores on CIFAR-100 and Tiny-ImageNet, its predictive probability densities assign nearly many in-distribution samples a probability very close to 1.0. This behavior indicates that the model is essentially overconfident rather than genuinely better at quantifying uncertainty.
>
> ## 2.
> In the original manuscript, we did not include Dirichlet distillation (Proxy-EnD²) in the experiments since the papers for Hydra and BE reported that their algorithms are superior to Dirichlet distillation. We focused mainly on comparing these two deterministic distillation methods with our DLF.
>
> In Table below, we add Dirichlet distillation. In addition, we add a new distribution distillation method called Ensemble Distillation via Flow Matching (EDFM; Park et al., 2025), which we found after we submitted our paper. Note that DLF outperforms Proxy-EnD² and EDFM with large margins in terms of the accuracy and NLL while DLF is not much worse than Proxy-EnD² in terms of ECE.
>
> |dataset|method|Acc(%) ↑|NLL ↓|ECE(%) ↓|
> |--|--|--|--|--|
> |CIFAR-10|Teachers|94.24|0.1539|0.9|
> ||small-Ens|92.87|0.2377|3.93|
> ||Hydra|93.16|0.2660|4.15|
> ||LBE|93.25|0.2480|4.11|
> ||Proxy-EnD²|90.92|0.2861|2.08|
> ||EDFM|90.62|0.2858|2.78|
> ||DLF (Ours)|93.40|0.2246|2.79|
> |--|--|--|--|--|
> |CIFAR-100|Teachers|81.36|0.7167|1.41|
> ||small-Ens|79.29|1.0413|12.90|
> ||Hydra|77.42|1.2912|12.70|
> ||LBE|79.58|1.0110|13.42|
> ||Proxy-EnD²|67.62|1.2355|7.35|
> ||EDFM|64.17|1.6741|11.35|
> ||DLF (Ours)|79.68|0.8974|9.45|
>
> We want to emphasize that superior performance of DLF compared to other distributional distillations  is not achieved by chance. Instead, there are various advantages of DLF compared to Proxy-EnD² and EDFM. We think that there are three properties thay any desirable models for distribution distillation are expected to have: (1) large support, (2) easy and stable learning algorithm and (3) flexibility to be applied to downstream tasks. Below, we explain why DLF is desirable in view of these three properties.
>
> - DLF is a Gaussian process, which is well known to cover large class of distributions (Garriga-Alonso et al., 2019; Neal et al., 1996; Calandra et al., 2016; Huang et al., 2015). Even if teacher ensembles are not Gaussian, a Gaussian process can approximate them closely.
> - Existence of easy and stable learning algorithms is a critical fact when choosing an algorithm for distributional distillation. Learning the DLF model is numerically easy and stable since the efficient EM algorithm is available. It is well known that estimating the parameters in the Dirichlet distribution is difficult (Minka et al., 2000; Chai et al., 2013) and thus it would not be easy to learn the Dirichlet distillation model which would be a partial reason for inferior performance. For Flow matching distillation, pretrained features should be fed when learning. That is, end-to-end learning is not possible. In contrast, DLF is learned in an end-to-end fashion (i.e. feature vectors and distribution are learned simultaneously).
> -  Flexibility is also an important aspect. As we have shown in Section **5.3**, the learned features by DLF are superior than those by a single DNN or Hydra for downstream tasks.
> - To sum up, the idea of estimating the distribution for teacher ensembles is not new but efficient implementation is not an easy task and DLF is such a method.
>
> ## 3.
> Thank you for this valuable suggestion. We categorize computational cost into two aspects: the cost incurred during training steps and that during inference. Since inference cost is roughly proportional to the number of parameters, we have reported each model's parameter count. Moreover, during training, the E-step expectation can be obtained in closed form; the formula is given in Appendix B of the paper, so training time does not increase significantly. In the revised manuscript, we will include a discussion of these computational cost. Specifically, we will report (1) Multiply–Accumulate Operations (MAC, measured in MMac), (2) inference latency (ms), and (3) training time (hours).
>
> Preliminary measurements on CIFAR-10/100 are provided below.
>
> |Method|CIFAR-10 MAC|CIFAR-10 Latency|CIFAR-10 Training Time|CIFAR-100 MAC|CIFAR-100 Latency|CIFAR-100 Training Time|
> |-|-|-|-|-|-|-|
> |Teacher|224|7.1|10.2|3400|7.6|17.4|
> |small-Ens|108|4.1|4.6|224|7.3|12.5|
> |Hydra|27|1.1|3.1|56|1.8|10.28|
> |LBE|108|2.4|6.5|224|4.3|21.5|
> |Proxy-EnD²|27|1.2|4.6|56|2.1|9.2|
> |DLF (Ours)|27|1.5|4.2|56|2.6|11.28|
>
> ## 4.
> Thank you for your insightful suggestion. To verify that our observations generalize to more complex classification tasks, we have conducted an ablation study of the latent factor dimension on the CIFAR-10 and CIFAR-100 datasets. In the revised manuscript, we will include the corresponding figures, and we have also updated the main result tables to report the applied latent factor dimensions where applicable.
>
> # Question
> ## 1.
> We also think that ensemble members trained on the same data with different initializations are not strictly independent. The goal of distribution distillation is to find a proxy distribution that approximates
> the distribution of teacher ensembles, and the proxy distribution assumes that teacher ensembles are independent.
> The validity of this assumption in the proxy distribution is justified by the performance of the corresponding
> predictive distribution.
>
> Note that most methods for distribution distillation assume that teacher ensembles are independent.
> Moreover, Dirichlet distillation as well as Ensemble Distillation via Flow Matching (EDFM; Park et al. 2025)
> assume further that prediction values from a same teacher model at different inputs are independent.
> In contrast, we assume that each teacher model is a Gaussian process and thus the prediction values from a same
> teacher model are dependent.
>
> That is, DLF estimates the joint distribution of teacher ensembles while Dirichlet distillation and EDFM estimate the marginal distribution only. Estimation of the joint distribution would be useful when we have to predict multiple inputs simultaneously.
>
> For example, consider a problem to decide whether two randomly given images belong to a same class or not. Let $\mathbf{x}_1$ and $\mathbf{x}_2$ be two given images and let $Y_1$ and $Y_2$ be the corresponding class labels.
> Then, the goal is to estimate $p(Y_1=Y_2|\mathbf{x}_1,\mathbf{x}_2)$.
>
> DLF estimates this predictive probability by $\int_f \sum_{c=1}^C p(y_1=c|\mathbf{x}_1,f)p(y_2=c|\mathbf{x}_2,f) \pi(f)df$ while Dirichlet distillation and EDFM estimate it by $\sum^C_c \int_f p(y_1=c|\mathbf{x}_1,f) \pi(f)df \int_f p(y_2=c|\mathbf{x}_2,f) \pi(f)df$. These two estimates are quite different and the former is expected to perform better.
>
> To justify this conjecture, we investigate the performance of these two estimates on randomly selected pairs of CIFAR-10 test images. The results are as follows:
>
> |Method|AUROC|AUPRC|
> |-------------------------|--------|-------|
> |Dirichlet distribution|0.967|0.816|
> |DLF with dependency|0.985|0.900|
> |DLF without dependency|0.938|0.781|
>
> (AUROC and AUPRC) values are (0.967, 0.816) for Dirichlet distribution and (0.985, 0.90) for DLF while the ECE values are similar (0.11). Moreover, AUROC and AUPRC values of DLF without considering the dependency between $Y_1$ and $Y_2$ (i.e. using $\sum_{c=1}^C \int_f p(y_1=c|\mathbf{x}_1,f)\pi(f)df \int_f p(y_2=c|\mathbf{x}_2,f) \pi(f)df$ where $\pi(f)$ is estimated by DLF) are (0.938, 0.781), which are much lower than those of DLF when considering the dependency. These results amply support that considering dependency of prediction values from the same teacher model would be beneficial and DLF is such a model.
> ## 2.
> Please refer to our response to Weakness 1.
>
> # Reference
> - Mukhoti, J., Kirsch, A., Van Amersfoort, J., Torr, P. H., \& Gal, Y. (2023). Deep deterministic uncertainty: A new simple baseline.
> - Please see the reference list of Reviewer e6dg.

---

> > ### Comment · Reviewer_jk47 · 2025-08-03
> > **Response to the authors**
> >
> > Dear authors,
> >
> > First of all, thank you for your detailed rebuttals. Most of my comments are addressed. I will reflect it in my scoring. There are some remaining comments/questions:
> >
> > In the reported comparisons regarding Dirichlet distillation, what kind of uncertainty estimates are used for OOD detection? Epistemic or total uncertainty estimates? An additional question is how your proposed method could distinguish between these two uncertainties?
> >
> > How could we prove these modes are independent? This seems not to be addressed.
> >
> > I also skimmed your other responses (*We will conduct further experiments on larger-scale datasets (such as ImageNet) during the revision phase and include those results in the updated manuscript.*). I also believe that if your methods could empirically show further improvements would enhance the strengths of the proposed method.
> >
> > Best regards,
> >
> > the reviewer

---

> > > ### Author Response · Authors · 2025-08-05
> > > **Response to the Reviewer jk47**
> > >
> > > Thank you for your attention to our paper and for your insightful follow-up questions.
> > >
> > > ## Q1.
> > > We use the predictive entropy $\mathbb{H}[y \mid \mathbf{x}]  \approx -\sum_{y}\hat{p}(y \mid \mathbf{x})\log \hat{p}(y \mid \mathbf{x})$  as a total uncertainty estimates for OOD detection. Samples with larger values of  predictive entropy are classified as OOD, while those with smaller values are treated as in-distribution. Rather than fixing any threshold, we quantify detection performance by measuring the area under the ROC curve (AUROC), which captures how well this uncertainty measure separates in-distribution from out-of-distribution samples across all possible thresholds.
> > > To distinguish epistemic and aleatoric uncertainty we follow decomposition in (Gal, 2016). From this, aleatoric uncertainty can be estimated by expected entropy
> > >
> > > $$
> > > \mathbb{E}\_{\theta}\Bigl[\mathbb{H}\bigl(y\mid \mathbf{x},\theta\bigr)\Bigr]
> > > \approx -\frac{1}{n}\sum_{i=1}^{n}\sum_{y}
> > >   \hat p\bigl(y\mid \mathbf{x},\theta_i\bigr)
> > >   \log\hat p\bigl(y\mid \mathbf{x},\theta_i\bigr).
> > > $$
> > > Epistemic uncertainty is then given by the difference between the total predictive entropy and this expected entropy.
> > >
> > > ## Q2.
> > > Distribution distillation aims to find a proxy distribution that approximates the distribution of teacher ensembles.
> > > And, typically, the proxy distribution treats each teacher model as independent for technical simplicity.
> > > The feasibility of the independent assumption could not be proved but
> > > can be justified by the predictive performance of the proxy distribution.
> > > It would be helpful to construct a proxy distribution which treats teacher ensemble dependent for distribution distillation, but it would not be easy and so we want leave it as a future work.
> > >
> > > Note that the assumption of independency is commonly assumed in previous distribution distillation methods, including Dirichlet distillation (Malinin et al., 2019) and EDFM (Park et al., 2025). Our Gaussian model, however, is more flexible than Dirichlet distillation
> > > and EDFM in the sense that our model treats $f(\mathbf{x}_1)$ and $f(\mathbf{x}_2)$ being dependent (and inferring their
> > > dependency by the covariance matrix) while Dirichlet distillation
> > > and EDFM treat them independent. This would be a partial reason for the superiority of our Gaussian model.
> > >
> > >
> > > ## Q3.
> > > We fully agree with your suggestion and are already extending our study in two directions:
> > >
> > > 1. Lager dataset : We are currently running additional experiments on ImageNet to confirm that our method scales well to more complex data.
> > > 2. Lager model : To explore larger models, we are distilling knowledge from LLaMA-v2-7B into MobileLLM-125M/350M (Liu et al., 2024).
> > >
> > > We will include the complete results of these experiments in the final version of the manuscript.
> > > We hope that the expanded experiments will strengthen your confidence in the reliability of our method.
> > >
> > > # Reference
> > > - Gal, Yarin. "Uncertainty in deep learning." (2016).
> > > - Liu, Z., Zhao, C., Iandola, F., Lai, C., Tian, Y., Fedorov, I., ... \& Chandra, V. (2024, February). Mobilellm: Optimizing sub-billion parameter language models for on-device use cases. In Forty-first International Conference on Machine Learning.

---

> > > > ### Comment · Reviewer_jk47 · 2025-08-05
> > > > **Reply to authors**
> > > >
> > > > Dear authors,
> > > >
> > > > Thank you so much for your follow-up responses. The answer to Q1 is quite clear to me. Would it be possible to add the comparison between your method and Dirichlet using the epistemic uncertainty estimates for OOD? As I know, in the Dirichlet paper, the authors show that they achieve the best scores using the epistemic (knowledge) uncertainty estimates for OOD.
> > > >
> > > > Regarding Q2, I understand the difficulties of providing a theoretical analysis/certificate. I would suggest adding a brief discussion in the future work.
> > > >
> > > > Regarding Q3, thank you so much for your replies regarding ongoing experiments. Good luck!
> > > >
> > > > For now, I will raise my score and confidence to 4.
> > > >
> > > > Best regards,

---

> > > > > ### Author Response · Authors · 2025-08-06
> > > > > **Response to the Reviewer jk47**
> > > > >
> > > > > ## Answer
> > > > > Thank you very much for your valuable suggestion to include a comparison that focuses specifically on epistemic uncertainty for out-of-distribution (OOD) detection. Following your advice, we have evaluated OOD performance using the epistemic uncertainty for DLF and the Dirichlet distillation (Proxy-EnD²).
> > > > >
> > > > > The experimental setting is identical to that described in our previous response: CIFAR-10 serves as the in-distribution dataset, while SVHN, CIFAR-100, and Tiny-ImageNet are treated as OOD.
> > > > > Every model uses the same Wide-ResNet-16-1 backbone (≈ 0.17 million parameters), and performance is reported with the AUROC metric.
> > > > > The full numbers are summarized in
> > > > >
> > > > > |method|# of parameters|SVHN|CIFAR-100|Tiny-ImageNet|
> > > > > |-|-|-|-|-|
> > > > > |Proxy-EnD²|0.18M|0.8958|0.8288|0.8334
> > > > > |DLF|0.18M|0.9393|0.8408|0.8348
> > > > >
> > > > > Employing epistemic uncertainty instead of total predictive uncertainty improves OOD detection across all datasets, and under this condition DLF consistently performs better than Proxy-EnD².
> > > > > We appreciate your insightful feedback and will include these results into the final version of the manuscript.

---

> > > > > > ### Comment · Reviewer_jk47 · 2025-08-08
> > > > > >
> > > > > > Dear authors,
> > > > > >
> > > > > > Thank you so much for your reply. I have updated my assessments as I promised. Good luck!
> > > > > >
> > > > > > best regards,

---

### Official Review · Reviewer_e6dg · 2025-07-01

**Clarity:** 2
**Significance:** 3
**Originality:** 3
**Rating:** 4
**Confidence:** 3

**Summary:**

This paper propose a novel distillation method that preserves the uncertainty estimation capabilities of the original model while achieving performance gains across multiple downstream tasks. The core innovation lies in the adoption of distributional distillation, wherein the teacher model is treated as samples from a Gaussian process via a deep latent factor model, and the optimization is performed using the Expectation-Maximization (EM) algorithm.

**Questions:**

The paper repeatedly claims that DLF reduces computational and inference overhead during distillation, yet lacks concrete analysis of actual costs and complexity. What is the practical computational cost of the model during distillation? According to Eq. in line169, both $\mu$  and  $\Phi$ are local networks dependent on design points—does the computational overhead increase with larger $m$?

**Ethical Concerns:**

["NO or VERY MINOR ethics concerns only"]

**Final Justification:**

My hesitation at the beginning of this article mainly came from the comparison of the missing baseline model and the further calrify of motivation, which were resolved by the author in the discussion. What's more, the author has further proved my doubts about computing expenses through theory and practical experiments. In summary, I decided to improve my score.

**Quality:**

2

**Strengths And Weaknesses:**

Strengths：

1.This paper conducts experiments across a wide range of tasks, including classification, regression, and distribution shift, demonstrating a well-rounded and comprehensive evaluation.

2. The proposed DLF framework explicitly models both the mean and covariance structure, effectively preserving the uncertainty estimation capability of the teacher ensemble.

 Weaknesses：

1.Table 1 does not compare the proposed method against other distributional distillation approaches, and instead focuses primarily on one-to-one distillation baselines. Although line 206 explains the exclusion of Dirichlet distillation, including such methods—especially those in Section 2.2.2 ([17, 18])—would strengthen the experimental completeness.

2.While Section 3.3 briefly discusses the benefits of modeling with Gaussian distributions, the theoretical advantages over alternative distributional approaches are not sufficiently elaborated. Given that this is a core contribution, further justification is needed. For example, Dirichlet distributions naturally model class probabilities due to non-negativity and normalization, whereas Gaussians require additional normalization—does this lead to better uncertainty estimation?

3. Most of the equations in the paper are not properly numbered, and the baseline models used in the experimental setup are not cited.

---

> ### Author Rebuttal · Authors · 2025-07-31
>
> Thank you for your concise review. We appreciate your ability to highlight key aspects of our work, and your feedback helped us refine our focus in the revision process.
>
> # Weakness
>
> ## 1.
> In the original manuscript, we did not include Dirichlet distillation (Proxy-EnD²) in the experiments since the papers for Hyrda and BE reported that their algorithms are superior to Dirichlet distillation. We focused mainly on comparing these two deterministic distillation methods with our DLF.
>
> In Table below, we add Dirichlet distillation. In addition, we add a new distribution distillation method called Ensemble Distillation via Flow Matching (EDFM; Park et al., 2025), which we found after we submitted our paper. Note that DLF outperforms Proxy-EnD² and EDFM with large margins in terms of the accuracy and NLL while DLF is not much worse than Proxy-EnD² in terms of ECE.
>
> |dataset|method|Acc(%) ↑|NLL ↓|ECE(%) ↓|
> |--|--|--|--|--|
> |CIFAR-10|Teachers|94.24|0.1539|0.9|
> ||small-Ens|92.87|0.2377|3.93|
> ||Hydra|93.16|0.2660|4.15|
> ||LBE|93.25|0.2480|4.11|
> ||Proxy-EnD²|90.92|0.2861|2.08|
> ||EDFM|90.62|0.2858|2.78|
> ||DLF (Ours)|93.40|0.2246|2.79|
> |--|--|--|--|--|
> |CIFAR-100|Teachers|81.36|0.7167|1.41|
> ||small-Ens|79.29|1.0413|12.90|
> ||Hydra|77.42|1.2912|12.70|
> ||LBE|79.58|1.0110|13.42|
> ||Proxy-EnD²|67.62|1.2355|7.35|
> ||EDFM|64.17|1.6741|11.35|
> ||DLF (Ours)|79.68|0.8974|9.45|
>
>
> ## 2.
> Several prior works have shown that deep neural networks converge to Gaussian processes, including (Lee et al., 2018), (Khan et al., 2019), and (Gal et al., 2016). Based on these findings, we consider it natural to assume that a teacher models trained with a deep neural networks induce a distribution that can be approximated by a Gaussian process.
>
> We want to emphasize that superior performance of DLF compared to other distributional distillations is not achieved by chance. Instead, there are various advantages of DLF compared to Proxy-EnD² and Ensemble Distillation via Flow Matching (EDFM; Park et al., 2025), a recent distribution distillation method. We think that there are three properties any desirable models for distribution distillation are expected to have: (1) large support, (2) easy and stable learning algorithm and (3) flexibility to be applied to downstream tasks. Below, we explain why DLF is desirable in view of these three properties.
>
> -  DLF is a Gaussian process, which is well known to cover large class of distributions (Garriga-Alonso et al., 2019; Neal et al., 1996; Calandra et al., 2016; Huang et al., 2015). Even if teacher ensembles are not Gaussian, a Gaussian process can approximate it closely. While Dirichlet distributions are often used to represent class probabilities due to their inherent non-negativity and normalization constraints, they are inherently limited to classification tasks as mentioned in answer of Weakness 1. In addition, DLF treats teacher ensembles as random functions rather than random vectors as Dirichlet and Flow matching distillations do. See our reply to Question 1 of Reviewer jk47 for detailed discussions for this point.
>
> - Existence of easy and stable learning algorithms is a critical fact when choosing an algorithm for distributional distillation. Learning the DLF model is numerical easy and stable since the efficient EM algorithm is available. It is well known that estimating the parameters in the Dirichlet distribution is difficult (Minka et al., 2000; Chai et al., 2013) and thus it would not be easy to learn the Dirichlet distillation model which would a partial reason for inferior performance. For Flow matching distillation, pretrained features should be fed when learning. That is, end-to-end learning is not possible. In contrast, DLF is learned in a end-to-end fashion (i.e. feature vectors and distribution are learned simultaneously).
>
> - Flexibility is also an important aspect. As we have shown in Section **5.3**,
> the learned features by DLF is superior than those by a single DNN or Hydra for downstream tasks.
>
> - To sum up, the idea of estimating the distribution for teacher ensembles is not new but
> efficient implementation is not an easy task and DLF is such a method.
>
>
> As explained in Introduction, the predictive distribution is obtained by $\hat{p}(y|\mathbf{x})=\sum_{i=1}^n p(y|\mathbf{x},\theta_i)/n,$ where $\theta_1,\ldots,\theta_n$ are ensemble members. For DLF, As shown in Figure 1 (“Inference on new data”), we can generate $\theta_1,\ldots,\theta_n$ by  sampling $n$ many independent latent vectors  $\mathbf{z}_1,\ldots,\mathbf{z}_n$ from $\mathcal N_q(\mathbf 0,\mathbf I_q)$, forwarding each sample through the network to have ensemble members.  All of the uncertainty measures are calculated based on $\hat{p}(y|\mathbf{x}).$ We will state explicitly this procedure in the revision.
>
>
> ## 3.
> In the revised manuscript, we will number all displayed equations. Citations for the baseline methods (e.g., Hydra, Batch-Ensemble) are already provided in Section 2.2, but were omitted later due to the page limitation. For clarity, we will also include the same citations in the description of our experimental setup.
>
>
> # Question
> ## 1.
> As stated in line 137, $\mu_{\theta}$ and $\Phi_{\theta}$ are local networks whose output sizes depend only on the target dimension and latent dimension; they do not scale with the number of design points. Thus $m$ increases the amount of data processed during training but does not change the per sample inference cost. Moreover, DLF is that it can be learned easily using the EM algorithm. As detailed in Appendix B, the E‑step has the closed‑form solution, so training time is not computationally demanding. To confirm this argument, we measure the computing times for the training as well as inference in the table below:(1) Multiply–Accumulate Operations (MAC, measured in MMac) for inference, (2) inference latency (ms), and (3) training time (hours).
> We will add this result in the revised manuscript.
>
> |Method|CIFAR-10 MAC|CIFAR-10 Latency|CIFAR-10 Training Time|CIFAR-100 MAC|CIFAR-100 Latency|CIFAR-100 Training Time|
> |-|-|-|-|-|-|-|
> |Teacher|224|7.1|10.2|3400|7.6|17.4|
> |small-Ens|108|4.1|4.6|224|7.4|12.5|
> |Hydra|27|1.1|3.1|56|1.9|10.28|
> |LBE|108|2.4|6.5|224|4.6|21.5|
> |Proxy-EnD²|27|1.2|4.6|56|2.1|9.2|
> |DLF (Ours)|27|1.5|4.2|850|2.7|11.28|
>
>
> # Refference
> - Lee, J., Bahri, Y., Novak, R., Schoenholz, S. S., Pennington, J., \& Sohl-Dickstein, J. (2017). Deep neural networks as gaussian processes. arXiv preprint arXiv:1711.00165.
> - Khan, M. E., Immer, A., Abedi, E., \& Korzepa, M. (2019). Approximate inference turns deep networks into Gaussian processes. Advances in neural information processing systems, 32.
> - Gal, Y., \& Ghahramani, Z. (2016, June). Dropout as a bayesian approximation: Representing model uncertainty in deep learning. In international conference on machine learning (pp. 1050-1059). PMLR.
> - Park, J., Nam, G., Kim, H., Yoon, J., \& Lee, J. Ensemble Distribution Distillation via Flow Matching. In Forty-second International Conference on Machine Learning.
> - Minka, T. (2000, November). Estimating a Dirichlet distribution.
> - Chai, D., Forstner, W., \& Lafarge, F. (2013). Recovering line-networks in images by junction-point processes. In Proceedings of the IEEE Conference on Computer Vision and Pattern Recognition (pp. 1894-1901).
> - Garriga-Alonso, A., Rasmussen, C. E., \& Aitchison, L. (2018). Deep convolutional networks as shallow gaussian processes. arXiv preprint arXiv:1808.05587.
> - Neal, R. M. (1996). Priors for infinite networks. In Bayesian learning for neural networks (pp. 29-53). New York, NY: Springer New York.
> - Calandra, R., Peters, J., Rasmussen, C. E., \& Deisenroth, M. P. (2016, July). Manifold Gaussian processes for regression. In 2016 International joint conference on neural networks (IJCNN) (pp. 3338-3345). IEEE.
> - Huang, W. B., Zhao, D., Sun, F., Liu, H., \& Chang, E. Y. (2015, June). Scalable Gaussian Process Regression Using Deep Neural Networks. In IJCAI (pp. 3576-3582).

---

> > ### Comment · Reviewer_e6dg · 2025-08-03
> >
> > Thank you for your response.  W1 and W2 have been addressed. Concerning Q1, the experimental results suggest that DLF’s computational complexity is comparable to that of the other methods; I therefore recommend adding a complementary theoretical complexity analysis.

---

> > > ### Author Response · Authors · 2025-08-05
> > > **Response to the Reviewer e6dg**
> > >
> > > # Answer
> > > Thank you for your valuable suggestion.
> > > We have added the computational complexity of Hydra, Batch Ensemble (BE), and DLF in terms of FLOPs (Floating Point Operations), using Wide-ResNet-16-1 as the shared backbone architecture for all methods.
> > >
> > > FLOPs provide a hardware-independent measure of computational cost and are widely used for theoretical complexity analysis in deep learning research.
> > > For example, ResNet (He et al., 2016), and EfficientNet (Tan et al., 2019) use FLOPs to evaluate and compare model efficiency across architectures.
> > >
> > > ## Notation
> > >
> > > | Symbol               | Meaning                                                                |
> > > | -------------------- | ---------------------------------------------------------------------- |
> > > | $c$                | Number of classes                                               |
> > > | $B$                | Number of samples in a mini-batch                                      |
> > > | $N$                | Number of ensemble members         |
> > > | $q$                | Latent dimension in DLF                                                |
> > > | $F\_{\text{wide}}$ | FLOPs for one forward pass through the Wide-ResNet-16-1 body       |
> > > | $F\_{\text{head}}$ | FLOPs for one Hydra head                                           |
> > > | $F\_{\mu, \Phi}$    | FLOPs for the small FC layer that produces $\mu$ and $\Phi$ in DLF |
> > > | $\alpha$           | Forward + backward multiplier         |
> > >
> > > Below, we summarize the FLOPs per training step for Hydra, Batch Ensemble, and our proposed DLF method:
> > >
> > > * **Hydra**:
> > >   $$\text{FLOPs}\_{\text{Hydra}} = \alpha B \bigl( F\_{\text{wide}} + N F\_{\text{head}} \bigr)$$
> > >
> > > * **Batch Ensemble**:
> > >   $$\text{FLOPs}\_{\text{BE}} \approx \alpha B (1+\epsilon) F\_{\text{wide}} $$
> > >   where $\epsilon$ (extra cost from rank-1 matrix) is usually $\le 0.05$.
> > >
> > > * **DLF**:
> > >   $$\text{FLOPs}\_{\text{DLF}} = \alpha B\bigl(F\_{\text{wide}} + F\_{\mu, \Phi}\bigr) + B q^{2} + q^{3} + N(B c q + q^2)$$
> > >
> > > In all three methods, $F\_{\text{wide}}$ represents the FLOPs of the shared backbone. Note that the backbone FLOPs $F_{\text{wide}}$ is typically on the order of GFLOPs, i.e., approximately $10^9$ FLOPs.
> > >
> > > In Batch Ensemble, the additional cost comes from the overhead factor $(1+\epsilon)$ applied to the full backbone. Although $\epsilon$ is usually small (typically less than 0.05), its effect is not negligible due to the large scale of $F_{\text{wide}}$.
> > >
> > > In contrast, Hydra shares the backbone and only adds a small cost from the lightweight head modules. Since $F_{\text{head}} \ll F_{\text{wide}}$, the total complexity remains dominated by the shared body, even with large ensemble sizes $N$.
> > >
> > > Similarly, although DLF requires additional computations such as $F\_{\mu, \Phi}$, and matrix operations such as $Bq^2, q^3, N(Bcq + q^2)$, their contributions are negligible since $q$ is typically small (often less than 20 in practice) and $F\_{\mu, \Phi}, Bq^2, q^3, N(Bcq + q^2) \ll F\_{\text{wide}}$.
> > > Consequently, the dominant cost in DLF also comes from the shared backbone.
> > >
> > > We believe this theoretical comparison complements the empirical results and helps clarify the relative computational efficiency of DLF.
> > >
> > > # Reference
> > > - He, K., Zhang, X., Ren, S., & Sun, J. (2016). Deep residual learning for image recognition. In Proceedings of the IEEE conference on computer vision and pattern recognition (pp. 770-778).
> > > - Tan, M., & Le, Q. (2019, May). Efficientnet: Rethinking model scaling for convolutional neural networks. In International conference on machine learning (pp. 6105-6114). PMLR.

---

> > > > ### Comment · Reviewer_e6dg · 2025-08-05
> > > >
> > > > Thank you for the author's detailed reply. My problems have basically been solved. I hope you can reflect some of the modifications in our discussion in the final pdf. I will raise my score to 4 and confidence to 3.

---

### Official Review · Reviewer_Lsh6 · 2025-07-03

**Clarity:** 3
**Significance:** 2
**Originality:** 3
**Rating:** 4
**Confidence:** 4

**Summary:**

This paper proposes Gaussian distillation, a distribution-distillation technique that treats every teacher in a deep ensemble as a realization of a Gaussian process whose mean and covariance are parameterized by a student network called the Deep Latent Factor (DLF) model. The parameters are learned with an EM procedure that integrates out latent factors, and ensemble members for inference are sampled from the estimated process. The empirical results on regression and classification tasks show the advantage of Gaussian distillation over several benchmarks, however, this paper failed to compare with current state-of-art methods, and are limited on small datasets such as small-to-medium vision and tabular benchmarks. Moreover, the computational burden and the stability of EM optimization are also concerns.

**Questions:**

•	Why “closer to the coverage probability of a teacher ensemble is better” in table 1? normally closer to nominal value should be better.

•	In real-world applications, teachers models are pre-trained and likely with varying structures and trained over not necessarily the same data. I am interested in such settings, in the performance of the method proposed.

•	The authors should attempt and evaluate the methods that have been suggested with larger dataset, such as ImageNet, and larger models.

•	The comparison of state-of-art baselines are necessary.

•	How does DLF scale to high-dimensional outputs (e.g., pixel-wise regression)?

•	On line 187, any specific reason to choose to set \sigma generated from the inverse gamma distribution?

**Ethical Concerns:**

["NO or VERY MINOR ethics concerns only"]

**Limitations:**

yes

**Quality:**

2

**Strengths And Weaknesses:**

Quality

Strengths:

•	The paper is technically sound, with support through empirical explorations and theoretical confirmation.
The suggested scheme is complete, aided by particulars such as design-point selection, choice of initial point etc.

Weaknesses:

•	Scalability of EM: The EM is computationally slow. A discussion of complexity would explain how the technique scales to, e.g. ImageNet. Computational time comparison with other baselines would also be required.

•	The experimental part only compare techniques like Hydra, which are somewhat obsolete, and do not compare with a few recent distribution distillation baselines like packed-ensembles https://doi.org/10.48550/arXiv.2210.09184 or Bayesian neural-network distillation.

•	The authors proposed the methods in the field of distribution distillation methods, and distribution distillation methods are introduced in, e.g., Dirichlet distillation, is introduced in section 2.2, but not compared in the experiment section.

•	No Large-Scale Tests: Small models are used in experiments. Not known whether DLF scales to billion-scale LLMs.

Clarity

Strengths:

•	The paper is well written and well structured.

•	Core ideas, such as viewing teacher outputs as realizations of a DLF process, are illustrated clearly.

Weaknesses:

•	Clarity suffers from heavy notation and frequent cross-references to appendices for critical arguments.

•	The theoretical findings are approximately discussed in the main part. Although some details are given in Appendix, it is not very well-organized. Also, the authors give many mathematical result in appendix, but the insights or conclusions gained are not clearly discussed in the main text. This issue also exists in the part of ablation study.

•	It is not clear how authors define LLM in section 5.2. In today’s standard, models like RoBERTa are usually not called LLM.

Significance

Strengths:

•	Compressing deep ensembles and retaining calibrated uncertainty is a timely practical problem, and the Gaussian distillation is a principled solution to it.

•	The technique developed is applicable both to regression and classification methods, while original KD is designed for classifications.
Weaknesses:

•	Most benchmarks are small to medium, the compared methods are naive, and the gains over small-Ens are modest, especially the coverage rate proposed in Table 1.

•	The LLM example relies on DistilRoBERTa rather than a modern billion-parameter model, so the reader is left to extrapolate the method’s benefit in truly large-scale settings.

•	The assumptions of the theorem are strong. The authors should be able to clearly illuminate in what way theoretical assumptions (e.g., strong convexity, good initialization) might constrain practical applications.

Originality

Strengths:

•	The paper is novel in framing distribution distillation as a latent-factor Gaussian process parametrizable through a student network.
Weaknesses:

•	The method integrates many components, Gaussian latent factors, EM optimization, and factor-analytic covariance parameterization, from established literature. Its move from Dirichlet to Gaussian distribution distillation is a careful extension, but could be perceived as incremental rather than a complete paradigm shift.

---

> ### Author Rebuttal · Authors · 2025-07-31
>
> We sincerely appreciate your thorough review and the thoughtful questions you raised. Your comments not only helped us clarify key components of our work but also encouraged us to consider valuable directions for future extensions. Thank you for engaging deeply with our paper.
> # Weakness
> ## Quality 1.
> A novelty of DLF is that it can be learned easily using the EM algorithm. As detailed in Appendix B, the E‑step has the closed‑form solution, so training time is not computationally demanding. To confirm this argument, we measure the computing times for the training as well as inference in the table below: (1) Multiply–Accumulate Operations (MAC, measured in MMac) for inference, (2) inference latency (ms), and (3) training time (hours). We will add this result in the revised manuscript.
> |Method|CIFAR-10 MAC|CIFAR-10 Latency|CIFAR-10 Training Time|CIFAR-100 MAC|CIFAR-100 Latency|CIFAR-100 Training Time|
> |-|-|-|-|-|-|-|
> |Teacher|224|7.1|10.2|3400|7.6|17.4|
> |small-Ens|108|4.1|4.6|224|7.3|12.5|
> |Hydra|27|1.1|3.1|56|1.8|10.28|
> |LBE|108|2.4|6.5|224|4.3|21.5|
> |Proxy-EnD²|27|1.2|4.6|56|2.1|9.2|
> |DLF (Ours)|27|1.5|4.2|56|2.6|11.28|
> ## Quality 2.
> We would like to clarify that packed-ensembles is not a knowledge-distillation method, but rather an efficient ensemble construction technique. We agree that including a recent distribution-distillation baseline would strengthen the evaluation. We have therefore added experiments with “Ensemble Distribution Distillation via Flow Matching” (EDFM; Park et al., 2025) which we found after the submission.
> ## Quality 3.
> In the original manuscript, we omitted Dirichlet distillation (Proxy-EnD²) from our experiments because the Hydra and BE studies demonstrated superior performance over it; instead, we focused on comparing those two deterministic distillation methods with our DLF.
>
> In Table below, we add Dirichlet distillation and EDFM. Note that DLF outperforms Proxy-EnD² and EDFM with large margins in terms of the accuracy and NLL while DLF is not much worse than Proxy-EnD² in terms of ECE.
> |dataset|method|Acc(%) ↑|NLL ↓|ECE(%) ↓|
> |-|-|-|-|-|
> |CIFAR-10|Proxy-EnD²|90.92|0.2861|2.08|
> ||EDFM|90.62|0.2858|2.78|
> ||DLF (Ours)|93.40|0.2246|2.79|
> |CIFAR-100|Proxy-EnD²|67.62|1.2355|7.35|
> ||EDFM|64.17|1.6741|11.35|
> ||DLF (Ours)|79.68|0.8974|9.45|
> ## Quality 4.
> We agreed that RoBERTa is not of billion-scale LLMs. However, RoBERTa is used popularly for studying finetuning LLM (Yang et al., 2023). We could try to distill larger LLM but it would take significant amount of times. In our experiments, we only distilled the LoRA part of RoBERTa while we used the pretrained distilled RoBERTa for distillation of RoBERTa. We are trying to apply DLF to large LLMs such as (Liu et al., 2024) but we do not think we can get the results by the due date of rebuttal. If possible, we will report the results in the discussion period.
> ## Clarity 1.
> We are sorry for notational complexity. Due to page limitation, we moved most of notations into Appendix. In revision, we will try our best to improve the readabilty.
> ## Clarity 2.
> The main motivation of the theoretical result in Section 3.4 is to answer the question about the choice of good design points for distribution distillation. The theoretical result suggests that the design points whose distribution is close to that of test data is promising. We will polish Section 3.4 more in the revision.
> ## Clarity 3.
> We used the term LLM for RoBERTa since the previous literature (Yang et al., 2023) use this terms. It would be better to name it just SLM (smalle laguage model) or a high-capacity pretrained transformer. We will clarify this terminology in the revision.
> ## Significance 1.
> For a large and hard problem, in Section 5.2, we have conducted experiments using RoBERTa (approximately 355 million parameters) to evaluate our method. As the referee suggested, we analyzed Tiny-ImageNet, a 200-class subset of ImageNet. DLF still outperforms the other baselines even including small ensembles.
> |dataset|method|Acc(%) ↑|NLL ↓|ECE(%) ↓|MAC|Latency|Training Time
> |-|-|-|-|-|-|-|-|
> |Tiny-ImageNet|Teachers|68.74|1.2806|3.22|166571|51.6|32.9|
> ||small-Ens|57.67|1.7887|9.36|77382|26.0|53.6|
> ||Hydra|55.94|1.7765|6.49|19353|6.3|24.2|
> ||LBE|54.44|2.2804|6.04|77377|29.0|60.4|
> ||DLF (Ours)|61.92|1.5450|3.06|19353|7.2|34.6|
>
> Because “small-Ens” consists of multiple models, it uses far more parameters. Although its accuracy gains are modest, DLF matches or exceeds its predictive performance, delivers higher coverage, and does so with only a fraction of the parameters and computation. We believe this efficiency–performance balance is exactly the strength of DLF. Please note that baselines such as Hydra and BE also share parameters in small-Ens.
> ## Significance 2.
> We are trying to apply DLF to large LLMs but we do not think we can get the results by the due date of rebuttal. If possible, we will report the results in the discussion period.
> ## Significance 3.
> Assumptions like Lipschitz continuity and sparsity are common in theoretical analyses of deep generative models and nonparametric neural network regression (e.g., Schmidt-Hieber 2020; Chae 2023; Wing Hung Wong et al. 1995), and we agree they may not always hold in practice.  However, the main motivation of the theoretical result in Section 3.4 is to address how to choose design points for distribution distillation. Even if there are many strong assumptions (Gaussianity of teacher ensembles and so on), the theoretical result suggests that the design points whose distribution is close to that of test data is promising. We will polish Section 3.4 more in the revision.
> # Originality 1.
> We want to emphasize that superior performance of DLF compared to other baselines
> is not achieved by chance. Instead, there are various advantages of DLF compared to Proxy-EnD² and EDFM.
> We think that there are three properties any desirable models for distribution distillation are expected to have:
> (1) large support, (2) easy and stable learning algorithm and (3) flexibility to be applied to downstream tasks.
> Below, we explain why DLF is desirable in view of these three properties.
> - DLF is a Gaussian process, which is well known to cover large class of distributions (Garriga-Alonso et al., 2019; Neal et al., 1996; Calandra et al., 2016; Huang et al., 2015). Even if teacher ensembles are not Gaussian, a Gaussian process can approximate it closely. In addition, DLF treats teacher ensembles as random functions rather than random vectors as Dirichlet and Flow matching distillations do. See our reply to Question 1 of Referee 4 for detailed discussions for this point.
> - Existence of easy and stable learning algorithms is a critical fact when choosing an algorithm for distributional distillation. Learning the DLF model is numerical easy and stable since the efficient EM algorithm is available. It is well known that estimating the parameters in the Dirichlet distribution is difficult (Minka et al., 2000; Chai et al., 2013) and thus it would not be easy to learn the Dirichlet distillation model which would a partial reason for inferior performance. For Flow matching distillation, pretrained features should be fed when learning. That is, end-to-end learning is not possible. In contrast, DLF is learned in a end-to-end fashion (i.e. feature vectors and distribution are learned simultaneously).
> - Flexibility is also an important aspect. As we have shown in Section **5.3**, the learned features by DLF is superior than those by a single DNN or Hydra for downstream tasks.
> - To sum up, the idea of estimating the distribution for teacher ensembles is not new but efficient implementation is not an easy task and DLF is such a method.
> # Question
> ## 1.
> Because our student model is distilled directly from the teacher ensemble, we chose the ensemble’s empirical coverage as the reference: matching it indicates that the student has inherited the teacher’s calibration. We agree, however, that proximity to the nominal target (e.g., 95 %) is also informative. In the revision we will report both values so that readers can assess calibration against the ensemble and against the nominal level.
> ## 2.
> Thank you for raising this interesting point. Due to the brevity of the rebuttal period, we were unfortunately unable to conduct experiments with teacher models pretrained on different datasets. We plan to use the upcoming discussion phase to perform these experiments and incorporate the results into the final manuscript. We appreciate your valuable suggestion.
> ## 3.
> Please refer to our response to Significance 1.
> ## 4.
> Please refer to our response to Quality 3.
> ## 5.
> As noted in line 146, DLF Framework depends on the output dimension—so as outputs grow, both the parameter count and $\mathbf{Z}$’s dimensionality increase, which can slow training. However, we believe that selecting an appropriate latent dimension $q$ ensures efficient learning.
>
> Moreover, because DLF framework can estimate class‑specific covariance structures, it captures complex relationships more accurately and delivers superior performance on high-dimensional datasets like Tiny-ImageNet. For these reasons, we believe that DLF not only scales to tasks like pixel‑wise regression but may actually gain relative advantage as output dimensionality increases.
> ## 6.
> We use an inverse-gamma prior for the variance parameter $\sigma^{2}_{\epsilon}$ because it is conjugate to the Gaussian likelihood in our model. This conjugacy leads to closed-form updates within the EM algorithm and simplifies marginal-likelihood computations.
>
> # Reference
> - Yang, A. X., Robeyns, M., Wang, X., & Aitchison, L. (2023). Bayesian low-rank adaptation for large language models.
> - Liu, Z., Zhao, C., Iandola, F., Lai, C., Tian, Y., Fedorov, I., ... & Chandra, V. (2024, February). Mobilellm: Optimizing sub-billion parameter language models for on-device use cases.
> - Please see the reference list of Reviewer e6dg.

---

> > ### Comment · Reviewer_Lsh6 · 2025-08-04
> > **response to the authors**
> >
> > Thank you very much for your detailed response. Some of my questions have been addressed, and I look forward to seeing results on more complex models and datasets. Please make sure to include the updated results and revised descriptions in your final manuscript. Thank you.

---

> > > ### Author Response · Authors · 2025-08-05
> > > **Response to the Reviewer Lsh6**
> > >
> > > # Answer
> > > Thank you very much for your feedback and for highlighting the importance of evaluating our approach on more challenging settings. We agree with your suggestion and are already expanding our study in two directions:
> > >
> > > 1. **Lager dataset** : We are currently running additional experiments on ImageNet to confirm that our method scales well to more complex data.
> > > 2. **Lager model** : To explore larger models, we are distilling knowledge from LLaMA-v2-7B into MobileLLM-125M/350M (Liu et al., 2024).
> > >
> > > We will include the complete results of these experiments in the final version of the manuscript.
> > > We hope that the expanded experiments will strengthen your confidence in the reliability of our method.
> > >
> > > # Reference
> > > - Liu, Z., Zhao, C., Iandola, F., Lai, C., Tian, Y., Fedorov, I., ... \& Chandra, V. (2024, February). Mobilellm: Optimizing sub-billion parameter language models for on-device use cases. In Forty-first International Conference on Machine Learning.

---

> > > > ### Comment · Reviewer_Lsh6 · 2025-08-08
> > > > **response to the authors**
> > > >
> > > > Thank you for your efforts! I believe the additional experiments will strengthen your paper.
> > > >
> > > > One further comment regarding Q1: although we distill from the teachers, the nominal target should be the gold standard we aim to achieve. Your results show that small-Ens achieves the best performance with respect to this target (and even when using the criterion of closeness to the teachers, it still performs comparably to DLF), and in some cases even outperforms the teachers. This is worth further exploration and at least a discussion.
> > > >
> > > > Good luck!

---

### Official Review · Reviewer_vWb7 · 2025-07-04

**Clarity:** 2
**Significance:** 3
**Originality:** 3
**Rating:** 4
**Confidence:** 2

**Summary:**

The paper considers a problem of knowledge distillation of deep ensembles (though it is mentioned it could be applied to Bayesian NNs as well) and proposes a novel method for knowledge distillation based on a deep latent factor model where the latent factors follows the normal distribution. Parameters of the model are estimated with the EM-algorithm. The estimate of the parameters is based on predictions of the teacher ensemble models at so-called design points. The paper proposes a theoretical analysis for the choice of the design points.

**Questions:**

1. Could you please elaborate about why Dirichlet distillation has been left out of the comparison?
2. Could you please comment anything about the computational cost?
3. How are predictions are made and uncertainty is estimated with the proposed model?

**Ethical Concerns:**

["NO or VERY MINOR ethics concerns only"]

**Final Justification:**

I have read the other reviews and authors' responses. The authors have addressed most of my concerns therefore I have increased my score.

**Limitations:**

The authors have discussed the areas of future work in the conclusion which can be considered as discussion of limitations to some extent, but it seems it could be more explicit discussion of limitations. No discussion about potential negative societal impact of their work is discussed, though it is a theoretical work which doesn't seem to have a lot of potential negative societal impact.

**Paper Formatting Concerns:**

Formulas are not numbered

**Quality:**

2

**Strengths And Weaknesses:**

**Strengths:**

*Quality*: The paper claims are supported with rather extensive empirical evidence, covering regression tasks, classification of image data, fine-tuning of LLMs. There is a theoretical base for the choice of design points. However, please see below.

*Clarity*: The paper is generally easy to read, but please see below.

*Significance*: The paper considers an important problem the experiments show promising results of the proposed methods over baselines.

*Originality*: The idea appears to be novel.

**Weaknesses:**

*Quality*:

* I am not convinced why Dirichlet distillation was left out of empirical comparison. This line of work seems like a very interesting baseline as the only other distribution distillation. Please see below for the details, why I am not convinced.
* The main motivation of the paper is to provide a cost effective alternative to a deep ensemble, yet there is no computational cost discussion in the paper.
* Image experiments are done on CIFAR-10/100 only, which is not too extensive. Regression datasets are UCI only, which is again is not very large and hard.

*Clarity*: The paper is seemingly easy to read and follow, but it seems to be missing important details, or maybe I am missing them. After reading both the paper and the appendix, I realise I do not know how the model is making predictions and actually estimate uncertainty, which is the main selling point of the paper. I.e., the paper describes in details (well done) how to estimate the parameters with the EM-algorithm, but I am not clear how to use those parameters further for predictions.

*Originality*: The paper puts the proposed work rather well in the literature on knowledge distillation, but nothing at all is discussed about deep latent factor models existing in the literature. I believe it helps for the context and also to clarify what exactly is the novelty proposed in the paper.

Other comments/suggestions:
1. Lines 136-140. q is not defined
2. Lines 169-171. $\mathbf{z}_1, …, \mathbf{z}_n$ are not defined
3. Line 205-206. I am confused about the claim “It is known that Dirichlet distillation is inferior to one-to-one distillation [11, 13]”. I am not too familiar with this literature, but quick look at the provided references do not fully support this claim. Namely, [11] does show some inferiority of Dirichlet distillation it is not overwhelming and shows some trade-off in terms of computational costs. Whereas, I may be missing something, but I can’t see the direct comparison with Dirichlet distillation at all in [13]. The comparison seems to be with just Knowledge distillation [Hinton et al.], whereas [Malinin et al.] only mentioned in the related works but not in the direct comparison. Again, I am sorry if I am missing something as I only briefly looked at these mentioned references, but they do not seem to support the claim that Dirichlet distribution can be ignored for comparison with the proposed work.
4. Line 271. The authors of DistilRoBERTa ask to cite this work Sahn et al. “DistilBERT, a distilled version of BERT: smaller, faster, cheaper and lighter”, which is missing. (I do appreciate the link to hugging face as the paper covers a different distillation)
5. Table 2. It seems the proposed method is referred to as KD here, which is very confusing considering it has been referred to as DLF before that.
6. Figure 2. X-axis is not labelled.
7. Line 303. “significantly reduced inference costs” – the claim is not confirmed by any theoretical or empirical findings.
8. Lines 305-306. “Bayesian DNNs are known to be superior to deep ensembles in uncertainty quantification [25, 26, 27]”. This claim again is too strong and some argue the opposite and the provided references do not fully support this claim. [25] doesn’t show undouble inferiority of deep ensembles, [26] doesn’t compare to deep ensembles at all, [27] does show more undouble inferiority of deep ensembles, but it seems to be for the specifics of node-sparse models (sorry, I am not familiar with this), as it contradicts to findings on the same datasets by Ovadia et al. "Can you trust your model's uncertainty? Evaluating predictive uncertainty under dataset shift." Advances in neural information processing systems 32 (2019). And Ashukha et al. "Pitfalls of In-Domain Uncertainty Estimation and Ensembling in Deep Learning." International Conference on Learning Representations. (2020).
9. Formulas are not numbered
10. Line 579. Missing reference for the Adam optimiser
11. Section 3.4. It is not reading well. Firstly, the theoretical start is quite heavy and, personally, I would prefer having some kind of high-level of motivation before diving into maths. Secondly, the leap from the design points being similar to test data to validation data being a good candidate should be elaborated. As validation data as described follows the training data distribution which is not necessarily the same as the test data distribution.
12. Lines 612-614. L is overdefined. It has been already defined as a lower-triangular matrix for a multivariate case definition.
13. Appendix C.5. Could it be really true that all the experiments were executed using only CPU?
14. Lines 660-663. What are mixup samples?
15. Figure 8. What data is used in this experiment? How would the authors explain why DLF obtain better results than small ensembles?
16. Table 3. MRPC acc results. Bold results for KD is misleading as small-Ens has the same average
17. The provided code does not cover all the experiments from the paper.

---

> ### Author Rebuttal · Authors · 2025-07-31
>
> Thank you for your detailed and thoughtful review. We appreciate the time and care you took in carefully examining various aspects of our work. Your insightful comments helped us identify points for clarification and further improvement.
> Thank you for your detailed feedback on items 2, 4, 5, 6, 9, 10, 12, 13 in the “Other comments/suggestions” part. I will address these points in the revision.
>
> # Weakness
>
> ## Quality 1.
> In the original manuscript, we did not include Dirichlet distillation (Proxy-EnD²) in the experiments since the papers for Hydra and BE reported that their algorithms are superior to Dirichlet distillation. We focused mainly on comparing these two deterministic distillation methods with our DLF.
>
> In the Table below, we add Dirichlet distillation. In addition, we add a new distribution distillation method called Ensemble Distillation via Flow Matching (EDFM; Park et al., 2025), which we found after we submitted our paper. Note that DLF outperforms Proxy-EnD² and EDFM with large margins in terms of the accuracy and NLL while DLF is not much worse than Proxy-EnD² in terms of ECE.
>
> |dataset|method|Acc(%) ↑|NLL ↓|ECE(%) ↓|
> |--|--|--|--|--|
> |CIFAR-10|Teachers|94.24|0.1539|0.9|
> ||small-Ens|92.87|0.2377|3.93|
> ||Hydra|93.16|0.2660|4.15|
> ||LBE|93.25|0.2480|4.11|
> ||Proxy-EnD²|90.92|0.2861|2.08|
> ||EDFM|90.62|0.2858|2.78|
> ||DLF (Ours)|93.40|0.2246|2.79|
> |CIFAR-100|Teachers|81.36|0.7167|1.41|
> ||small-Ens|79.29|1.0413|12.90|
> ||Hydra|77.42|1.2912|12.70|
> ||LBE|79.58|1.0110|13.42|
> ||Proxy-EnD²|67.62|1.2355|7.35|
> ||EDFM|64.17|1.6741|11.35|
> ||DLF (Ours)|79.68|0.8974|9.45|
>
> We want to emphasize that superior performance of DLF compared to other distributional distillations is not achieved by chance. Instead, there are various advantages of DLF compared to Proxy-EnD² and EDFM. We think that there are three properties that any desirable models for distribution distillation are expected to have: (1) large support, (2) easy and stable learning algorithm and (3) flexibility to be applied to downstream tasks. Below, we explain why DLF is desirable in view of these three properties.
>
> - DLF is a Gaussian process, which is well known to cover a large class of distributions (Garriga-Alonso et al., 2019; Neal et al., 1996; Calandra et al., 2016; Huang et al., 2015). Even if teacher ensembles are not Gaussian, a Gaussian process can approximate them closely. In addition, DLF treats teacher ensembles as random functions rather than random vectors as Dirichlet and Flow matching distillations do. See our reply to Question 1 of Referee 4 for detailed discussions for this point.
>
> - Existence of easy and stable learning algorithms is a critical fact when choosing an algorithm for distributional distillation. Learning the DLF model is numerically easy and stable since the efficient EM algorithm is available. It is well known that estimating the parameters in the Dirichlet distribution is difficult (Minka et al., 2000; Chai et al., 2013) and thus it would not be easy to learn the Dirichlet distillation model which would be a partial reason for inferior performance. For Flow matching distillation, pretrained features should be fed when learning. That is, end-to-end learning is not possible. In contrast, DLF is learned in an end-to-end fashion (i.e. feature vectors and distribution are learned simultaneously).
>
> - Flexibility is also an important aspect. As we have shown in Section **5.3**,
> the learned features by DLF is superior to those by a single DNN or Hydra for downstream tasks.
>
> - To sum up, the idea of estimating the distribution for teacher ensembles is not new but an efficient implementation is not an easy task and DLF is such a method.
>
>
> ## Quality 2.
> For inference overhead, we reported the number of parameters in each model in the manuscript in Section C as inference cost is roughly proportional to the number of parameters. As the referee pointed out, in this rebuttal, we measure the computational costs not only for the   inference phase but also the training phase in the table below: (1) Multiply–Accumulate Operations (MAC, measured in MMac) for inference, (2) inference latency (ms), and (3) training time (hours). We did not include EDFM since it uses pretrained features (instead of learning features).
>
> |Method|CIFAR-10 MAC|CIFAR-10 Latency|CIFAR-10 Training Time|CIFAR-100 MAC|CIFAR-100 Latency|CIFAR-100 Training Time|
> |-|-|-|-|-|-|-|
> |Teacher|224|7.1|10.2|3400|7.6|17.4|
> |small-Ens|108|4.1|4.6|224|7.3|12.5|
> |Hydra|27|1.1|3.1|56|1.8|10.28|
> |LBE|108|2.4|6.5|224|4.3|21.5|
> |Proxy-EnD²|27|1.2|4.6|56|2.1|9.2|
> |DLF (Ours)|27|1.5|4.2|56|2.6|11.28|
>
> ## Quality 3.
> For a large and hard problem, in Section 5.2, we have conducted experiments using RoBERTa (approximately 355 million parameters) to evaluate our method. As the referee suggested, we analyzed Tiny-ImageNet, a 200-class subset of ImageNet whose results are provided below. DLF still outperforms the other baselines even including small ensembles.
>
> |dataset|method|Acc(%) ↑|NLL ↓|ECE(%) ↓|MAC|Latency|Training Time
> |--|--|--|--|--|--|--|--|
> |Tiny-ImageNet|Teachers|68.74|1.2806|3.22|166571|51.6|32.9|
> ||small-Ens|57.67|1.7887|9.36|77382|26.0|53.6|
> ||Hydra|55.94|1.7765|6.49|19353|6.3|24.2|
> ||LBE|54.44|2.2804|6.04|77377|29.0|60.4|
> ||DLF (Ours)|61.92|1.5450|3.06|19353|7.2|34.6|
>
> We will conduct further experiments on larger-scale datasets (such as ImageNet) during the revision phase and include those results in the updated manuscript.
>
> ## Clarity.
> As explained in the Introduction, the predictive distribution is obtained by $\hat{p}(y|\mathbf{x})=\sum_{i=1}^n p(y|\mathbf{x},\theta_i)/n,$ where $\theta_1,\ldots,\theta_n$ are ensemble members. For DLF, as shown in Figure 1 (“Inference on new data”), we can generate $\theta_1,\ldots,\theta_n$ by  sampling $n$ independent latent vectors  $\mathbf{z}_1,\ldots,\mathbf{z}_n$ from $\mathcal N_q(\mathbf 0,\mathbf I_q)$, forwarding each sample through the network to have ensemble members.  All of the uncertainty measures are calculated based on $\hat{p}(y|\mathbf{x}).$ We will state explicitly this procedure in the revision.
>
> ## Originality.
> DLF is a kind of the conditional factor model (Chen et al., 2022; Gagliardini et al., 2019) where $\mathbf{Z}$ is the factors and $\Phi_{\theta}(\cdot)$ is the factor loadings depending on covariate $\mathbf{x}.$ Even though there are much literature for the conditional factor models, as far as we know, there is no previous attempt to use deep neural networks for the conditional factor loadings (most models use linear factor loadings) presumably because the model is too complex to fit noisy data. Our key contribution is to suggest DLF to ensemble distillation. As we explained in the reply of Quality 1, our suggestion of using DLF is scientifically well motivated.
>
> # Other comments/suggestions:
> ## 1.
> In lines 137-138, we wrote $\boldsymbol{Z} \sim \mathcal{N}_q(\mathbf{0}, \mathbb{I}_q)$ which indicates that $q$ is the dimension of $\boldsymbol{Z}.$
>
> ## 3 .
> Please refer to our response to Quality 1.
>
> ## 7 .
> Please refer to our response to Quality 2.
>
> ## 8.
> We agree that the wording in Lines 305–306 was too strong. We mentioned Bayesian DNN since we tried to write that the learned DLF can be used as a prior for on-device update of the posterior. Due to the page limitation, we deleted this sentence. We will clarify this point in the revision.
>
> ## 11.
> The main motivation of the theoretical result in Section 3.4 is to answer the question about the choice of good design points for distribution distillation. Even if there are many strong assumptions (Gaussianity of teacher ensembles and so on), the theoretical result suggests that the design points whose distribution is close to that of test data are promising.
>
> Regarding the second point, we assume that the distributions of the training and test data are the same, and suggest using validation data randomly sampled from the entire data. To support this choice, we have empirically confirmed that using validation data for the design points works well in Appendix D.1.1. We will clarify this point in the revision.
>
> ## 14.
> The entire training dataset is denoted as $\mathcal{D}^{\text{train}} = \{(\boldsymbol{x}_1,y_1), \ldots, (\boldsymbol{x}_n,y_n)\}$. For each $i$-th sample in the training set, another sample $(\boldsymbol{x}_i^c, y_i^c)$ is randomly selected. A new mixed sample $(\boldsymbol{x}_i^m, y_i^m)$ is generated by linearly combining the two samples. $(\boldsymbol{x}_i^m, y_i^m) := \lambda_i(\boldsymbol{x}_i, y_i) + (1-\lambda_i)(\boldsymbol{x}_i^c, y_i^c), \quad \lambda_i \in [0,1]$. We will add more explanation of mixup samples in the revision.
>
> ## 15.
> The experiment in Figure 8 uses the Boston Housing dataset, as stated in line 642.
>
> ## 16.
> While the values are displayed as identical to two decimal places, DLF slightly outperforms small-Ens (83.094 vs. 83.0915). For clarity, we will report results to three decimal places in the revised version to avoid any potential misunderstanding regarding the bold formatting.
>
> ## 17.
> We plan to release the full code and scripts for reproducing all experiments in a follow-up update.
>
> # Question
> ## 1.
> Please refer to our response to Quality 1.
> ## 2.
> Please refer to our response to Quality 2.
> ## 3.
> Please refer to our response to Clarity.
>
> # Reference
> - Chen, Q. (2022). A unified framework for estimation of high-dimensional conditional factor models. arXiv preprint arXiv:2209.00391.
> - Gagliardini, P., \& Ma, H. (2019). Extracting statistical factors when betas are time-varying. Swiss Finance Institute Research Paper, (19-65).
> - Please see the reference list of Reviewer e6dg.

---

> > ### Comment · Reviewer_vWb7 · 2025-08-04
> >
> > Thank you for your response. Thank you for including results for Dirichlet Distribution. From the results in Table in your response Quality 1 it looks like an important baseline. Thank you for your detailed answers for why DLF would outperform the others. Do you have any idea why Dirichlet Distribution would have better results in terms of ECE? I appreciate you may not know the reason especially given the short period of rebuttal. Moreover, it underperforms in terms of 2 other metrics, but if you have any thoughts I would be interested to know them. (Please do not spend too much time on this, better focus on other reviewers' comments).
> > In recognition to the rebuttal addressing my main concerns, I am raising my score. Thank you.

---

> > > ### Author Response · Authors · 2025-08-05
> > > **Response to the Reviewer vWb7**
> > >
> > > # Answer
> > >
> > > Thank you for your interest and valuable question.
> > > As noted in  (Tao et al., 2023), a low ECE by itself does not guarantee a good model.
> > > For example, in a binary classification task, a model that assigns a constant confidence of 0.5 to every test sample would achieve an ECE near zero, yet its accuracy would be only about 50\%.
> > > This model would not be regarded as effective despite its perfect calibration.
> > > In this regard, to make a reliable comparison of calibration, we have to compare only models that have similar predictive performance.
> > > In our evaluation, we look for models that consider high accuracy with low expected calibration error.
> > >
> > > We hope this explanation meets your needs.
> > >
> > > # Reference
> > > - Tao, L., Zhu, Y., Guo, H., Dong, M., \& Xu, C. (2023). A benchmark study on calibration.

---

> > > > ### Comment · Reviewer_vWb7 · 2025-08-07
> > > >
> > > > Thank you for your reply and providing the reference. I do not believe it actually answers about Dirichlet Distillation performance, as its' accuracy is rather similar to the best results for its' outperformance in terms of ECE to be ignored. However, as I said, I appreciate it can be difficult to find out the reason especially during the short period of discussions.

---

### Note · Authors · 2025-08-15

We thank the AC and referees for your careful evaluations and constructive discussions. Our goal is to provide a new distribution distillation method that preserves teacher ensemble uncertainty while reducing computational cost.

## Baselines and completeness

* We added comparisons to Dirichlet distillation (Proxy-EnD²) and EDFM (Flow Matching).
* On CIFAR-10/100, DLF improves accuracy and NLL. Proxy-EnD² attains lower ECE, and we are further investigating the relationship between accuracy and ECE.
* For OOD detection using epistemic uncertainty with matched backbones, DLF consistently outperforms Proxy-EnD².

## Computation and scalability

* We reported MACs, latency for inference, and training time for all methods.
* DLF matches the light footprint of Hydra and Proxy-EnD², because training is practical due to closed-form E-steps.
* Our FLOPs analysis shows that the shared backbone dominates the cost, so Hydra and DLF require fewer FLOPs than BE.

## Prediction and uncertainty

* We clarified the inference procedure in our response and will update the final version accordingly.
* We explained how epistemic and aleatoric components are separated in our uncertainty estimation and use epistemic uncertainty for OOD detection.

## Modeling rationale and advantages

* DLF serves as an effective proxy distribution for teacher ensembles, offering broad function support, stable learning via EM, and flexible downstream use.
* We clarified the theory and motivation behind design-point selection, and we will revise this section in the final version.

## Larger-scale updates

* We have reported results on Tiny-ImageNet and are currently running experiments on the large ImageNet dataset.
* We add initial ARC-Challenge results (LLaMA-2-7B → MobileLLM), indicating applicability to modern language models. Broader comparisons will be included in the revision.

 |Model|Acc(%) ↑|NLL ↓|ECE(%) ↓|
 |--|--|--|--|
 |Teacher|69.60|1.5313|21.86|
 |Small-Ens|31.94|1.5724|16.22|
 |Hydra|29.51|1.8566|31.14|
 |LBE|31.25|3.1763|57.99|
 |Proxy-EnD²|29.51|1.6742|26.73|
 |KD|31.86|1.6316|20.29|

Please refer to the result instead of the report in the “Author AC Confidential Comment”.

## Presentation

* We revised notation, equation numbering, and citations, and toned down wording where appropriate.

We appreciate referees who updated their scores after these additions. We will integrate all clarifications and results into the final manuscript.
Thank you for your consideration.

---

### Decision · Program_Chairs · 2025-09-17

**Decision:**

Accept (poster)

**Comment:**

The paper is interested in the task of distilling an ensemble of deep networks into a single model for computational efficiency while keeping the desirable properties of an ensemble, particularly its predictive uncertainty quality. It uses student DNNs to model the mean and covariance of the teacher ensemble treated as a Gaussian process.  It tests the approach on various small datasets from UCI as well as CIFAR10, CIFAR100, and GLUE. It favorably evaluates the proposed model in terms of accuracy, negative log likelihood, and calibration error on their test data.

There was extensive rebuttal provided and the reviewers discussed it with the authors. After the rebuttal, all reviewers lean towards acceptance. During AC-reviewers discussion, all reviewers acknowledge the relevance and novelty of the work, clarity of the presentation, and significance of the obtained results with respect to prior work. Their only remaining concern/question is the transfer of the improved performance to larger benchmarks and/or more complex tasks. The authors provided additional results on Tiny ImageNet during the review process and have promised to add results on ImageNet.

The AC agrees with the reviewers that the strengths of the paper outweigh the criticism on missing large benchmarks. The fields of uncertainty estimation and distillation are of wide interest in academia and relevance in industry, and this paper makes a step forward. Therefore, the AC suggests acceptance.

The AC strongly recommends the authors to include the promised ImageNet results and possibly more large benchmarks and refer to the following currently-missing but directly-related works in the final version (and ideally compare with some of them as baselines):

[1] "Bayesian Dark Knowledge", NIPS 2015

[2] "Efficient Evaluation-Time Uncertainty Estimation by Improved Distillation", ICML 2019

[3] "A Simple Approach to Improve Single-Model Deep Uncertainty via Distance-Awareness", JMLR 2022

[4] "Ensemble Distribution Distillation via Flow Matching", ICML 2025

[5] "Contextual similarity distillation: Ensemble uncertainties with a single model", arXiv 2025